# A Counterfactual Semantics for Hybrid Dynamical Systems

**Andy Zane**[1,2]
andy@basis.ai

**Dmitry Batenkov**[1]
dima@basis.ai

**Rafal Urbaniak**[1]
rafal@basis.ai

**Jeremy Zucker**[3]
jeremy.zucker@pnnl.gov

**Sam Witty**[4]*
sam@sorbus.ai

[1]Basis Research Institute, New York, NY
[2]University of Massachusetts Amherst, Amherst, MA
[3]Pacific Northwest National Laboratory, Richland, WA
[4]Sorbus AI

## Abstract

Models of hybrid dynamical systems are widely used to answer questions about the causes and effects of dynamic events in time. Unfortunately, existing causal reasoning formalisms lack support for queries involving the dynamically triggered, discontinuous interventions that characterize hybrid dynamical systems. This mismatch can lead to ad-hoc and error-prone causal analysis workflows in practice. To bridge the gap between the needs of hybrid systems users and current causal inference capabilities, we develop a rigorous counterfactual semantics by formalizing interventions as transformations to the constraints of hybrid systems. Unlike interventions in a typical structural causal model, however, interventions in hybrid systems can easily render the model ill-posed. Thus, we identify mild conditions under which our interventions maintain solution existence, uniqueness, and measurability by making explicit connections to established hybrid systems theory. To illustrate the utility of our framework, we formalize a number of canonical causal estimands and explore a case study on the probabilities of causation with applications to fishery management. Our work simultaneously expands the modeling possibilities available to causal inference practitioners and begins to unlock decades of causality research for users of hybrid systems.

## 1 Introduction

Models of continuous-time dynamical systems are powerful tools for describing real-world mechanisms. From contrastive queries about system behavior under different control policies (Kirk, 2004), to sensitivity analyses designed to aid in understanding which parameters drive system variation (Cacuci, 2003), scientists, policy makers, and engineers often use such models to answer their "what-if" and causal questions. Unfortunately, causal reasoning with continuous-time systems can be ad-hoc, manual, and error-prone in daily practice.

In parallel, researchers in causal inference have built rigorous tools for answering an expansive taxonomy of causal queries. For example, causal questions about effect estimation (Pearl, 2009; Rubin, 1974; Imbens & Rubin, 2015), counterfactual reasoning (Pearl, 2009, Ch. 7), mediation

---

*Research conducted at Basis.

39th Conference on Neural Information Processing Systems (NeurIPS 2025).

analysis (Pearl, 2001), responsibility, blame (Chockler & Halpern, 2004), attribution, and explanation (Halpern & Pearl, 2005a,b; Beckers, 2022) can all be succinctly expressed as estimands constructed from parallel worlds (Balke & Pearl, 1994; Avin et al., 2005; Shpitser & Pearl, 2008) or potential outcomes (Rubin, 1974). The toolkit also affords a formal means of determining when those estimands can be reduced to computationally tractable, probabilistic estimation problems (Pearl, 1995; Shpitser & Pearl, 2006; Hernán & Robins, 2023). These insights have made it possible to build general-purpose technology for causal reasoning, such as the causal probabilistic programming language ChiRho (Bingham et al., 2021; Witty, 2023; Basis-Research, 2025).[2]

Despite significant progress over the last decade (Mooij et al., 2013; Hansen & Sokol, 2014; Blom et al., 2019; Forré & Mooij, 2020; Peters et al., 2020; Blom et al., 2021; Bongers, 2022; Blom & Mooij, 2023; Boeken & Mooij, 2024; Peters & Halpern, 2025), however, gaps remain in the technical capacity of modern causal reasoning machinery to operate on the full breadth of interventions that can be encoded in continuous-time dynamical systems. In particular, a counterfactual semantics for dynamically triggered, instantaneous intervention has not yet been established. With such an intervention semantics in hand, causal reasoning can be more fully mechanized for causal questions about dynamic temporal events, dramatically expanding the rigor and variety of queries available to users of continuous-time dynamical systems.

Such interventions underpin many closed-loop control problems: for example, HVAC systems activate when temperature thresholds are reached; lockdown and masking measures can be implemented according to levels of Sars-CoV-2 in wastewater (Kappus-Kron et al., 2024); commercial fishing pressure can be reduced once annual harvest limits are reached (Anon, 2007b; Warlick et al., 2018); central banks adjust interest rates depending on economic indicators like inflation and unemployment; reservoir managers release water depending on storage thresholds and agricultural needs (Ray, 2003); and power grids activate "peaker plants" (or stored energy) when demand exceeds certain thresholds (Zhuk et al., 2016). Despite limited attention from the causality community, these systems have garnered significant interest from control theorists in the form of continuous-time, *hybrid* dynamical systems (Schaft et al., 2000; Goebel et al., 2012; Sanfelice, 2021) that encode both continuous and instantaneous dynamics in a set of differential and difference constraining equations.

To construct a counterfactual semantics for state-dependent, instantaneous intervention, we formalize a class of transformations on hybrid system constraints that induce the desired counterfactual behavior. An intervention creates a twin, parallel world with transformed constraints, but in a way that ensures both the twin and original worlds share randomly sampled values for initial conditions and parameters. This induces a familiar joint distribution over counterfactual outcomes (Rubin, 1974; Balke & Pearl, 1994; Shpitser & Pearl, 2006, 2008) that can, in turn, be used as input to established causal estimands, such as an expected treatment effect or the probabilities of necessary and sufficient causation.

Our contributions are:

1. A formal, counterfactual semantics for dynamically triggered, instantaneous interventions in continuous-time dynamical systems.

2. Under minimal requirements on interventional specifications, proof that sufficient conditions for solution existence, uniqueness, and finite-time measurability are preserved in the intervened system. Our framework also explicitly connects to established well-posedness conditions on hybrid dynamical systems.

3. A case study on the probabilities of necessary and sufficient causation applied to fishery management, demonstrating extensibility to non-trivial causal estimands rarely applied to dynamical systems.

## 2   Related Work

**In Causality.**   Many researchers have contributed to the systematization of causality for dynamical systems. Hansen & Sokol (2014), for example, show that dynamical systems can be unrolled into directed, structural causal models (SCMs).[3] In the context of ordinary differential equations (ODEs),

---

[2] https://github.com/BasisResearch/chirho

[3] Structural causal models (SCMs) are systems of deterministic functions on endogenous variables that additionally incorporate exogenous random noise. SCMs come equipped with a widely studied interventional semantics. We refer the reader to highly influential work of Pearl (2009) for further background.

if $f$ is the right-hand side of the continuous-time differential equation $x' = f(x)$, we can write structural equations $x_t = x_{t-\Delta t} + f(x_{t-\Delta t}, u)\Delta t$, where $t \geqslant 0$, $x_t \in \mathbb{R}$ is the value of the state variable $x$ at time $t$, $u \in \mathbb{R}$ is a fixed realization of exogenous noise, and $x_0$ is fixed. Taking $\Delta t \to 0$, we can recover the system's dynamics arbitrarily well. This limit results in SCMs with infinitely many variables — a modality that has been recently studied as "Generalized Structural Equation Models" (GSEMs) (Peters & Halpern, 2021; Halpern & Peters, 2022; Peters & Halpern, 2025). With $\Delta t > 0$, this becomes the familiar discrete time approximation, which has been widely researched in causal inference (Spirtes, 2013; Pearl, 2009; Murphy, 2002; Wang et al., 2018; Assaad et al., 2022; Runge et al., 2023; Zan et al., 2024).

This forward-Euler representation, however, is not the preferred tool of hybrid systems theorists, making it ill-suited for identifying conditions under which intervention preserves established well-posedness conditions. Additionally, under the forward-Euler representation, interventional transformations that induce *state-dependent* jumps require "soft" intervention (Correa & Bareinboim, 2020) on *all* endogenous nodes that might jump. Indeed, the state-dependent jump conditions must be "checked" at all points in time. We discuss this more precisely in appendix J.

Somewhat sidestepping the temporal representation issue, most causal research on continuous-time dynamical systems has employed foundational ideas in cyclic graphical models (Iwasaki & Simon, 1994; Spirtes, 2013; Lacerda et al., 2008; Hyttinen et al., 2012) to develop causal abstractions of a system's *equilibrium* behavior (Dash, 2003; Mooij et al., 2013; Hansen & Sokol, 2014; Blom et al., 2019; Forré & Mooij, 2020; Bongers, 2022; Blom & Mooij, 2023). Equilibrium-focused frameworks, however, can fail to expose complex causal relationships in transient dynamics (Peters et al., 2020).

Extensions such as the "time-splitting" operation (Boeken & Mooij, 2024), or the application of GSEMs to hybrid automata by Peters & Halpern (2025), enhance the expressiveness of graphical approaches by supporting static-time discontinuities. In contrast, our work targets dynamically triggered interventions, which cannot be straightforwardly analyzed using methods like time-splitting. Indeed, the order — and, therefore, the induced time-split graph structure — of dynamically triggered interventions depends on state evolution, and therefore on exogenous noise. Our approach avoids these issues by directly defining counterfactual interventions on hybrid system constraints. While the non-graphical framing means that standard graphical identifiability criteria are not immediately available, building our semantics on established hybrid systems theory opens pathways to leveraging longstanding methods and conditions for system identification of dynamical systems (Walter & Pronzato, 1997; Ljung, 2012; Raue et al., 2009; Stuart, 2010), such as the "persistence of excitation", which has been studied directly in the context of hybrid systems (Johnson, 2023; Saoud et al., 2024).

Our approach follows the spirit of recent developments in constraint-based causal modeling. For example, Beckers et al. (2023) extend SCMs in order to handle logical constraints (such as unit conversions), while Blom et al. (2019) interpret equilibrium equations of dynamical systems, along with their corresponding algebraic invariants, as a collection of constraints. In both cases, a model is characterized by a collection of constraints, and interventions are defined as transformations of those constraints (e.g., by changing, disabling, or enabling them). At a high level, we take a similar approach. Hybrid systems, however, are characterized by a unique class of constraints requiring special considerations around Zeno behavior, set-valued theory, non-uniqueness even in "well-posed" cases, set-valued stable points, etc. In short, analyzing the post-intervention properties of hybrid systems is made easier via direct use of existing hybrid systems frameworks, rather than existing causal frameworks. Naturally, each school of thought is best suited to different tasks, and we look forward to future work that deftly exercises the comparative advantages of each.

**In Control Theory.** Control theory and causality share overlapping goals, yet historically operate separately. This paper integrates causal reasoning directly into established, hybrid dynamical systems frameworks (Goebel et al., 2012; Sanfelice, 2021). In particular, our formalization of dynamically triggered intervention as constraint transformation mirrors controller-plant compositions from hybrid control theory, which are also shown to preserve established conditions for system well-posedness (Sanfelice, 2021). Hybrid system theory presents challenges, however, due to potential non-uniqueness of solutions under general conditions (Goebel et al., 2012), complicating counterfactual reasoning. To address this practically, we follow common simulation practices (e.g., preferring flowing solutions when multiple are possible) and explicitly formalize these assumptions (Sanfelice et al., 2023a). Our contributions thus link causal semantics to established hybrid systems theory and practice, enabling rigorous and computationally feasible causal analysis.

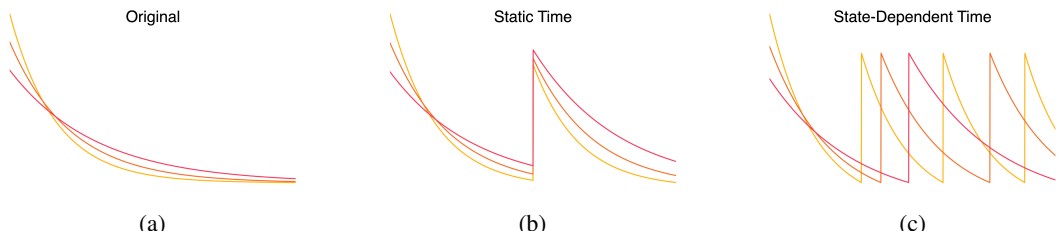

| Original | Static Time | State-Dependent Time |
| --- | --- | --- |
| (a) | (b) | (c) |

Figure 1: Three parallel worlds constructed by starting with a dose-decay model (fig. 1a) and then transforming that model to reflect dosage at a fixed, static time (fig. 1b), and dosage when the concentration hits a threshold (fig. 1c and example 1). This paper develops the first explicitly counterfactual semantics for the dynamically triggered, state-dependent case (fig. 1c). Three sample trajectories are shown for each world, with initial condition and dose-decay rate held fixed across worlds for each sample trajectory. Notice that the state-dependent interventions occur at different times for different trajectories induced by different initial conditions and/or parameters.

# 3  Parameterized Hybrid Systems

As a first approximation, the present work focuses on continuous, ordinary differential equations models with random initial conditions and parameters. Many intuitive interventions, however, can be conveniently defined as *instantaneous* (discontinuous) changes to the dynamically evolving state. Thus, we focus on *hybrid* systems that afford both continuous "flow" and event-based "jumps" in state. Jumps can arise as a product of interventions and/or discontinuous dynamics in the unintervened system. With state space $\mathcal{S} \subseteq \mathbb{R}^n$ and following the framework laid out by Goebel et al. (2012), many hybrid systems can be characterized as comprising four elements: a flow set $C \subset \mathcal{S}$; a differential inclusion $F : \mathcal{S} \rightrightarrows \mathbb{R}^n$; a jump set $D \subset \mathcal{S}$; and a set-valued jump map $G : \mathcal{S} \rightrightarrows \mathcal{S}$. In general, the system evolves according to its differential inclusion $F$ when its state is in the flow set $C$ and according to the jump map $G$ when in the jump set $D$. Readers who are unfamiliar with inclusions and set-valued maps should refer to appendix A.1. Hybrid systems often alternate between continuous flow and discontinuous jumps, though consecutive jumps remain well-defined. Many hybrid systems, then, can be characterized with the tuple $(C, F, D, G)$. We ground this out in the following example.

**Example 1** (Dosage Model). Consider modeling the exponential decay of drug concentration $x$ at rate $\beta$, where medical providers intervene to administer additional dosage when $x$ reaches a threshold $\gamma$. To model these dynamics, we can seek state evolutions obeying

$$\begin{cases} x \in C = \mathbb{R} \backslash D & \dot{x} \in F(x) = \{-\beta x\} \\ x \in D = \{x \in \mathbb{R} : x \leqslant \gamma\} & x^+ \in G(x) = \{x + 1\} \end{cases}$$

where $\dot{x}$ denotes the time derivative of the state, and $x^+$ the state immediately following a jump. The solution map of a hybrid system typically takes as "input" an initial condition $\boldsymbol{\xi} \in \mathcal{S}$, but can also be parameterized to additionally incorporate a vector $\boldsymbol{\theta}$ — in example 1, $\boldsymbol{\theta} = [\beta, \gamma]$.

**Definition 1** (Parameterized Hybrid System). Let $\mathcal{S} \subseteq \mathbb{R}^n$, $\Theta \subseteq \mathbb{R}^m$. A parametrized hybrid system $\mathcal{P}$ is a tuple $\mathcal{P} = (\mathcal{H}, \mathcal{S}, \Theta)$ where for each $\boldsymbol{\theta} \in \Theta$, $\mathcal{H}(\boldsymbol{\theta}) = (C(\boldsymbol{\theta}), F_{\boldsymbol{\theta}}, D(\boldsymbol{\theta}), G_{\boldsymbol{\theta}})$ is a standard hybrid system (Goebel et al., 2012, Def. 2.2), i.e.

- $C : \Theta \rightrightarrows \mathcal{S}$ is a set-valued mapping returning the flow set,

- $F_{\boldsymbol{\theta}}(\boldsymbol{x}) = F(\boldsymbol{x}, \boldsymbol{\theta}) \; \forall \boldsymbol{x} \in \mathcal{S}$ and $\forall \boldsymbol{\theta} \in \Theta$, where $F : \mathcal{S} \times \Theta \rightrightarrows \mathbb{R}^n$ is a differential inclusion, with $C(\boldsymbol{\theta}) \subset \operatorname{dom} F_{\boldsymbol{\theta}} \subseteq \mathcal{S}$ for all $\boldsymbol{\theta} \in \Theta$,

- $D : \Theta \rightrightarrows \mathcal{S}$ is a set-valued mapping returning the jump set, and

- $G_{\boldsymbol{\theta}}(\boldsymbol{x}) = G(\boldsymbol{x}, \boldsymbol{\theta}) \; \forall \boldsymbol{x} \in \mathcal{S}$ and $\forall \boldsymbol{\theta} \in \Theta$, where $G : \mathcal{S} \times \Theta \rightrightarrows \mathcal{S}$ is an ordered (i.e. returning an ordered collection of sets to keep track of interventions, cf. definition 6) set-valued jump map, with $D(\boldsymbol{\theta}) \subset \operatorname{dom} G_{\boldsymbol{\theta}} \subseteq \mathcal{S}$ for all $\boldsymbol{\theta} \in \Theta$.

Without explicit parametrization, we write $\mathcal{H} = (C, F, D, G)$, and also often expand $\mathcal{H}$ in $\mathcal{P}$, writing equivalently $\mathcal{P} = (\mathcal{H}, \mathcal{S}, \Theta) = (C, F, D, G, \mathcal{S}, \Theta)$.

Canonically, solutions to hybrid systems are functions of both continuous time $t \in \mathbb{R}_{\geqslant 0}$ and discrete event indices $j \in \mathbb{N}$. Following (Goebel et al., 2012, Sects. 2.2–2.3), we define, for each possible parameterization $\boldsymbol{\theta} \in \Theta$ and initial condition $\boldsymbol{\xi} \in \mathcal{S}$, a "solution" to $\mathcal{H}(\boldsymbol{\theta})$ to be a "hybrid arc", which is formally a set-valued map $\phi(\cdot; \boldsymbol{\xi}, \boldsymbol{\theta}) : \mathbb{R}_{\geqslant 0} \times \mathbb{N} \rightrightarrows \mathbb{R}^n$. We review Goebel et al.'s (2012) rigorous characterization of hybrid arcs as solutions to hybrid systems in appendix A.3. For ease of exposition in the main body of this paper, however, we use the concept of a time-parameterized solution map $\varphi$, which we describe informally, below, in definition 2. An expanded, formal treatment of time-parameterized solution maps can be found in appendix A.4.

**Definition 2** ((Informal) Time-Parameterized Solution Map). Let $\varphi(\cdot; \boldsymbol{\xi}, \boldsymbol{\theta}) : [0, t^+) \to \mathbb{R}^n$ be called the time-parameterized solution map of $\mathcal{P} = (\mathcal{H}, \mathcal{S}, \Theta)$, where $t^+ = \min_{\boldsymbol{\xi}, \boldsymbol{\theta}} \sup_t \operatorname{dom} \phi(\cdot; \boldsymbol{\xi}, \boldsymbol{\theta})$ and where the hybrid arc $\phi(\cdot; \boldsymbol{\xi}, \boldsymbol{\theta})$ uniquely satisfies $\mathcal{H}(\boldsymbol{\theta})$ from initial state $\boldsymbol{\xi}$, $\forall \boldsymbol{\xi}, \boldsymbol{\theta} \in \mathcal{S} \times \Theta$.

The reader will note that $[0, t^+) \subset \mathbb{R}$. In this paper, we focus strictly on finite time horizons, leaving the analysis of hybrid equilibria to future work — indeed, only the simplest hybrid systems equilibrate to a point, so equilibrium states are most productively defined as belonging to a set. Analyzing the causally relevant behavior of such sets requires machinery beyond our current scope, but our direct connection to established hybrid systems theory, in conjunction with the rich history of causal research on equilibrium models, provides a firm foundation to explore this in the future. Additionally, because hybrid arcs can dynamically evolve in event indices, Zeno and non-flowing solutions are possible, which can make $t^+ = 0$ (if it only jumps) or arbitrarily small (if allowable initial conditions are close to Zeno accumulation points). We do not provide universal criteria in this paper under which $t^+$ is arbitrarily large.

While we take the hybrid system $\mathcal{P}$ to accurately describe causally relevant mechanisms in the world, we impose assumptions on $\mathcal{P}$ *indirectly*. In particular, we assume that some auxiliary "upstream" system $\mathcal{P}_\uparrow$ can be "lowered" to produce $\mathcal{P} = \texttt{lower}(\mathcal{P}_\uparrow)$, and that the upstream $\mathcal{P}_\uparrow$ satisfies standard hybrid well-posedness conditions from the literature (the so-called *hybrid basic conditions*, detailed in assumption 4 of the appendix, and folded into assumption 1 below). While these conditions support our theoretical results and facilitate future extensions (e.g., to stability analyses), they inherently admit solution non-uniqueness, particularly at state-space boundaries where solutions could either jump or flow. Non-uniqueness, however, complicates both measurability arguments and downstream causal analysis. In this work, then, we formalize a practical approach that is standard in *simulating* hybrid systems by specifying that the solutions should be "flow preferring" — if a solution could both flow and jump, we choose the solution that flows (Sanfelice et al., 2023a; Sanfelice & Teel, 2010).[4] Note also that a flow-preferring specification is consistent with computational implementations that trigger jumps when the jump-set boundary is *crossed*.[5]

A key component of the hybrid basic conditions is the outer semi-continuity of the jump set $G$ in the upstream system. Maintaining this property through intervention requires some bookkeeping on the boundaries between interventional jump sets, but must be handled such that "lowering" favors more recently applied model transformations. We achieve this bookkeeping through the use of an ordered set-valued map $G = x \mapsto \bigcup_{k=1}^{K} G_k(x)$, where $\texttt{last}(G) = G_K$. We fully formalize the ordered set-valued map in the appendix (c.f. definition 6).

**Definition 3.** Let $\mathcal{P} = (C, F, D, G, \mathcal{S}, \Theta) = (\mathcal{H}, \mathcal{S}, \Theta)$. We set

$\texttt{preferflow}(D, C, F) = \boldsymbol{\theta} \mapsto D(\boldsymbol{\theta}) \backslash \{\boldsymbol{\xi} \in \mathcal{S} : \text{there is a flowing solution to } \mathcal{H}(\boldsymbol{\theta}) \text{ from } \boldsymbol{\xi}\}$;

$$\texttt{lower}(\mathcal{P}) = (C, F, D', G', \mathcal{S}, \Theta); \qquad D' = \texttt{preferflow}(D, C, F), \ G' = \texttt{last}(G).$$

The existence of a flowing solution from $\boldsymbol{\xi}$ is meant in the sense established in the hybrid systems literature. See appendix A.5 (definitions 13 and 14) for details. We can now state our collected assumptions on a hybrid system, and prove the sufficiency of those assumptions for the existence, uniqueness, and measurability of the system's solution. See appendix A.6 (assumptions 3 to 5) for details, and appendix F for proof of lemma 1.

**Assumption 1.** The parameterized hybrid system can be written as $\mathcal{P} = \texttt{lower}(\mathcal{P}_\uparrow)$, where $\mathcal{P}_\uparrow = (C, F, D, G, \mathcal{S}, \Theta) = (\mathcal{H}, \mathcal{S}, \Theta)$, and the following hold for all $\boldsymbol{\xi} \in \mathcal{S}, \boldsymbol{\theta} \in \Theta$:

---

[4]Other approaches include preferring solutions that jump, or by resolving ambiguities randomly (Teel & Hespanha, 2015).

[5]We should say, of a *thick* jump set, similar to what we have described in definition 17. Thickening jump sets is also common in practical computational environments (Sanfelice et al., 2023b).

1. there exists a unique, nontrivial solution to the differential inclusion $F$ (i.e. the *continuous* part of $\mathcal{H}(\boldsymbol{\theta})$) that is Borel measurable in $\boldsymbol{\xi}, \boldsymbol{\theta}$ at any fixed $t \in [0, \infty)$;[6]

2. $C$ is outer semi-continuous (osc) and $C(\boldsymbol{\theta})$ closed; $F$ is osc, locally bounded, and $F(\boldsymbol{x}, \boldsymbol{\theta})$ is convex $\forall \boldsymbol{x} \in \mathcal{S}$;

3. $D(\boldsymbol{\theta})$ is closed and $\mathcal{G}(D)$ is Borel; $G$ is osc, locally bounded; $\texttt{last}(G)$ is single-valued and Borel measurable in $\boldsymbol{\xi}, \boldsymbol{\theta}$ at any fixed $t \in [0, \infty)$.

**Lemma 1.** *Let $\mathcal{P}$ satisfy assumption 1. Then $\mathcal{P}$ has a unique time-parameterized solution $\varphi(\cdot; \boldsymbol{\xi}, \boldsymbol{\theta}) : [0, t^{+}) \to \mathbb{R}^n$ that is Borel-measurable in initial conditions $\boldsymbol{\xi}$ and parameters $\boldsymbol{\theta}$ at any fixed $t \in [0, t^{+})$.*

## 4 Instantaneous Interventions as Constraint Transformations

We now formally define a general class of instantaneous interventions. We show that under certain natural assumptions, the class of systems meeting assumption 1 is "closed" under intervention — i.e., intervened systems will meet assumption 1 if the original system does.

An instantaneous intervention can be implemented via modifications to the jump map and the jump set functions, respectively $G$ and $D$ in definition 1. To support parameterized interventions and/or stateful jumps, one can simply augment the state space $\mathcal{S}$ and the parameter space $\Theta$, essentially preserving all properties of interest (appendix B).[7]

**Definition 4** (Instantaneous Intervention). Consider set-valued mappings $\tilde{D} : \Theta \rightrightarrows \mathcal{S}$ and $\tilde{G} : \mathcal{S} \times \Theta \rightrightarrows \mathcal{S}$ and parameterized hybrid system $\mathcal{P} = (C, F, D, G, \mathcal{S}, \Theta)$. Now, let

$$C'(\boldsymbol{\theta}) = C(\boldsymbol{\theta}) \backslash \text{int } \tilde{D}(\boldsymbol{\theta}) \tag{1}$$

$$D'(\boldsymbol{\theta}) = \texttt{preferflow}\left(\tilde{D}, C', F\right)(\boldsymbol{\theta}) \cup D(\boldsymbol{\theta}) \tag{2}$$

$$G'(\boldsymbol{x}, \boldsymbol{\theta}) = \begin{cases} \boldsymbol{x} \in D(\boldsymbol{\theta}) \backslash \tilde{D}(\boldsymbol{\theta}) & G(\boldsymbol{x}, \boldsymbol{\theta}) \\ \boldsymbol{x} \in \tilde{D}(\boldsymbol{\theta}) & \tilde{G}(\boldsymbol{x}, \boldsymbol{\theta}) \end{cases} \tag{3}$$

then $\qquad \mathcal{P}' = \texttt{instint}\left(\mathcal{P}, \tilde{D}, \tilde{G}\right) = \left(C', F, D', G', \mathcal{S}, \Theta\right).$

In words, $\tilde{D}$ defines *when* (or *where* in the state space) the intervention will occur, while $\tilde{G}$ defines the state transition induced by the intervention. We make two important set-subtractions in this definition to preserve some useful and simplifying properties. First, we define $G'$ in eq. (3) as preferring the new (i.e., the interventional) jump map $\tilde{G}$ wherever the original and new jump sets overlap. Second, the new flow set in eq. (1) has the interior of the new jump set removed. This preserves non-overlap between flow and jump sets, except possibly on the boundary, which we discuss below.

$D'(\boldsymbol{\theta})$ is defined (eq. (2)) as the union over jump sets, except that a flow-preferring subtraction (definition 3) is made first on $\tilde{D}(\boldsymbol{\theta})$. Because $C'(\boldsymbol{\theta})$ has the interior of $\tilde{D}(\boldsymbol{\theta})$ already removed, this subtraction operates only on the boundary of $\tilde{D}(\boldsymbol{\theta})$. In other words, $D'(\boldsymbol{\theta})$ will always contain the interior of $\tilde{D}(\boldsymbol{\theta})$, and will have the parts of its boundary removed where $F_{\boldsymbol{\theta}}$ flows tangentially to or away from the interventional jump set.

### 4.1 Intervention Preserves Existence, Uniqueness, and Measurability

Interventional transformations should preserve key model properties. For causal models with explicit forward simulations, this is largely trivial. Hybrid systems, however, only *implicitly* characterize forward simulations (i.e., solutions) by specifying a set of constraints. Transformations to these constraints can easily fail to maintain key properties in the intervened world, such as whether a

---

[6]This assumption is insufficient, on its own, to guarantee a unique, measurable solution for the *full* hybrid system. Indeed, this insufficiency constitutes a key challenge addressed by hybrid systems researchers.

[7]This covers interventions with random parameters, in addition to those that require some "memory" of past system events. For example, if a jump event should only occur $k \in \mathbb{N}$ times, that event could increment a counter $i$, and include only $i < k$ in its interventional jump set.

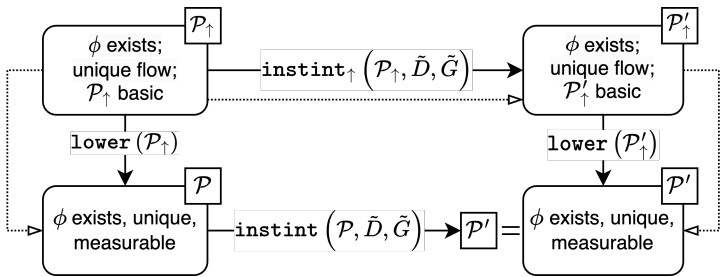

Figure 2: Depiction of our "lowering" proof strategy for theorem 1. Proving theorem 1 requires a simple inductive generalization from lemma 4, which asserts that a single interventional transformation preserves key system properties, and is what we visualize here. Solid arrows indicate constraint transformations, while dotted arrows indicate that properties of one system imply properties of another. Assume that the parameterized hybrid system $\mathcal{P}$ accurately describes a domain of interest, and that it can be constructed by applying the `lower` transformation (definition 3) to a system $\mathcal{P}_\uparrow$ that fulfills assumption 1. Applying `lower` to such a system preserves existence and induces uniqueness and measurability (lemma 1). To simulate the effects of an intervention, we transform $\mathcal{P}$ into the model $\mathcal{P}'$ that describes the intervened world. $\mathcal{P}'$ can also be constructed, however, by applying a slightly modified intervention (`instint`$_\uparrow$, definition 20) to $\mathcal{P}_\uparrow$ and then "lowering". Intervention on $\mathcal{P}_\uparrow$ maintains key properties in $\mathcal{P}'_\uparrow$, which can, as before, be lowered to a system $\mathcal{P}'$ that must have a unique, measurable solution (lemma 1).

solution is unique and measurable, or exists at all. The key theoretical contribution of this work, then, is to identify assumptions sufficient to ensure our interventional semantics preserves these properties through model transformation. Formal proof of theorem 1 is provided in appendix D, but we also include fig. 2 as a visual aid and proof sketch.

**Assumption 2** (Assumptions on Interventional Specifications). Consider mappings $\tilde{D} : \Theta \rightrightarrows \mathcal{S}$ and $\tilde{G} : \mathcal{S} \times \Theta \rightrightarrows \mathcal{S}$ and parameterized hybrid system $\mathcal{P} = (C, F, D, G, \mathcal{S}, \Theta)$. For all $\boldsymbol{\theta} \in \Theta$, assume

(I1) $\tilde{D}(\boldsymbol{\theta})$ is closed and well-behaved relative to $\mathcal{P}$ (assumption 6).[8] Additionally, the interior graph $\mathcal{G}(\text{int } \tilde{D})$ is open[9] and the graph $\mathcal{G}(\tilde{D})$ is Borel;

(I2) $\tilde{G}_{\boldsymbol{\theta}} : \mathcal{S} \rightrightarrows \mathcal{S}$ is outer semi-continuous and locally bounded relative to $\tilde{D}(\boldsymbol{\theta})$, and $\tilde{D}(\boldsymbol{\theta}) \subset \text{dom } \tilde{G}_{\boldsymbol{\theta}}$. Additionally, $\tilde{G}$ is single-valued and Borel-measurable.

**Theorem 1** (Compositions of Instantaneous Interventions Preserve Key Properties). *Consider parameterized hybrid system $\mathcal{P}$ that meets assumption 1, and any finite sequence of $K$ set-valued mappings $(\tilde{D}_k)$ and $(\tilde{G}_k)$, where each $\tilde{D}_k$ and $\tilde{G}_k$ fulfill assumption 2 relative to $\mathcal{P}$. Let* $\text{instint}_k = \text{instint}(\cdot, \tilde{D}_k, \tilde{G}_k)$ *(definition 4) and*

$$\mathcal{P}' = \left( \text{instint}_1 \circ \cdots \circ \text{instint}_k \circ \cdots \circ \text{instint}_K \right)(\mathcal{P}), \tag{4}$$

*$\mathcal{P}'$ then meets assumption 1, and by lemma 1 $\mathcal{P}'$ has a unique time-parameterized solution $\varphi(\cdot; \boldsymbol{\xi}, \boldsymbol{\theta}) : [0, t^+) \to \mathbb{R}^n$, Borel-measurable in initial conditions $\boldsymbol{\xi}$ and parameters $\boldsymbol{\theta}$ at any fixed $t \in [0, t^+)$.*

## 5 Causal Estimands as Functionals of Twin Distributions

In this section, we exercise our framework to define three basic causal estimands. Importantly, many of the more complex causal analyses build on these basic inference capabilities. Most targets of causal inference take the form of (conditional) expectations, and so we must now use our measurability results to define those expectations with respect to random solution maps.[10] First, we will generalize

---

[8]An assumption that the interventional jump set is well-behaved reduces, essentially, to asserting that a flowing solution cannot oscillate across the boundary of $\tilde{D}$ infinitely often. This is satisfied by many systems of interest under reasonable regularity assumptions – for instance, if the flow map is analytic and $\partial \tilde{D}$ is Lipschitz.

[9]This ensures its interior does not suddenly appear/disappear as $\boldsymbol{\theta}$ varies.

[10]In this paper, we do not address the random dynamics that characterize stochastic differential constraints (Øksendal, 2003; Cassandras & Lygeros, 2010; Hansen & Sokol, 2014; Boeken & Mooij, 2024), or independent, per-jump randomness (Teel, 2013; Teel et al., 2014; Teel & Hespanha, 2015).

the parameterized hybrid system to include random initial conditions and parameters. Then, we will establish some notation and define the expected treatment effect, data-conditional treatment effect, and the basic counterfactual query using our machinery.

**Definition 5** (Hybrid System with Random Inputs). A parameterized hybrid system with random inputs is characterized by the tuples

$$\mathcal{R} = (\mathcal{P}, \boldsymbol{\xi}, \boldsymbol{\theta}) \, ; \; \mathcal{P} = (C, F, D, G, \mathcal{S}, \Theta) \, .$$

We take the probability space $(\Omega, \mathcal{F}, \mathbb{P})$ as implied by $\mathcal{R}$, where $\boldsymbol{\xi} : \Omega \to \mathcal{S}$ and $\boldsymbol{\theta} : \Omega \to \Theta$ are measurable with respect to $\mathcal{F}$ and the Borel $\sigma$-algebras on $\mathcal{S} \subseteq \mathbb{R}^n$ and $\Theta \subseteq \mathbb{R}^m$.

When clear from context, for some $\omega \in \Omega$, we often write $\boldsymbol{\xi}$ and $\boldsymbol{\theta}$ in place of $\boldsymbol{\xi}(\omega)$ and $\boldsymbol{\theta}(\omega)$, respectively. We distinguish random variables $\boldsymbol{\xi}$ and $\boldsymbol{\theta}$ from possible values $\boldsymbol{\xi} \in \mathcal{S}$ and $\boldsymbol{\theta} \in \Theta$ by the upright font. From here, we can directly consider evaluations of the solution as a measurable random variable. A direct consequence of lemma 1, which states that `lower` induces measurability when applied to a system that fulfills assumption 1, is the following.

**Corollary 1** (Random Time-Parameterized Solution is Measurable). *Consider parameterized hybrid system with random inputs $\mathcal{R} = (\mathcal{P}, \boldsymbol{\xi}, \boldsymbol{\theta})$, where $\mathcal{P}$ satisfies assumption 1. Then, by lemma 1, $\mathcal{P}$ has a unique, time-parameterized solution map $\varphi$, and the composition $\omega \mapsto \varphi(t; \boldsymbol{\xi}(\omega), \boldsymbol{\theta}(\omega))$ defines an $\mathcal{F}$-measurable random variable at any fixed $t \in [0, t^+)$.*

Having established conditions under which intervention preserves measurability, we can begin constructing estimands from the parallel worlds created through intervention. In estimands, we use symbolic subscripts to delineate parallel worlds. Consider an original system $\mathcal{R}_0 = (\mathcal{P}_0, \boldsymbol{\xi}, \boldsymbol{\theta})$. We might then apply an intervention characterized by $\tilde{D}_s$ and $\tilde{G}_s$ to yield $\mathcal{P}_s = \texttt{instint}(\mathcal{P}_0, \tilde{D}_s, \tilde{G}_s)$. By convention, we use $\mathcal{R}_s = (\mathcal{P}_s, \boldsymbol{\xi}, \boldsymbol{\theta})$ in reference to the full specification for the intervened world, and $t \mapsto \varphi_s(t; \boldsymbol{\xi}, \boldsymbol{\theta})$ for its random, time-parameterized solution (corollary 1). We often write $\varphi_s^t$ in place of $\varphi_s(t; \boldsymbol{\xi}, \boldsymbol{\theta})$ for brevity. Lastly, supposing we wish to focus on a particular element of the state vector at time $t$, we sometimes define a random function that appropriately indexes into the solution vector. For example, we might have that $h_s(t; \boldsymbol{\xi}, \boldsymbol{\theta}) = \varphi_s^{(i)}(t; \boldsymbol{\xi}, \boldsymbol{\theta})$ always, where $h$ represents the solution map for the $(i)$'th state element. We similarly sometimes use $h_s^t = h_s(t; \boldsymbol{\xi}, \boldsymbol{\theta})$. We can exercise this notation with the following examples.

**Example 2** (Expected Treatment Effect). Consider $\mathcal{R}_0 = (\mathcal{P}_0, \boldsymbol{\xi}, \boldsymbol{\theta})$ and interventional jump set $\tilde{D}$ and map $\tilde{G}$. Assume these components fulfill assumptions 1 and 2. Let $\mathcal{P}_1 = \texttt{instint}(\mathcal{P}_0, \tilde{D}, \tilde{G})$ and $\varphi_0$ and $\varphi_1$ be the time-parameterized solution maps of the original and intervened worlds. Let $y_0$ and $y_1$ be the solution maps for the $(i)$'th element of the state vector. The expected treatment effect at some time $\tau \in [0, \min [t_0^+, t_1^+]) = [0, t^+)$ can be written equivalently as

$$\mathbb{E}\left[y_1^\tau - y_0^\tau\right] = \mathbb{E}\left[y_1(\tau; \boldsymbol{\xi}, \boldsymbol{\theta}) - y_0(\tau; \boldsymbol{\xi}, \boldsymbol{\theta})\right] = \mathbb{E}\left[\varphi_1^{(i)}(\tau; \boldsymbol{\xi}, \boldsymbol{\theta}) - \varphi_0^{(i)}(\tau; \boldsymbol{\xi}, \boldsymbol{\theta})\right].$$

**Example 3** (Data-Conditional Treatment Effect). Building immediately off example 2, we can specify a data-conditional treatment effect that takes factual observations into account.[11] Let $w_0$ be the solution map for some element of the state vector. For some finite set of observation times $\{t_k\}_{k=1}^K \subset [0, t_0^+)$, the data-conditional treatment effect can then be written as[12]

$$\mathbb{E}\left[y_1^\tau - y_0^\tau \mid \boldsymbol{v}_0\right]; \; \boldsymbol{v}_0 \sim \mathcal{N}\left(\boldsymbol{w}_0, \sigma^2\right); \; \boldsymbol{w}_0 = \left[w_0^{t_k}\right]_{k=1}^K.$$

**Example 4** (Counterfactual Outcome). Also building off example 2, consider factual outcome event that $y_0(\tau; \boldsymbol{\xi}, \boldsymbol{\theta}) = \bar{y}_0^\tau \in \mathbb{R}$. The counterfactual outcome, then, can be derived by conditioning on that factual event.

$$\mathbb{E}[y_1^\tau \mid y_0^\tau = \bar{y}_0^\tau] = \mathbb{E}[y_1(\tau; \boldsymbol{\xi}, \boldsymbol{\theta}) \mid y_0(\tau; \boldsymbol{\xi}, \boldsymbol{\theta}) = \bar{y}_0^\tau].$$

---

[11]While identification results for specific causal estimands are beyond the scope of this paper, system identification has already been studied for hybrid systems under the condition of "persistence of excitation" (Johnson, 2023; Saoud et al., 2024). Under such conditions, a posterior density $p(\boldsymbol{\xi}, \boldsymbol{\theta} \mid \boldsymbol{v}_0 = \mathcal{D})$, for example, where $\mathcal{D}$ is a realization of $\boldsymbol{v}_0$, is sufficiently well-behaved to estimate targets defined in this paper.

[12]Without loss of generality, we write that the data are subject to Gaussian observation noise. Many practical settings call for observation noise, but we also note that the deterministic relationship between inputs $(\boldsymbol{\xi}, \boldsymbol{\theta})$ and state trajectories means that inference behaves poorly without observation noise.

| query | outcome | probability |
|---|---|---|
| nec. | $Y = \mathbb{I}[b_{q_2}^\tau \leqslant \gamma]$ | $\Pr\left(Y_{x'} = 0 \mid X = 1, Y = 1\right) = \Pr\left(\left[b_{q_2}^\tau \leqslant \gamma\right] \mid b_{q_1}^\tau > \gamma\right)$ |
| suf. | $Y = \mathbb{I}[b_{q_2}^\tau > \gamma]$ | $\Pr\left(Y_{x'} = 1 \mid X = 0, Y = 0\right) = \Pr\left(\left[b_{q_2}^\tau > \gamma\right] \mid b_{q_1}^\tau \leqslant \gamma\right)$ |
| nec. and suf. | $Y = \mathbb{I}[b_{q_2}^\tau \leqslant \gamma]\mathbb{I}[b_{q_1}^\tau \geqslant \gamma]$ | $\Pr\left(Y_x = 1, Y_{x'} = 0\right) = \Pr\left(b_{q_1}^\tau > \gamma, b_{q_2}^\tau \leqslant \gamma\right)$ |

Table 1: Identities for the probabilities of causation in the fishery management example. Under TAC quota $q_i$, the biomass of the fished species at time $\tau$ is given by $b_{q_i}^\tau$. The outcome $Y$ is achieved when that biomass meets or exceeds $\gamma$. We rely on the standard exogeneity conditions $Y_x \perp\!\!\!\perp X$ and $Y_{x'} \perp\!\!\!\perp X$,[15] and the fact that, conditioned on $X = 1$ ($X_2$), $Y$ reduces to the outcome only in the world with allowable catch set to $q_1$ ($q_2$).

Many of the more complex causal inference tasks — such as mediation analysis, the estimation of population-level conditional average treatment effects, or even actual cause assessments — are constructed from the counterfactual building blocks we propose here. Indeed, once a counterfactual semantics is established, and a twin-world or potential-outcomes syntax (e.g., differentiating $y_0$ from $y_1$) is enumerated, many estimands are straightforward and familiar to develop. In the next section, we explore just such a class of estimand: the probabilities of causation.

# 6 Case Study: Necessary and Sufficient Causation

To illustrate a more sophisticated application of our interventional semantics, we map the standard definitions for the probabilities of necessary and sufficient causation (originally formalized by Pearl (1999)) onto dynamically triggered, discontinuous interventions in hybrid systems. In particular, we work in the fishery management domain where regulators employ Total Allowable Catch (TAC) policies to dynamically end the commercial fishing season after caught biomass reaches certain quotas. If interested, the reader may wish to review appendix G.1, in which we provide motivating historical context for this domain. Additionally, we review Pearl's original formulation of the probabilities of causation (PoC) in appendix G.2. Throughout appendix G, we provide full simulation analyses of the case study. Code is available here,[13] and relies on the dynamical systems package from the causal probabilistic programming language ChiRho (Basis-Research, 2025).[14]

We focus on a hypothetical fishery involving three trophic levels — apex predators, intermediate predators (the fished species), and forage fish — with dynamics captured by the differential equations presented by Zhou & Smith (2017). Throughout a single season, fishing pressure is modeled at a constant rate applied to the intermediate predator, plus some bycatch on the apex trophic level. Regulators intervene by ending the fishing season (setting the catch rate to zero) when the integrated catch reaches a predefined TAC quota. The goal of these policies is to ensure that the biomass of the target fishery species recovers to sustainable level $\gamma$ by the beginning of the next season.

In this context, stakeholders may debate the necessity and/or sufficiency of certain regulatory policies in maintaining joint ecological and economic goals for the fishery. The probabilities of causation are formal tools supporting the assessment of causal attribution between causes and their (supposed) effects. Pearl (1999) first formalized the PoC for *binary* treatments and outcomes — here, however, both the TAC quota and the biomass are scalar valued. We therefore follow Kawakami et al.'s (2024) generalization of the PoCs to support contrastive queries between scalar-valued treatments and their thresholded outcomes (see their Def. 3.1). Consider two TAC quotas $q_1$ and $q_2$, and the following natural language queries. In table 1, we provide the formalized estimands written in our notation.

- **necessity:** in worlds where the end-of-year biomass levels exceed the target level $\gamma$ (success) under quota $q_1$, what is the probability of failure had regulators used quota $q_2$ instead?
- **sufficiency:** in worlds where the end-of-year biomass levels remain below the target level $\gamma$ (failure) under $q_1$, what is the probability of success had regulators used $q_2$ instead?
- **necessity and sufficiency:** what is the probability that both (1) $q_1$ results in success and (2) $q_2$ results in failure?

---

[13] https://basisresearch.github.io/counterfactuals-for-hybrid-systems

[14] https://github.com/BasisResearch/chirho

[15] If some parameter or initial condition were influenced by a confounder that also influenced $X$, this would not be the case, and conditioning on $X$ would be required in the identities listed in table 1.

For readers less familiar with the applications of the PoC to decision and policy making, we provide an expanded narrative scaffolding for this example in appendix G.4. In appendix G.5, we provide an additional example designed to highlight how certain natural language ambiguities in causal attribution queries — particularly those involving multi-faceted, real world events and policies — can be formally clarified.

The PoC queries above rely on the construction of twin, contrastive worlds — one with TAC quota $q_1$, and the other with $q_2$. To model these worlds, we start with a system $\mathcal{P}$ characterizing year-round fishing pressure (i.e., no regulatory intervention), and then transform its constraints to add a dynamic, season-ending intervention. Notationally, let $h_i$ represent the harvest rate, and $b_i$ the biomass, at trophic level $i$, and let $z$ be the total catch (integral of $\dot{z} = h_2 b_2$) at the intermediate trophic level. The system state can be conceptualized as $[z, h_{1:3}, b_{1:3}] = \boldsymbol{x} \in \mathcal{S} = \mathbb{R}_{\geqslant}^7$.

The regulatory, season-ending intervention can be modeled by dynamically setting harvest rates to zero when the catch exceeds a threshold $q_i$ (with $i \in \{1, 2\}$). By using our interventional semantics, we can construct parallel worlds with the same random initial conditions and parameters. See appendix G.3 for a generalization of this model to multi-season time scales.

$$\tilde{D}_{q_i}(\boldsymbol{\theta}) = \{z \in \mathbb{R}_{\geqslant 0} \mid z \geqslant q_i\} \times \mathbb{R}_{\geqslant 0}^6; \qquad \tilde{G}_{q_i}(\boldsymbol{x}, \boldsymbol{\theta}) = \{[z, 0, 0, 0, b_{1:3}]\}; \qquad (5)$$

$$\mathcal{R}_s = (\mathcal{P}, \boldsymbol{\xi}, \boldsymbol{\theta}); \ \ \mathcal{R}_{q_1} = (\mathcal{P}_{q_1}, \boldsymbol{\xi}, \boldsymbol{\theta}); \ \ \mathcal{R}_{q_2} = (\mathcal{P}_{q_2}, \boldsymbol{\xi}, \boldsymbol{\theta}); \ \ \mathcal{P}_{q_i} = \texttt{instint}(\mathcal{P}, \tilde{D}_{q_i}, \tilde{G}_{q_i}). \quad (6)$$

## 7 Limitations and Future Work

Most research developing causal inference tools starts by casting a problem in the format of structural causal models (SCMs) (Pearl, 2009). Our work differs in that we construct our counterfactual semantics directly in the parlance of hybrid systems. These two tacks are compatible, however. For example, with our measurability results in hand, the time-parameterized solution map $\varphi$ can be treated as a structural equation with initial conditions $\boldsymbol{\xi}$ and parameters $\boldsymbol{\theta}$ viewed as parent variables in a larger SCM. Our interventional semantics, then, exposes the causal dynamics of the hybrid system for manipulation. When $\varphi$ is interpreted as a structural equation, our semantics could be viewed as characterizing a family of "soft-interventions" (Correa & Bareinboim, 2020) on the solution map. Importantly, the adjoint method (Chen et al., 2018) can be used in tandem with auto-differentiation machinery to learn "event function" (i.e. jump set) parameters,[16] thereby supporting end-to-end differentiation of composite SCM and hybrid system models. Relatedly, equivalent forward-Euler representations may prove useful in actual cause analysis of hybrid systems (Halpern & Peters, 2022).

This leaves a few limitations to review. First, we do not present non-parametric, estimand-specific identification results — indeed, there may exist sufficient conditions for estimand identification that are weaker than those established for full system identification. Second, as discussed following definition 1, we focus only on finite time regimes, leaving analysis of hybrid equilibria to future work. Furthermore, we do not provide conditions under which intervention preserves non-Zeno behavior.

## 8 Conclusion

This paper has strengthened the connection between the modeling capabilities offered by hybrid systems theory and the causal reasoning capabilities developed by the causal inference research community.

We characterize and demonstrate a counterfactual semantics for a class of dynamically triggered, instantaneous interventions that underpin many closed-loop control problems. Bypassing an explicit re-casting of hybrid systems as structural causal models, we use hybrid systems as the primary modeling substrate. This allows clear connections to the extensive body of work on hybrid systems theory, in which we can derive and characterize mild conditions under which solution existence, uniqueness and measurability are preserved in the intervened system.

Finally, we illustrate the flexibility and power of the resulting framework by first formalizing common causal estimands for hybrid systems, and then by developing a case study using the three probabilities of causation in the context of fishery management.

---

[16]See `https://github.com/rtqichen/torchdiffeq/blob/master/examples/bouncing_ball.py` for an example.

## Acknowledgments and Disclosure of Funding

AZ, DB, RU, JZ, and SW were supported on DARPA Automating Scientific Knowledge Extraction and Modeling (ASKEM) program Grant HR00112220036. We thank Anirban Chaudhuri, Sabina Altus, Joseph Cottam, and Neeraj Kumar for their insights and contributions throughout the ASKEM program; our colleagues at Basis for helpful comments and discussions; Eli Bingham for guiding our thinking and software design choices around these ideas; Paul Wintz for answering many questions about hybrid systems and pointing us to relevant literature; and David Jensen for helpful comments and discussion. Pacific Northwest National Laboratory (PNNL) is a multiprogram national laboratory operated by Battelle for the DOE under Contract DEAC05-76RLO 1830. The views, opinions and/or findings expressed are those of the author and should not be interpreted as representing the official views or policies of the Department of Defense or the U.S. Government. Distribution Statement "A" (Approved for Public Release, Distribution Unlimited).

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

## A  Supplementary Definitions and Standard Assumptions

### A.1  Differential Inclusions and Set-Valued Maps

We follow Goebel et al. (2012) in generalizing to hybrid systems with *inclusion* constraints. A differential inclusion $F : \mathcal{S} \rightrightarrows \mathbb{R}^n$, for example, specifies the constraint that the time derivative $\dot{x}$ of the state must be included in the set $F(x) \subseteq \mathbb{R}^n$. Note that the equality constraint $\dot{x} = f(x)$ for some $f : \mathcal{S} \to \mathbb{R}^n$ is a special case of the broader notion of differential inclusion. To clarify, the stacked double arrows in, for example, $\mathcal{S} \rightrightarrows \mathbb{R}^n$ indicate a *set-valued* mapping from $\mathcal{S}$ to a *subset* of $\mathbb{R}^n$. Goebel et al. (2012) define the domain of a set-valued mapping $V : \mathcal{X} \rightrightarrows \mathcal{Y}$ as dom $V = \{x \in \mathcal{X} : V(x) \neq \varnothing\}$. The graph of $V$ is then

$$\mathcal{G}(V) = \{(x, y) \in \mathcal{X} \times \mathcal{Y} : x \in \text{dom } V, \ y \in V(x)\}. \tag{7}$$

### A.2  Ordered Set-Valued Maps

Ordered set-valued maps are special cases of set-valued maps, which we use in this paper to keep track of interventions.

**Definition 6** (Ordered Set-Valued Map)**.** Let $G = (G_1, \ldots, G_K)$ be a finite sequence of set-valued maps. We call $G$ an *ordered set-valued map*, which means it is equipped with the following operation:

$$G^\dagger = x \mapsto \bigcup_{k=1}^K G_k(x); \quad \texttt{last}(G) = G_K. \tag{8}$$

Therefore, dom $G^\dagger = \bigcup_{k=1}^K \text{dom } G_k$. Given two sequences $G = (G_1, \ldots, G_K)$ and $H = (H_1, \ldots, H_L)$, we denote $G \sqcup H = (G_1, \ldots, G_K, H_1, \ldots, H_L)$. By slight abuse of notation, we sometimes identify a map $G$ with the corresponding one-element sequence $(G)$, and also use $G$ in place of $G^\dagger$ when the context requires a "vanilla" set-valued map.

### A.3  Solution Concept

The following definitions and propositions are given almost exactly as stated by Goebel et al. (2012), except that we adapt them slightly for explicitly parameterized hybrid systems (definition 1).

The nature of hybrid systems implies that their solutions should be functions of both continuous time $t \in \mathbb{R}_{\geq 0}$ and discrete time $j \in \mathbb{N}$. Let $t_j$ denote the time of the $j$-th discrete event, with $t_j \leq t_{j+1}$ for all $j \in \mathbb{N}$ and $t_0 = 0$. Following (Goebel et al., 2012, Sects. 2.2–2.3), we define, for each possible parameterization $\boldsymbol{\theta} \in \Theta$ and initial condition $\boldsymbol{\xi} \in \mathcal{S}$, a "solution" to $\mathcal{H}(\boldsymbol{\theta})$ to be a "hybrid arc", which is formally a set-valued map $\phi(\cdot; \boldsymbol{\xi}, \boldsymbol{\theta}) : \mathbb{R}_{\geq 0} \times \mathbb{N} \rightrightarrows \mathbb{R}^n$. We can formalize this time-event space (of which dom $\phi$ is an example) as follows:

**Definition 7** (Hybrid Time Domain from Goebel et al. (2012) (Def. 2.3))**.** $E \subset \mathbb{R}_{\geq 0} \times \mathbb{N}$ is a compact hybrid time domain if it is a finite union of sequence of closed intervals $E = \bigcup_{j=0}^{J-1} ([t_j, t_{j+1}] \times \{j\})$, where $0 = t_0 \leq t_1 \leq \cdots t_J$, and $E$ is a hybrid time domain if for each $(T, J) \in E$, the set $E \cap ([0, T] \times \{0, 1, \ldots, J\})$ is a compact hybrid time domain.

Generally, dom $\phi$ is unknown until after a particular solution $\phi$ is found, as it depends on the exact sequence of state-dependent jump times; therefore, it is natural to consider $\phi$ as a-priori set-valued.

**Definition 8** (Solution Concept adapted from Goebel et al. (2012) (Def 2.6))**.** Consider parameterized hybrid system $\mathcal{P} = (\mathcal{H}, \mathcal{S}, \Theta)$, with $\mathcal{H} = (C, F, D, G)$. For $\boldsymbol{\theta} \in \Theta$, any solution $\phi(\cdot; \boldsymbol{\xi}, \boldsymbol{\theta})$ to $\mathcal{H}(\boldsymbol{\theta})$ must satisfy $\phi(0, 0; \boldsymbol{\xi}, \boldsymbol{\theta}) = \boldsymbol{\xi} \in \overline{C(\boldsymbol{\theta})} \cup D(\boldsymbol{\theta})$ and the constraints implied by $\mathcal{H}(\boldsymbol{\theta})$, i.e.:

1. for all $j \in \mathbb{N}$ such that $I^j := \{t : (t, j) \in \text{dom } \phi\}$ has nonempty interior, we have $\phi(t, j) \in C(\boldsymbol{\theta}), \forall t \in \text{int } I^j$, and $\dot{\phi}(t, j) \in F_{\boldsymbol{\theta}}(\phi(t, j))$, for almost all $t \in I^j$; [continuous flow regime]

2. for all $(t, j) \in \text{dom } \phi$ s.t. $(t, j+1) \in \text{dom } \phi$, we have $\phi(t, j; \boldsymbol{\xi}, \boldsymbol{\theta}) \in D(\boldsymbol{\theta})$ and $\phi(t, j + 1; \boldsymbol{\xi}, \boldsymbol{\theta}) \in G_{\boldsymbol{\theta}}(\phi(t, j; \boldsymbol{\xi}, \boldsymbol{\theta}))$ [discrete jump regime] .

It is convenient to work with solutions that cannot be extended, as formalized by the following concept.

**Definition 9** (Maximal Solutions adapted from Goebel et al. (2012) (Def 2.7)). A solution $\phi(\cdot; \boldsymbol{\xi}, \boldsymbol{\theta})$ to $\mathcal{H}(\boldsymbol{\theta})$ (as in definition 8) is *maximal* if there does not exist another solution $\phi'(\cdot; \boldsymbol{\xi}, \boldsymbol{\theta})$ to $\mathcal{H}(\boldsymbol{\theta})$ such that dom $\phi(\cdot; \boldsymbol{\xi}, \boldsymbol{\theta})$ is a proper subset of dom $\phi'(\cdot; \boldsymbol{\xi}, \boldsymbol{\theta})$ and $\phi(t, j; \boldsymbol{\xi}, \boldsymbol{\theta}) = \phi'(t, j; \boldsymbol{\xi}, \boldsymbol{\theta})$ for all $(t, j) \in$ dom $\phi(\cdot; \boldsymbol{\xi}, \boldsymbol{\theta})$.

Unless specified otherwise, we always consider maximal solutions in this paper. With the solution concept established, we can now state conditions for the existence and uniqueness of solutions. We again borrow from Goebel et al. (2012), and adapt accordingly to support parameterized systems (definition 1).

**Proposition 1** (Basic Existence adapted from Goebel et al. (2012) (Proposition 2.10)). *Consider parameterized hybrid system $\mathcal{P} = (\mathcal{H}, \mathcal{S}, \Theta) = (C, F, D, G, \mathcal{S}, \Theta)$, and a standard hybrid system $\mathcal{H}(\boldsymbol{\theta}) = (C(\boldsymbol{\theta}), F_{\boldsymbol{\theta}}, D(\boldsymbol{\theta}), G_{\boldsymbol{\theta}})$ for some $\boldsymbol{\theta} \in \Theta$. Let $\boldsymbol{\xi} \in \overline{C(\boldsymbol{\theta})} \cup D(\boldsymbol{\theta})$. If $\boldsymbol{\xi} \in D(\boldsymbol{\theta})$ or*

*(VC)    there exists $\epsilon > 0$ and an absolutely continuous function $z : [0, \epsilon] \to \mathbb{R}^n$ such that $z(0) = \boldsymbol{\xi}$, $\dot{z}(t) \in F_{\boldsymbol{\theta}}(z(t))$ for almost all $t \in [0, \epsilon]$ and $z(t) \in C(\boldsymbol{\theta})$ for all $t \in (0, \epsilon]$,*

*then there exists a non-trivial solution $\phi(\cdot; \boldsymbol{\xi}, \boldsymbol{\theta})$ to $\mathcal{H}$ with $\phi(0, 0; \boldsymbol{\xi}, \boldsymbol{\theta}) = \boldsymbol{\xi}$.[17] If (VC) holds for every $\boldsymbol{\xi} \in \overline{C(\boldsymbol{\theta})} \cup D(\boldsymbol{\theta})$, then there exists a nontrivial solution to $\mathcal{H}(\boldsymbol{\theta})$ from every point of $\overline{C(\boldsymbol{\theta})} \cup D(\boldsymbol{\theta})$. If the foregoing further holds for $\mathcal{H}(\boldsymbol{\theta})$ at every $\boldsymbol{\theta} \in \Theta$, we say $\mathcal{P}$ fulfills the conditions for basic existence.*

**Proposition 2** (Basic Uniqueness adapted from Goebel et al. (2012) (Proposition 2.11)). *Consider parameterized hybrid system $\mathcal{P} = (\mathcal{H}, \mathcal{S}, \Theta) = (C, F, D, G, \mathcal{S}, \Theta)$, and a standard hybrid system $\mathcal{H}(\boldsymbol{\theta}) = (C(\boldsymbol{\theta}), F_{\boldsymbol{\theta}}, D(\boldsymbol{\theta}), G_{\boldsymbol{\theta}})$ for some $\boldsymbol{\theta} \in \Theta$. For every $\boldsymbol{\xi} \in \overline{C(\boldsymbol{\theta})} \cup D(\boldsymbol{\theta})$ there exists a unique maximal solution $\phi(\cdot; \boldsymbol{\xi}, \boldsymbol{\theta})$ with $\phi(0, 0; \boldsymbol{\xi}, \boldsymbol{\theta}) = \boldsymbol{\xi}$ provided that the following conditions hold.*

*(a) For every $\boldsymbol{\xi} \in \overline{C(\boldsymbol{\theta})} \backslash D(\boldsymbol{\theta})$, $T > 0$, if two absolutely continuous $z_1, z_2 : [0, T] \to \mathcal{S}$ are such that $\dot{z}_i(t) \in F_{\boldsymbol{\theta}}(z_i(t))$ for almost all $t \in [0, T]$, $z_i(t) \in C(\boldsymbol{\theta})$ for all $t \in (0, T]$, and $z_i(0) = \boldsymbol{\xi}$, $i = 1, 2$, then $z_1(t) = z_2(t)$ for all $t \in [0, T]$;*

*(b) for every $\boldsymbol{\xi} \in \overline{C(\boldsymbol{\theta})} \cap D(\boldsymbol{\theta})$, there does not exist $\epsilon > 0$ and an absolutely continuous function $z : [0, \epsilon] \to \mathcal{S}$ such that $z(0) = \boldsymbol{\xi}$, $\dot{z}(t) \in F_{\boldsymbol{\theta}}(z(t))$ for almost all $t \in [0, \epsilon]$ and $z(t) \in C(\boldsymbol{\theta})$ for all $t \in (0, \epsilon]$;*

*(c) for every $\boldsymbol{\xi} \in D(\boldsymbol{\theta})$, $G_{\boldsymbol{\theta}}(\boldsymbol{\xi})$ consists of one point.*

*If the foregoing further holds at every $\boldsymbol{\theta} \in \Theta$, we say that $\mathcal{P}$ fulfills the conditions for basic uniqueness.*

### A.4    Finite-Time Measurability of Solution in Initial Conditions and Parameters

Measurability is key to coherently defining causal estimands as (conditional) expectations. In particular, we use the measurability of a time-parameterized "solution map" jointly in the initial state and parameters. By "solution map", we refer either to functions $\phi$ or $\varphi$ that, when provided some $\boldsymbol{\xi} \in \mathcal{S}$ and $\boldsymbol{\theta} \in \Theta$, yield hybrid arc $(t, j) \mapsto \phi(t, j; \boldsymbol{\xi}, \boldsymbol{\theta})$ and time-parameterized function $t \mapsto \varphi(t; \boldsymbol{\xi}, \boldsymbol{\theta})$ respectively.

As stated following definition 1, in this paper, we focus strictly on finite time horizons. Definition 10, below, makes this finite-time limitation precise, and then employs that definition to formalize the time-parameterized solution map and its measurability.

**Definition 10** ($t^+$ Uniquely Evaluable). Consider parameterized hybrid system $\mathcal{P} = (C, F, D, G, \mathcal{S}, \Theta)$ that fulfills conditions for basic existence and uniqueness (propositions 1 and 2). Define $t^+ = \min_{\boldsymbol{\xi}, \boldsymbol{\theta}} \sup_t$ dom $\phi(\cdot; \boldsymbol{\xi}, \boldsymbol{\theta})$, meaning that for every $\boldsymbol{\xi} \in \mathcal{S}, \boldsymbol{\theta} \in \Theta$ yielding unique solution $t, j \mapsto \phi(t, j; \boldsymbol{\xi}, \boldsymbol{\theta})$, $\forall t \in [0, t^+)$ there exists $j \in \mathbb{N}$ such that $(t, j) \in$ dom $\phi(\cdot, \cdot; \boldsymbol{\xi}, \boldsymbol{\theta})$. We then say that $\mathcal{P}$ is $t^+$ *uniquely evaluable*.[18]

---

[17]A non-trivial solution is one with more than a single point in its domain (Goebel et al., 2012, Def 2.5)

[18]Unless the space of initial conditions and parameters are limited to exclude reachable states arbitrarily close to Zeno points, Zeno systems will always have arbitrarily small $t^+$. Additionally, this definition implicitly excludes evaluation times at the end (in continuous time) of eventually discrete solutions. If such solutions are not complete, one might wish to include the final time in the evaluable interval.

**Definition 11** (Time Parameterized Solution Map). Consider $t^+$ uniquely evaluable parameterized hybrid system $\mathcal{P} = (C, F, D, G, \mathcal{S}, \Theta)$ and its solution map $\phi$. Define for all $t \in [0, t^+)$, $\boldsymbol{\xi} \in \mathcal{S}$, $\boldsymbol{\theta} \in \Theta$ its *time-parameterized solution*

$$\varphi(t; \boldsymbol{\xi}, \boldsymbol{\theta}) = \phi\left(t, j_t^+(\boldsymbol{\xi}, \boldsymbol{\theta}); \boldsymbol{\xi}, \boldsymbol{\theta}\right) \tag{9}$$

where $j_t^+(\boldsymbol{\xi}, \boldsymbol{\theta})$ is the index of the last discrete jump at time $t$

$$j_t^+(\boldsymbol{\xi}, \boldsymbol{\theta}) = \max\{j : (t, j) \in \operatorname{dom} \phi(\cdot; \boldsymbol{\xi}, \boldsymbol{\theta})\} \tag{10}$$

**Definition 12** ($t^+$ Measurable). Consider $t^+$ uniquely evaluable parameterized hybrid system $\mathcal{P}$ and its time-parameterized solution map $\varphi$. If, for every fixed $t \in [0, t^+)$, $\boldsymbol{\xi}, \boldsymbol{\theta} \mapsto \varphi(t; \boldsymbol{\xi}, \boldsymbol{\theta})$ is a Borel-measurable function, we say that $\mathcal{P}$ has a $t^+$ *measurable* time-parameterized solution map $\varphi$.

### A.5 Flow Preferring Subtraction and Lowering

**Definition 13** (Flow-Preferring Subtraction). Consider parameterized hybrid system $\mathcal{P} = (C, F, D, G, \mathcal{S}, \Theta)$ that meets the hybrid basic conditions (assumption 4). We borrow the following viability condition from proposition 1 on a point $\boldsymbol{\xi} \in \mathcal{S}$, for some $\boldsymbol{\theta} \in \Theta$.

(VC)  there exists $\epsilon > 0$ and an absolutely continuous function $z : [0, \epsilon] \to \mathbb{R}^n$ such that $z(0) = \boldsymbol{\xi}$, $\dot{z}(t) \in F(z(t), \boldsymbol{\theta})$ for almost all $t \in [0, \epsilon]$ and $z(t) \in C(\boldsymbol{\theta})$ for all $t \in (0, \epsilon]$,

We can then transform $D$ to be flow preferring by writing

$$\texttt{preferflow}(D, C, F) = \boldsymbol{\theta} \mapsto D(\boldsymbol{\theta}) \backslash \{\boldsymbol{\xi} \in \mathcal{S} : (\text{VC}) \text{ holds for } \boldsymbol{\theta} \text{ from } \boldsymbol{\xi}\} \tag{11}$$

Recall the definition of ordered set-values maps (definition 6) affording the $\texttt{last}(G)$ operation on $G$, the jump map.

**Definition 14** (Lowering). Consider parameterized hybrid system $\mathcal{P} = (C, F, D, G, \mathcal{S}, \Theta)$ that meets the hybrid basic conditions (assumption 4). We write that

$$D' = \texttt{preferflow}(D, C, F) \tag{12}$$

$$G' = \texttt{last}(G), \tag{13}$$

$$\texttt{lower}(\mathcal{P}) = (C, F, D', G', \mathcal{S}, \Theta) \tag{14}$$

### A.6 Collected Assumptions on the Hybrid System

**Assumption 3** (Unique, Complete, and Borel Solution Exists for Differential Inclusion for all $\mathcal{S}$). Consider parameterized hybrid system $\mathcal{P} = (C, F, D, G, \mathcal{S}, \Theta)$. Assume that

(F1)  for every $\boldsymbol{\xi} \in \mathcal{S}$, $\boldsymbol{\theta} \in \Theta$, $T > 0$, if two absolutely continuous $z_1, z_2 : [0, T] \to \mathcal{S}$ are such that $\dot{z}_i(t) \in F_{\boldsymbol{\theta}}(z_i(t))$ for almost all $t \in [0, T]$, $z_i(t) \in \mathcal{S}$ for all $t \in (0, T]$, and $z_i(0) = \boldsymbol{\xi}$, $i = 1, 2$, then $z_1(t) = z_2(t)$ for all $t \in [0, T]$;

(F2)  for all $\boldsymbol{\xi} \in \mathcal{S}$ and $\boldsymbol{\theta} \in \Theta$, such a $z_1$ exists for every $T \in (0, \infty)$;

(F3)  with $z(t; \boldsymbol{\xi}, \boldsymbol{\theta}) = z_1(t)$ for all $\boldsymbol{\xi} \in \mathcal{S}$, $\boldsymbol{\theta} \in \Theta$, and $t \in [0, \infty)$, $\boldsymbol{\xi}, \boldsymbol{\theta} \mapsto z(t; \boldsymbol{\xi}, \boldsymbol{\theta})$ is a Borel-measurable function for every $t \in [0, \infty)$.

Importantly, note that assumption 3 only relates to the differential inclusion, and does not preclude $\mathcal{P}$ from jumping, or from pathologies associated with jumps. Additionally, observe that $z(t; \boldsymbol{\xi}, \boldsymbol{\theta}) \equiv \phi(t, 0; \boldsymbol{\xi}, \boldsymbol{\theta})$— that is, statements on $z$ trivially apply to the solution mapping up to and including the time of the first jump.

**Assumption 4** (Hybrid Basic Conditions adapted from Goebel et al. (2012) (Assump. 6.5)). Consider parameterized hybrid system $\mathcal{P} = (C, F, D, G, \mathcal{S}, \Theta)$, and assume for all $\boldsymbol{\theta} \in \Theta$ that the following hold.

(A1)  $C(\boldsymbol{\theta})$ and $D(\boldsymbol{\theta})$ are closed subsets of $\mathcal{S}$;

(A2)  $F_{\boldsymbol{\theta}} : \mathcal{S} \rightrightarrows \mathbb{R}^n$ is outer semi-continuous and locally bounded relative to $C(\boldsymbol{\theta})$, $C(\boldsymbol{\theta}) \subset \operatorname{dom} F_{\boldsymbol{\theta}}$, and $F(\boldsymbol{x}, \boldsymbol{\theta})$ is convex for every $\boldsymbol{x} \in C(\boldsymbol{\theta})$;

(A3)  $G_{\boldsymbol{\theta}} : \mathcal{S} \rightrightarrows \mathcal{S}$ is outer semi-continuous and locally bounded relative to $D(\boldsymbol{\theta})$, and $D(\boldsymbol{\theta}) \subset \operatorname{dom} G_{\boldsymbol{\theta}}$.

In particular, (A1) implies that $D(\boldsymbol{\theta})$ and $C(\boldsymbol{\theta})$ must overlap on any shared boundary — solutions that start at or graze this boundary can, non-uniquely, either jump or flow. Additionally, the outer semi-continuity of $G_{\boldsymbol{\theta}}$ (A3) requires that, at the boundaries of the pieces in a piecewise $G_{\boldsymbol{\theta}}$, $G_{\boldsymbol{\theta}}$ must return values from multiple pieces. Solutions hitting those boundaries can jump to multiple states.

**Assumption 5** (Collected Assumptions on the Original System). The parameterized hybrid system $\mathcal{P}$ can be constructed as $\mathcal{P} = \texttt{lower}\,(\mathcal{P}_{\uparrow})$, where $\mathcal{P}_{\uparrow} = (C, F, D, G, \mathcal{S}, \Theta)$, such that:

(P1)   $\mathcal{P}_{\uparrow}$ satisfies assumption 4;

(P2)   $\mathcal{P}_{\uparrow}$ fulfills the conditions for basic existence (proposition 1);

(P3)   $\mathcal{P}_{\uparrow}$ has a unique solution to its differential inclusion $F$ from everywhere in $\mathcal{S}$ and $\Theta$ (assumption 3);

(P4)   $C(\boldsymbol{\theta})$ is outer semi-continuous at every $\boldsymbol{\theta} \in \Theta$;

(P5)   the graph $\mathcal{G}(D)$ of the jump set mapping $D$ is Borel;

(P6)   $\texttt{last}(G)$ is single-valued on its domain, with $\texttt{last}(G)(\boldsymbol{x}, \boldsymbol{\theta}) = \{g(\boldsymbol{x}, \boldsymbol{\theta})\}$, and $g$ Borel-measurable for all $\boldsymbol{x}, \boldsymbol{\theta} \in \text{dom}\,\texttt{last}(G)$.

## A.7   Well-Behaved Jump Set

**Definition 15** (Well-Behaved Set). Consider $\Theta \subseteq \mathbb{R}^m$, $\mathcal{S} \subseteq \mathbb{R}^n$, arbitrary set-valued mapping $A : \Theta \rightrightarrows \mathcal{S}$, and differential inclusion $F : \mathcal{S} \times \Theta \rightrightarrows \mathbb{R}^n$. Suppose that for every $\boldsymbol{\theta} \in \Theta$ and $\boldsymbol{\xi} \in \mathcal{S}$ where

(VC$_{\mathcal{S}}$)   there exists $\epsilon > 0$ and an absolutely continuous function $z : [0, \epsilon] \to \mathbb{R}^n$ such that $z(0) = \boldsymbol{\xi}$, $\dot{z}(t) \in F(z(t), \boldsymbol{\theta})$ for almost all $t \in [0, \epsilon]$ and $z(t) \in \mathcal{S}$ for all $t \in (0, \epsilon]$,

there also exists some $\epsilon' \in (0, \epsilon]$ such that

$$z((0, \epsilon']) \subseteq \text{int}\, A(\boldsymbol{\theta}) \text{ or } z((0, \epsilon']) \subseteq \mathcal{S}\backslash\text{int}\, A(\boldsymbol{\theta}) \tag{15}$$

In such a case, we say that $A$ is *well-behaved* relative to $\mathcal{S}$ for $\Theta$ and $F$. For a parameterized hybrid system $\mathcal{P} = (C, F, D, G, \mathcal{S}, \Theta)$, we sometimes say that $A$ is well-behaved relative to $\mathcal{P}$.

**Assumption 6** (Well-Behaved Interventional Subset). Consider set-valued mapping $\tilde{D} : \Theta \rightrightarrows \mathcal{S}$ and parameterized hybrid system $\mathcal{P}$. Assume $\tilde{D}$ is well-behaved relative to $\mathcal{P}$ (definition 15).

**Observation 1** (Flow into Subdivisions of $C$ by $\tilde{D}$). *Consider set-valued mapping $\tilde{D} : \Theta \rightrightarrows \mathcal{S}$ that meets assumption 6 relative to some parameterized hybrid system $\mathcal{P} = (C, F, D, G, \mathcal{S}, \Theta)$. It is then the case that, for every $\boldsymbol{\theta} \in \Theta$ and $\boldsymbol{\xi} \in \mathcal{S}$ where*

*(VC)   there exists $\epsilon > 0$ and an absolutely continuous function $z : [0, \epsilon] \to \mathbb{R}^n$ such that $z(0) = \boldsymbol{\xi}$, $\dot{z}(t) \in F(z(t), \boldsymbol{\theta})$ for almost all $t \in [0, \epsilon]$ and $z(t) \in C(\boldsymbol{\theta})$ for all $t \in (0, \epsilon]$,*

*there also exists some $\epsilon' \in (0, \epsilon]$ such that*

$$z((0, \epsilon']) \subseteq \text{int}\, \tilde{D}(\boldsymbol{\theta}) \text{ or } z((0, \epsilon']) \subseteq C(\boldsymbol{\theta})\backslash\text{int}\, \tilde{D}(\boldsymbol{\theta}). \tag{16}$$

*Proof.* Suppose the proposed antecedent and note that a trajectory $z((0, \epsilon]) \subseteq C(\boldsymbol{\theta}) \subseteq \mathcal{S}$ fulfills the antecedent of the assumed well-behaved property of $\tilde{D}$ relative to $\mathcal{S}$ (assumption 6). This implies that there exists $\epsilon' \in (0, \epsilon]$ such that either $z((0, \epsilon']) \subseteq \text{int}\, \tilde{D}(\boldsymbol{\theta})$ or $z((0, \epsilon']) \subseteq \mathcal{S}\backslash\text{int}\, \tilde{D}(\boldsymbol{\theta})$. $z((0, \epsilon']) \subseteq \text{int}\, \tilde{D}(\boldsymbol{\theta})$ is precisely the first case of our desired consequent. Thus, we need only show that $z((0, \epsilon']) \subseteq \mathcal{S}\backslash\text{int}\, \tilde{D}(\boldsymbol{\theta})$ and $z((0, \epsilon]) \subseteq C(\boldsymbol{\theta})$ imply $z((0, \epsilon']) \subseteq C(\boldsymbol{\theta})\backslash\text{int}\, \tilde{D}(\boldsymbol{\theta})$. We have $z((0, \epsilon']) \subseteq z((0, \epsilon]) \subseteq C(\boldsymbol{\theta})$, and can thus take the intersection to see this implies the second case of the desired consequent:

$$z((0, \epsilon']) \subseteq C(\boldsymbol{\theta}) \cap \left( \mathcal{S}\backslash\text{int}\, \tilde{D}(\boldsymbol{\theta}) \right) = C(\boldsymbol{\theta})\backslash\text{int}\, \tilde{D}(\boldsymbol{\theta}). \qquad \square$$

**Remark 1** (Universality of Assumption 6). If and only if assumption 6 holds relative to some parameterized hybrid system $\mathcal{P}$, then it also holds relative to $\texttt{instint}(\mathcal{P})$, $\texttt{instint}_{\uparrow}(\mathcal{P})$, and $\texttt{lower}(\mathcal{P})$.

*Proof.* Assumption 6 holding relative to $\mathcal{P} = (C, F, D, G, \mathcal{S}, \Theta)$ pertains only to $F, \mathcal{S}, \Theta$, which are unaffected by $\texttt{instint}$, $\texttt{instint}_{\uparrow}$, and $\texttt{lower}$. $\qquad \square$

## B  Space Augmentation

It is often useful to parameterize interventions, and a fully expressive interventional semantics benefits from *stateful* jump maps/sets. Thus, it will be useful to establish a primitive transformation that simply augments the parameter and state spaces, without changing the component functions of the system. Subsequent transformations can then operate on this augmented system. Note that, in eq. (22), we write the transformed jump map in its expanded form as an ordered set-valued map (definition 6).

**Definition 16.** (Space Augmentation) Consider $\tilde{\mathcal{S}} \subseteq \mathbb{R}^{\tilde{n}}$, and $\tilde{\Theta} \subseteq \mathbb{R}^{\tilde{m}}$. For any parameterized hybrid system $\mathcal{P} = (C, F, D, G, \mathcal{S}, \Theta)$ with $G = (G_1, \ldots, G_L)$, let, for all $\boldsymbol{x} \in \mathcal{S}$, $\tilde{\boldsymbol{x}} \in \tilde{\mathcal{S}}$, $\boldsymbol{\theta} \in \Theta$, $\tilde{\boldsymbol{\theta}} \in \tilde{\Theta}$,

$$\boldsymbol{x}' = [\boldsymbol{x}, \tilde{\boldsymbol{x}}], \quad \boldsymbol{\theta}' = \left[\boldsymbol{\theta}, \tilde{\boldsymbol{\theta}}\right] \tag{17}$$

$$C'\left(\boldsymbol{\theta}'\right) = C(\boldsymbol{\theta}) \times \tilde{\mathcal{S}} \tag{18}$$

$$F'\left(\boldsymbol{x}', \boldsymbol{\theta}'\right) = F\left(\boldsymbol{x}, \boldsymbol{\theta}\right) \times \{\boldsymbol{0}\} \tag{19}$$

$$D'\left(\boldsymbol{\theta}'\right) = D(\boldsymbol{\theta}) \times \tilde{\mathcal{S}} \tag{20}$$

$$G'_\ell\left(\boldsymbol{x}, \boldsymbol{\theta}\right) = G_\ell\left(\boldsymbol{x}, \boldsymbol{\theta}\right) \times \{\tilde{\boldsymbol{x}}\} \tag{21}$$

$$G'\left(\boldsymbol{x}', \boldsymbol{\theta}'\right) = \left(G'_1\left(\boldsymbol{x}, \boldsymbol{\theta}\right), \ldots, G'_L\left(\boldsymbol{x}, \boldsymbol{\theta}\right)\right) \tag{22}$$

$$\mathcal{S}' = \mathcal{S} \times \tilde{\mathcal{S}}, \quad \Theta' = \Theta \times \tilde{\Theta}, \tag{23}$$

then

$$\texttt{spaug}\left(\mathcal{P}, \tilde{\mathcal{S}}, \tilde{\Theta}\right) = \left(\left(C', F', D', G'\right), \mathcal{S}', \Theta'\right). \tag{24}$$

**Observation 2** (Compositions of Space Augmentation Preserves Key Properties)**.** *Consider parameterized hybrid system $\mathcal{P}$ that meets assumption 5, and any finite sequence $(\tilde{\mathcal{S}}_k, \tilde{\Theta}_k)$ of length $K$ such that $\tilde{\mathcal{S}}_k \subseteq \mathbb{R}^{\tilde{n}_k}$ and $\tilde{\Theta}_k \subseteq \mathbb{R}^{\tilde{m}_k}$. Let $\texttt{spaug}_k = \texttt{spaug}(\cdot, \tilde{\mathcal{S}}_k, \tilde{\Theta}_k)$ (definition 16) and*

$$\mathcal{P}' = \left(\texttt{spaug}_1 \circ \cdots \circ \texttt{spaug}_k \circ \cdots \circ \texttt{spaug}_K\right)(\mathcal{P}). \tag{25}$$

*$\mathcal{P}'$ then meets assumption 5 and has a unique solution (propositions 1 and 2) with a $t^+$ measurable (definition 12) time-parameterized solution map $\varphi$ (definition 11).*

*Proof.* The proposition follows from induction, $K < \infty$, and the fact that the space augmentation operation fulfills the same pattern described in fig. 2 for `instint`. That is, `spaug` commutes with `lower`, and it preserves (P1-6) (assumption 5) on an upstream system $\mathcal{P}'_\uparrow = \texttt{spaug}(\mathcal{P}_\uparrow, \ldots)$. For commutativity, recall that lowering makes a flow-preferring subtraction from the jump set (definition 13), and chooses the last map in the ordered jump map. A flow-preferring subtraction on $D'(\boldsymbol{\theta}) = D(\boldsymbol{\theta}) \times \tilde{\mathcal{S}}$ is dictated entirely by the behavior of $F(\boldsymbol{\theta})$ on $C(\boldsymbol{\theta})$ — i.e. $\texttt{preferflow}(D', C', F') = \texttt{preferflow}(D, C, F) \times \tilde{\mathcal{S}}$, which implies commutativity on $D'$. Commutativity of the jump map is more straightforward, as, by construction, $\texttt{last}(G') = G_L(\boldsymbol{x}, \boldsymbol{\theta}) \times \{\tilde{\boldsymbol{x}}\} = \texttt{last}(G)(\boldsymbol{x}, \boldsymbol{\theta}) \times \{\boldsymbol{x}\}$. Assumptions (P1-6) (collected in assumption 5) straightforwardly follow after noting, as we have used in the proof of observation 3, that since every topological space is both open and closed in itself (i.e., clopen), any product with such a space as a factor inherits the open (or closed) property from the other factor relative to the product topology. From here, along similar lines argued in the proof of observation 3, properties like graph closure, outer semi-continuity, Borelness, etc. are preserved obviously by construction. $\square$

## C  Static-Time and Do Interventions as Special Cases

As a special case of `instint`, we can also define an intervention that occurs at a fixed, predefined time.[19]

---

[19]Note that, while this definition lets us analyze static-time interventions in this theoretical framework, we do find computational implementations in line with the time-splitting operation (Boeken & Mooij, 2024) more practical.

**Definition 17** (Static-Time Intervention). Consider a parameterized hybrid system $\mathcal{P}$ defined as the tuple $\left(C, F, D, G, \mathbb{R}^2_{\geqslant 0} \times \mathcal{S}, \Theta\right)$. Let time be tracked in the first dimension of the state space, and, in the second dimension, a variable recording whether the intervention has occurred, such that $(t, k, \boldsymbol{x}) \in \mathbb{R}^2_{\geqslant 0} \times \mathcal{S}$. Assume $k = 0$ at $t = 0$ by convention and that $F$ is such that $dk/dt = 0$ always. Let $\tilde{D}(\boldsymbol{\theta}) = [\lambda, \lambda + \epsilon] \times [0, .1] \times \mathcal{S}$ for all $\boldsymbol{\theta} \in \Theta$, a fixed $\lambda \geqslant 0$, and any $\epsilon > 0$.[20] For some $\tilde{G} : \mathcal{S} \times \Theta \rightrightarrows \mathcal{S}$ and all $(t, k, \boldsymbol{x}, \boldsymbol{\theta}) \in \mathbb{R}^2_{\geqslant 0} \times \mathcal{S} \times \Theta$. We then define

$$\widehat{G}\left((t, k, \boldsymbol{x}), \boldsymbol{\theta}\right) = \{t, k + 1\} \times \tilde{G}\left(\boldsymbol{x}, \boldsymbol{\theta}\right) \tag{26}$$

$$\texttt{statint}\left(\mathcal{P}, \lambda, \tilde{G}\right) = \texttt{instint}\left(\mathcal{P}, \tilde{D}, \widehat{G}\right) \tag{27}$$

The definition above, it should be noted, is a special case of a more general "repeated" static-time intervention `rstatint` (definition 19), which is shown to satisfy the same existence, uniqueness, and measurability theory that we establish below for `instint`.

Driving one level more granular, we arrive at a transformation representing something akin to the canonical "do" intervention — again as a special case of `instint`. This notion has been defined for dynamical systems both via a time-splitting mechanism (Boeken & Mooij, 2024) and by casting a continuous time system as its infinitely precise Euler approximation interpreted as an SCM (Hansen & Sokol, 2014).

**Definition 18** (Do-Intervention). Building directly off definition 17, if $\tilde{G}\left(\boldsymbol{x}, \boldsymbol{\theta}\right) = \{\boldsymbol{v}\}$ for some fixed $\boldsymbol{v} \in \mathcal{S}$ and all $\boldsymbol{x}, \boldsymbol{\theta} \in \mathcal{S} \times \Theta$, then for some fixed $\lambda \geqslant 0$ we write

$$\texttt{do}\left(\mathcal{P}, \boldsymbol{x}(\lambda) = \boldsymbol{v}\right) = \texttt{statint}\left(\mathcal{P}, \lambda, \tilde{G}\right) \tag{28}$$

Alternatively, one might wish to fix an index $i \in \{1, \ldots, n\}$ and a value $v \in \mathbb{R}^1$. With $\tilde{G}(\boldsymbol{x}, \boldsymbol{\theta}) = \left[x^{(1:i-1)}, v, x^{(i+1:n)}\right] \forall \boldsymbol{x} \in \mathcal{S}$ and $\forall \boldsymbol{\theta} \in \Theta$, we write instead $\texttt{do}\left(\mathcal{P}, x_t^{(i)} = v\right)$.

These interventional classes form a sort of hierarchy. The jump map of a static-time intervention can be considered the "pre-treatment" model for a do intervention, and the trigger mechanism encoded in the jump set can be considered a pre-treatment model for *when* a static intervention occurs. At the highest level, a state-dependent intervention — especially those that can be triggered many times — can be thought of as a soft intervention on system dynamics. By couching these interventions directly in the language of established hybrid systems theory, we can more easily borrow theoretical results from that vast body of literature.

## C.1 Repeated Static-Time Intervention

**Definition 19** (Repeated Static-Time Intervention). Consider a parameterized hybrid system $\mathcal{P}$ defined as the tuple $\left(C, F, D, G, \mathbb{R}^2_{\geqslant 0} \times \mathcal{S}, \Theta\right)$. Without loss of generality with respect to positioning in the state vector, let time be tracked in the first dimension of the state space, and, in the second dimension, a variable recording whether a specified static intervention has recently occurred, such that $(t, k, \boldsymbol{x}) \in \mathbb{R}^2_{\geqslant 0} \times \mathcal{S}$. Assume $k = 0$ at $t = 0$ by convention and that $F$ is such that $dk/dt = 0$ always. Also, assume that, for some countable set of unique intervention times $\Lambda \subset \mathbb{R}^1_{\geqslant 0}$, there exists an $\epsilon$ such that $0 < \epsilon < \inf \left\{|\lambda_1 - \lambda_2| : \lambda_1, \lambda_2 \in \Lambda^2, \lambda_1 \neq \lambda_2\right\} \cup \{.1\}$. For all $(t, k, \boldsymbol{x}, \boldsymbol{\theta}) \in \mathbb{R}^2_{\geqslant 0} \times \mathcal{S} \times \Theta$

---

[20]The $\epsilon$ construction ensures the jump set is "thick". With a measure-zero jump set, there can exist a solution that reaches the jump set and immediately flows through it, never jumping. A flowing solution is viable from any closed boundary of a jump set where a vector field implied by $F_{\boldsymbol{\theta}}$ points into the flow set. Incidentally, (uniformly) "thick" static-time jump sets also ensure that jumps cannot occur infinitely often. We do not include $\epsilon$ as an argument here, as intervention's behavior is identical regardless of the choice of $\epsilon > 0$. The counter, $k$, is required to avoid repeated jumps from the thick jump following $\lambda$ in time.

and some $\tilde{G} : \mathcal{S} \times \Theta \rightrightarrows \mathcal{S}$, let

$$\tilde{D}_1(\boldsymbol{\theta}) = \left( \bigcup_{\lambda \in \Lambda} [\lambda, \lambda + \epsilon/2] \right) \times [0, .1] \times \mathcal{S} \tag{29}$$

$$\tilde{D}_2(\boldsymbol{\theta}) = \left( \bigcup_{\lambda \in \Lambda} [\lambda + \epsilon/2, \lambda + \epsilon] \right) \times [1, 1.1] \times \mathcal{S} \tag{30}$$

$$\tilde{G}_1\Big( (t, k, \boldsymbol{x}), \boldsymbol{\theta} \Big) = \{(t, k+1)\} \times \tilde{G}(\boldsymbol{x}, \boldsymbol{\theta}) \tag{31}$$

$$\tilde{G}_2\Big( (t, k, \boldsymbol{x}), \boldsymbol{\theta} \Big) = \{(t, k-1)\} \times \tilde{G}(\boldsymbol{x}, \boldsymbol{\theta}) \tag{32}$$

With $\texttt{instint}_i(\cdot) = \texttt{instint}(\cdot, \tilde{D}_i, \tilde{G}_i)$ for $i \in \{1, 2\}$, we can define

$$\mathcal{P}' = \texttt{rstatint}\left(\mathcal{P}, \Lambda, \tilde{G}\right) = (\texttt{instint}_2 \circ \texttt{instint}_1)(\mathcal{P}) \tag{33}$$

**Observation 3** ($\texttt{rstatint}$ Preserves Collected Assumptions). *Continuing from definition 19, if $\tilde{G}$ meets assumption 2, then $\mathcal{P}'$, $\tilde{D}_i$, and $\tilde{G}_i$ meet assumption 2, and theorem 1 would thus apply to rstatint.*

*Proof.* Assumption 2 comprises sub-conditions (I1) and (I2). (I1) first needs that $\tilde{D}_i(\boldsymbol{\theta})$ is closed $\forall \boldsymbol{\theta} \in \Theta$, which follows here from $\tilde{D}_i(\boldsymbol{\theta})$ being a product of closed sets with the topological space $\mathcal{S}$. Since every topological space is both open and closed in itself (i.e., clopen), any product with such a space as a factor inherits the open (or closed) property from the other factor relative to the product topology. Note that the intervals in the unions over intervals constructed from $\lambda \in \Lambda$ are guaranteed to be disjoint and uniformly separated by selecting $\epsilon$ to be positive and smaller than the closest two intervention times, which means the corresponding countable union must be closed. Similarly, by uniform separation, we have the (I1)-required well-behavedness (definition 15) of $\tilde{D}_i(\boldsymbol{\theta})$ relative to $\mathcal{P}$. (I1) also requires that the graph $\mathcal{G}(\text{int } \tilde{D})$ is open. Note that these jump sets are constant in $\Theta$, and therefore their interior graph is the product $\Theta \times A$, where $A \subset \mathbb{R}^n$ is open. $\Theta$ is a topological space, so the product with $A$ inherits the openness of $A$ — thus $\mathcal{G}(\text{int } D)$ is open. By a similar argument, $\tilde{D}_i$ is closed, which implies that $\mathcal{G}(\tilde{D}_i)$ is closed, thereby ensuring $\mathcal{G}(\tilde{D}_i)$ is Borel as required by (I1). (I2) asserts straightforward requirements on $\tilde{G}_i$, none of which are affected by taking a cartesian product with the single-valued, continuous (and therefore both inner and outer semi-continuous) set valued mappings $t, k \mapsto \{(t, k \pm 1)\}$. Theorem 1, then, applies here because rstatint is a composition of instint operations with specifications that meet assumption 2. $\qquad\square$

## D    Proof of Theorem 1

The following proof refers to assumption 5, which is an expanded version of assumption 1 that is referenced by theorem 1 in the main text.

*Proof.* By induction and $K < \infty$, we have via lemma 2 that $\mathcal{P}'$ will meet assumption 5. Note that, by remark 1, if assumption 2 holds relative to $\mathcal{P}$, it will hold relative to any intermediate system in the chain of transformations from $\mathcal{P}$ to $\mathcal{P}'$. Then, by lemma 5, existence, uniqueness, and measurability follow from $\mathcal{P}'$ fulfilling assumption 5. $\qquad\square$

## E    Proof that Instantaneous Intervention Preserves Key Properties

**Lemma 2** (Instantaneous Intervention Preserves Key Properties). *Consider a parameterized hybrid system $\mathcal{P}$ that meets assumption 5. Now, consider set-valued mappings $\tilde{D} : \Theta \rightrightarrows \mathcal{S}$ and $\tilde{G} : \mathcal{S} \times \Theta \rightrightarrows \mathcal{S}$ that fulfill assumption 2 relative to $\mathcal{P}$. The intervened system $\mathcal{P}' = \texttt{instint}(\mathcal{P})$ (definition 4) will then also meet assumption 5, and therefore will have a unique and $t^+$ measurable solution for each $\boldsymbol{\theta} \in \Theta, \boldsymbol{\xi} \in \overline{C(\boldsymbol{\theta})} \cup D(\boldsymbol{\theta})$ according to lemma 5.*

*Proof.* The proof closely follows fig. 2. In assumption 5, we have that $\mathcal{P}$ can be constructed by "lowering" (definition 14) from a system $\mathcal{P}_\uparrow$ that fulfills certain conditions. In lemma 3, we prove that assumptions 2 and 5 imply that the system $\mathcal{P}'$ is equivalent to a system reached by performing a slightly modified intervention on $\mathcal{P}_\uparrow$ (definition 20), and then applying `lower`. The intervention on $\mathcal{P}_\uparrow$ is proven in lemma 4 to preserve properties on the higher system sufficient to say that the lowered system $\mathcal{P}'$ meets assumption 5. Intermediate statements and proofs for lemmas 3 and 4 and definition 20 can be found in appendix E.1. $\qquad\square$

## E.1 Intermediate Results for Lemma 2

Lemma 2 argues that an intervened system $\mathcal{P}' = \texttt{instint}(\mathcal{P})$ can also be constructed by applying a slightly different interventional transformation to a different system : $\mathcal{P}_\uparrow$, and then "lowering" (definition 14). Additionally, if $\mathcal{P}$ meets assumption 5 by way of $\mathcal{P}_\uparrow$, then $\mathcal{P}'$ must also meet assumption 5. This can be established by showing that the intervention on $\mathcal{P}_\uparrow$ preserves properties that allow it to be properly lowered. First, we will define this alternative intervention, then prove commutativity between intervention and lowering, and finally prove that the alternative intervention preserves the properties listed in assumption 5. In the following definition, we use the fact that $G$ is an *ordered set-valued map* (definition 6), which supports appending $\tilde{G}$ to the sequence of maps that compose $G$.

**Definition 20** (Instantaneous Intervention for Higher System)**.** Consider set-valued mappings $\tilde{D}$ : $\Theta \rightrightarrows \mathcal{S}$ and $\tilde{G} : \mathcal{S} \times \Theta \rightrightarrows \mathcal{S}$ and parameterized hybrid system $\mathcal{P}_\uparrow = (C, F, D, G, \mathcal{S}, \Theta)$. Now, let

$$C'(\boldsymbol{\theta}) = C(\boldsymbol{\theta})\backslash\mathrm{int}\,\tilde{D}(\boldsymbol{\theta})$$

$$D'(\boldsymbol{\theta}) = \tilde{D}(\boldsymbol{\theta}) \cup D(\boldsymbol{\theta})$$

$$G_{\tilde{D}}(\boldsymbol{x}, \boldsymbol{\theta}) = \begin{cases} \boldsymbol{x} \in D(\boldsymbol{\theta})\backslash\tilde{D}(\boldsymbol{\theta}) & \mathtt{last}(G)\,(\boldsymbol{x}, \boldsymbol{\theta}) \\ \boldsymbol{x} \in \tilde{D}(\boldsymbol{\theta}) & \tilde{G}\,(\boldsymbol{x}, \boldsymbol{\theta}) \end{cases}$$

$$G_D(\boldsymbol{x}, \boldsymbol{\theta}) = \begin{cases} \boldsymbol{x} \in D(\boldsymbol{\theta}) & G\,(\boldsymbol{x}, \boldsymbol{\theta}) \\ \boldsymbol{x} \in \tilde{D}(\boldsymbol{\theta})\backslash D(\boldsymbol{\theta}) & \tilde{G}\,(\boldsymbol{x}, \boldsymbol{\theta}) \end{cases}$$

$$G' = G_D \sqcup G_{\tilde{D}}$$

then

$$\mathcal{P}'_\uparrow = \texttt{instint}_\uparrow\left(\mathcal{P}_\uparrow, \tilde{D}, \tilde{G}\right) = \left(C', F, D', G', \mathcal{S}, \Theta\right). \tag{34}$$

Since $G'$ is an ordered set-valued map (definition 6), we can derive the following identity, which helps establish some useful intuitions.

$$
\begin{aligned}
G'(\boldsymbol{x}, \boldsymbol{\theta}) &= G_D(\boldsymbol{x}, \boldsymbol{\theta}) \cup G_{\tilde{D}}(\boldsymbol{x}, \boldsymbol{\theta}) \\
&= \begin{cases} \boldsymbol{x} \in D(\boldsymbol{\theta})\backslash\tilde{D}(\boldsymbol{\theta}) & G\,(\boldsymbol{x}, \boldsymbol{\theta}) \\ \boldsymbol{x} \in D(\boldsymbol{\theta}) \cap \tilde{D}(\boldsymbol{\theta}) & G\,(\boldsymbol{x}, \boldsymbol{\theta}) \cup \tilde{G}\,(\boldsymbol{x}, \boldsymbol{\theta}) \\ \boldsymbol{x} \in \tilde{D}(\boldsymbol{\theta})\backslash D(\boldsymbol{\theta}) & \tilde{G}\,(\boldsymbol{x}, \boldsymbol{\theta}) \end{cases}
\end{aligned}
\tag{35}
$$

Below, we additionally use the fact that $\texttt{last}(G') = G_{\tilde{D}}$.

**Lemma 3** (Commutativity of `instint` and `lower`)**.** *Consider parameterized hybrid system $\mathcal{P}_\uparrow = (C, F, D, G, \mathcal{S}, \Theta)$ that meets assumption 5, and set-valued mappings $\tilde{D} : \Theta \rightrightarrows \mathcal{S}$ and $\tilde{G} : \mathcal{S} \times \Theta \rightrightarrows \mathcal{S}$ that meet assumption 2 relative to $\mathcal{P}_\uparrow$. The following equality then holds, with `instint` acting as in definition 4 and $\texttt{instint}_\uparrow$ as in definition 20:*

$$\texttt{instint}\left(\texttt{lower}\left(\mathcal{P}_\uparrow\right), \tilde{D}, \tilde{G}\right) = \texttt{lower}\left(\texttt{instint}_\uparrow\left(\mathcal{P}_\uparrow, \tilde{D}, \tilde{G}\right)\right).$$

*Proof.* First, we adopt the subscript convention, where we use $i$ as a symbol (not a variable) mapping to the intervention operation, and $l$ as a symbol mapping to the lowering operation. The subscript $il$,

for example, indicates a system that has been intervened upon and then lowered. With this convention, we have

$$\mathtt{lower}\,(\mathcal{P}_\uparrow) = \mathcal{P}_l = (C_l, F_l, D_l, G_l, \mathcal{S}_l, \Theta_l)$$

$$\mathtt{instint}_\uparrow\left(\mathcal{P}_\uparrow, \tilde{D}, \tilde{G}\right) = \mathcal{P}_i = (C_i, F_i, D_i, G_i, \mathcal{S}_i, \Theta_i)$$

$$\mathtt{instint}\left(\mathtt{lower}\,(\mathcal{P}_\uparrow), \tilde{D}, \tilde{G}\right) = \mathcal{P}_{li} = (C_{li}, F_{li}, D_{li}, G_{li}, \mathcal{S}_{li}, \Theta_{li})$$

$$\mathtt{lower}\left(\mathtt{instint}_\uparrow\left(\mathcal{P}_\uparrow, \tilde{D}, \tilde{G}\right)\right) = \mathcal{P}_{il} = (C_{il}, F_{il}, D_{il}, G_{il}, \mathcal{S}_{il}, \Theta_{il})$$

We now want to show that every element of the tuple $\mathcal{P}_{li}$ equals to the corresponding element in the tuple $\mathcal{P}_{il}$. We begin with tuple elements that are unaffected by both $\mathtt{lower}$ and $\mathtt{instint}$. These include the parameter space, the state space, and the flow map, meaning we trivially have that

$$(F_{li}, \mathcal{S}_{li}, \Theta_{li}) = (F_{il}, \mathcal{S}_{il}, \Theta_{il}) = (F_l, \mathcal{S}_l, \Theta_l) = (F_i, \mathcal{S}_i, \Theta_i) = (F, \mathcal{S}, \Theta). \tag{36}$$

For the flow set, note that $\mathtt{lower}$ leaves it unmodified and that both the higher and lower overloads of $\mathtt{instint}$ list the exact same transformation on the flow set. Thus $C_{li} = C_{il} = C_i = \boldsymbol{\theta} \mapsto C(\boldsymbol{\theta})\backslash\mathrm{int}\,\tilde{D}(\boldsymbol{\theta})$.

We now show equivalence in the jump map — a largely straightforward effort despite its verbosity. Consider the system $\mathtt{instint}_\uparrow(\mathcal{P}_\uparrow) = (C_i, F, D_i, G_i, \mathcal{S}, \Theta)$. We have that $G_i$ is an ordered set-valued map that, when lowered, yields its last component:

$$G_{il}\,(\boldsymbol{x}, \boldsymbol{\theta}) = \mathtt{last}(G_i)(\boldsymbol{x}, \boldsymbol{\theta}) = \begin{cases} \boldsymbol{x} \in D(\boldsymbol{\theta})\backslash\tilde{D}(\boldsymbol{\theta}) & \mathtt{last}(G)\,(\boldsymbol{x}, \boldsymbol{\theta}) \\ \boldsymbol{x} \in \tilde{D}(\boldsymbol{\theta}) & \tilde{G}\,(\boldsymbol{x}, \boldsymbol{\theta}) \end{cases}. \tag{37}$$

Now we consider the path wherein lowering occurs first. We have that $G_l\,(\boldsymbol{x}, \boldsymbol{\theta}) = \mathtt{last}(G)\,(\boldsymbol{x}, \boldsymbol{\theta})$. By plugging $G_l$ into the definition of $\mathtt{instint}$ for a lowered system (eq. (3)), equivalence between $G_{li}$ and $G_{il}$ becomes clear.

Finally, we show equivalence in the jump set. In what follows, let $C'(\boldsymbol{\theta}) = C(\boldsymbol{\theta})\backslash\mathrm{int}\,\tilde{D}(\boldsymbol{\theta})$ for all $\boldsymbol{\theta} \in \Theta$ and let $\tilde{D} \cup D$ refer to $\boldsymbol{\theta} \mapsto \tilde{D}(\boldsymbol{\theta}) \cup D(\boldsymbol{\theta})$ — we drop explicit dependence generally on $\boldsymbol{\theta}$ for brevity. Also, let the set $V_A = \{\boldsymbol{\xi} \in \mathcal{S} : (\mathrm{VC})\text{ holds for }\boldsymbol{\xi}\text{ relative to flow set }A(\boldsymbol{\theta})\text{ and }F_{\boldsymbol{\theta}}\}$. We can write the "intervention first" path as

$$D_i = \tilde{D} \cup D$$
$$D_{il} = \mathtt{preferflow}\,(D_i, C_i, F_i)$$
$$= \mathtt{preferflow}\left(\tilde{D} \cup D, C', F\right)$$
$$= \left[\tilde{D} \cup D\right]\backslash V_{C'} = \left[\tilde{D}\backslash V_{C'}\right] \cup [D\backslash V_{C'}]$$
$$= \mathtt{preferflow}\left(\tilde{D}, C', F\right) \cup \mathtt{preferflow}\,(D, C', F).$$

Following the "lower first" path and looking to $\mathtt{instint}$ as applied to lowered systems (definition 4), we have

$$D_l = \mathtt{preferflow}\,(D, C, F)$$
$$D_{li} = \mathtt{preferflow}\left(\tilde{D}, C', F\right) \cup D_l$$
$$= \mathtt{preferflow}\left(\tilde{D}, C', F\right) \cup \mathtt{preferflow}\,(D, C, F).$$

Now, note that because $C'(\boldsymbol{\theta}) \subseteq C(\boldsymbol{\theta})$, we have $V_{C'} \subseteq V_C$, and therefore:

$$D_{il} = \mathtt{preferflow}\left(\tilde{D}, C', F\right) \cup \mathtt{preferflow}\,(D, C', F)$$
$$= \left[\tilde{D}\backslash V_{C'}\right] \cup [D\backslash V_{C'}]$$
$$\supseteq \left[\tilde{D}\backslash V_{C'}\right] \cup [D\backslash V_C] = D_{li}.$$

Additionally, by assumption 6 and observation 1, we have that $V_C \subseteq V_{\text{int } \tilde{D}} \cup V_{C'} \subseteq \tilde{D} \cup V_{C'}$, where the second subset relation follows from the closure of $\tilde{D}$ — nothing can "flow into" the interior of $\tilde{D}$ without being in the closure of that interior. This leads to

$$
\begin{aligned}
D_{li} &= \texttt{preferflow}\left(\tilde{D}, C', F\right) \cup \texttt{preferflow}\left(D, C, F\right) \\
&= \left[\tilde{D}\backslash V_{C'}\right] \cup [D\backslash V_C] \\
&\supseteq \left[\tilde{D}\backslash V_{C'}\right] \cup \left[D\backslash\left(\tilde{D} \cup V_{C'}\right)\right] \\
&= \left[\tilde{D}\backslash V_{C'}\right] \cup \left[(D\backslash V_{C'}) \cap \complement\tilde{D}\right] \\
&= \left[\left(\tilde{D}\backslash V_{C'}\right) \cup (D\backslash V_{C'})\right] \cap \left[\left(\tilde{D}\backslash V_{C'}\right) \cup \complement\tilde{D}\right] \\
&= \left[\left(\tilde{D}\backslash V_{C'}\right) \cup (D\backslash V_{C'})\right] \cap \left[\left(\tilde{D} \cup \complement\tilde{D}\right) \cap \left(\complement V_{C'} \cup \complement\tilde{D}\right)\right] \\
&= \left[\left(\tilde{D}\backslash V_{C'}\right) \cup (D\backslash V_{C'})\right] \backslash \left[V_{C'} \cap \tilde{D}\right] \\
&\supseteq \left[\left(\tilde{D}\backslash V_{C'}\right) \cup (D\backslash V_{C'})\right] \backslash V_{C'} \\
&= \left[\tilde{D}\backslash V_{C'}\right] \cup [D\backslash V_{C'}] = D_{il}.
\end{aligned}
$$

With both $D_{li} \supseteq D_{il}$ and $D_{il} \supseteq D_{li}$, it must be that $D_{li} = D_{il}$.

This concludes the proof of equivalence between every element of $\mathcal{P}_{li}$ and $\mathcal{P}_{li}$, meaning $\mathcal{P}_{li} = \mathcal{P}_{il}$. $\qquad\square$

**Lemma 4** (Intervention on Higher System Preserves Key Properties). *Consider parameterized hybrid system $\mathcal{P} = \texttt{lower}(\mathcal{P}_{\uparrow})$ that meets assumption 5, and set-valued mappings $\tilde{D} : \Theta \rightrightarrows \mathcal{S}$ and $\tilde{G} : \mathcal{S} \times \Theta \rightrightarrows \mathcal{S}$ that meet assumption 2 relative to $\mathcal{P}$. Now, consider the following systems:*

$$
\mathcal{P}'_{\uparrow} = \texttt{instint}_{\uparrow}(\mathcal{P}_{\uparrow}, \tilde{D}, \tilde{G}) \tag{38}
$$

$$
\mathcal{P}' = \texttt{lower}\left(\mathcal{P}'_{\uparrow}\right) \tag{39}
$$

*Then $\mathcal{P}'$ satisfies assumption 5.*

*Proof.* We break the proof into six parts, one for each of the preserved assumptions listed in assumption 5. Recall the explicit form of $\mathcal{P}'_{\uparrow}$ given by eq. (34).

**Basic Hybrid Conditions (P1).** To show that $\texttt{instint}_{\uparrow}$ (definition 20) preserves assumption 4, we proceed through the three sub-conditions (A1), (A2), and (A3).

For (A1), we must demonstrate closure of the intervened jump and flow sets. For the flow set, note that by definition 20, $C'(\boldsymbol{\theta}) = C(\boldsymbol{\theta})\backslash\text{int } \tilde{D}(\boldsymbol{\theta})$, and that by (A1) holding for $\mathcal{P}_{\uparrow}$, $C(\boldsymbol{\theta})$ is closed. $C'(\boldsymbol{\theta})$ is thus the result of subtracting an open set from a closed set, and is therefore closed. We have similarly required that $\tilde{D}(\boldsymbol{\theta})$ is closed. Definition 20 specifies that $D'(\boldsymbol{\theta}) = D(\boldsymbol{\theta}) \cup \tilde{D}(\boldsymbol{\theta})$, which is the union of closed sets and therefore closed.

For (A2), note that since the flow map $F_{\boldsymbol{\theta}}$ is unchanged, it trivially remains outer semi-continuous. We then have that $C'(\boldsymbol{\theta}) \subseteq C(\boldsymbol{\theta})$, from which we can conclude that local boundedness relative to $C(\boldsymbol{\theta})$ and convexity of $F(\boldsymbol{x}, \boldsymbol{\theta})$ for every $\boldsymbol{x} \in C(\boldsymbol{\theta})$ implies those properties relative to $C'(\boldsymbol{\theta})$. Additionally, we have that $C'(\boldsymbol{\theta}) \subseteq C(\boldsymbol{\theta}) \subset \text{dom } F_{\boldsymbol{\theta}}$.

Finally, for (A3), we require the outer semi-continuinity of $G'_{\boldsymbol{\theta}}$, its local boundedness relative to $D'(\boldsymbol{\theta})$, and that $D'(\boldsymbol{\theta}) \subset \text{dom } G'_{\boldsymbol{\theta}}$. The following arguments closely mimic the developments in Definition 2.11 and Lemma 2.21 from Sanfelice (2021) — they show that the composition of a hybrid "plant" and hybrid "controller" into a closed loop hybrid system will meet the basic conditions if the plant and controller meet those conditions. In what follows, we work with the identity of $G'$ derived in eq. (35).

Outer semi-continuity of $G'_{\boldsymbol{\theta}}$ means that for every convergent sequence $(\boldsymbol{x}_i) \in D'(\boldsymbol{\theta})$ to $\boldsymbol{x}$ and every convergent sequence $(\boldsymbol{x}_i^+) \in \mathcal{S}$ to $\boldsymbol{x}^+$, where $\boldsymbol{x}_i^+ \in G'(\boldsymbol{x}_i, \boldsymbol{\theta})$ for each $i$, we have that $\boldsymbol{x}^+ \in G'(\boldsymbol{x}, \boldsymbol{\theta})$.

Note that this is equivalent to graph closure. Now, by closure of $D(\boldsymbol{\theta})$, $D'(\boldsymbol{\theta})$, $\mathcal{G}(G_{\boldsymbol{\theta}})$, and $\mathcal{G}(\tilde{G}_{\boldsymbol{\theta}})$, the only potentially problematic limiting points of sequences lying in $D(\boldsymbol{\theta})\backslash\tilde{D}(\boldsymbol{\theta})$, or $\tilde{D}(\boldsymbol{\theta})\backslash D(\boldsymbol{\theta})$, must lie on the intersection $D(\boldsymbol{\theta}) \cap \tilde{D}(\boldsymbol{\theta})$. The intersecting piece, however, returns $G(\boldsymbol{x},\boldsymbol{\theta}) \cup \tilde{G}(\boldsymbol{x},\boldsymbol{\theta})$, which will necessarily contain those limiting points.

Local boundedness of $G'_{\boldsymbol{\theta}}$ relative to $D'(\boldsymbol{\theta})$, then, follows from the local boundedness of $G_{\boldsymbol{\theta}}$ relative to $D(\boldsymbol{\theta})$ and of $\tilde{G}_{\boldsymbol{\theta}}$ relative to $\tilde{D}(\boldsymbol{\theta})$, and the fact that $G_{\boldsymbol{\theta}}$ and $\tilde{G}_{\boldsymbol{\theta}}$ are queried by $G'_{\boldsymbol{\theta}}$ only from the sets on which they are locally bounded.

Finally, we need that $D'(\boldsymbol{\theta}) \subset \mathrm{dom}\ G'_{\boldsymbol{\theta}}$. Recall the piecewise construction of $G'_{\boldsymbol{\theta}}$ in definition 20 and that $D(\boldsymbol{\theta}) \subset \mathrm{dom}\ G_{\boldsymbol{\theta}}$, $\tilde{D}(\boldsymbol{\theta}) \subset \mathrm{dom}\ \tilde{G}_{\boldsymbol{\theta}}$. We can then write the following, where we again drop dependence on $\boldsymbol{\theta}$ for brevity and write $G \cup \tilde{G}$ in place of $\boldsymbol{x} \mapsto G(\boldsymbol{x},\boldsymbol{\theta}) \cup \tilde{G}(\boldsymbol{x},\boldsymbol{\theta})$.

$$\mathrm{dom}\ G' = \left[\left(D\backslash\tilde{D}\right) \cap \mathrm{dom}\ G\right] \cup \left[\left(D \cap \tilde{D}\right) \cap \mathrm{dom}\ G \cup \tilde{G}\right] \cup \left[\left(\tilde{D}\backslash D\right) \cap \mathrm{dom}\ \tilde{G}\right]$$
$$\supset D \cup \left[\left(D \cap \tilde{D}\right) \cap \mathrm{dom}\ G \cup \tilde{G}\right] \cup \tilde{D} = D \cup \tilde{D} = D'.$$

Thus, we have that (A1), (A2) and (A3) are all preserved in $\mathcal{P}'_{\uparrow} = \mathtt{instint}_{\uparrow}(\mathcal{P}_{\uparrow})$, meaning it meets assumption 4.

**Basic Existence (P2).** To show that conditions for proposition 1 are preserved, we recall from the proof of lemma 6 that it is sufficient to show that (VC) is met (with respect to $\mathcal{P}'_{\uparrow}$) for all $\boldsymbol{\xi} \in \overline{C'(\boldsymbol{\theta})}\backslash D'(\boldsymbol{\theta})$ for any $\boldsymbol{\theta} \in \Theta$.

Ignoring whether the flow appropriately remains in the transformed flow set $C'(\boldsymbol{\theta})$, we know by assumption 3 that there must be some $\epsilon > 0$ amount of time from which some continuous function can flow from every $\boldsymbol{\xi} \in \mathcal{S}$ while respecting the differential inclusion. To confirm that (VC) holds at $\boldsymbol{\xi}$ for $\mathcal{P}'_{\uparrow}$, we can check whether some $\epsilon' \in (0, \epsilon]$ exists where $z(t) \in C'(\boldsymbol{\theta})$ for all $t \in (0, \epsilon']$. First, we can decompose the region where (VC) must hold into a union over two cases.

$$\overline{C'(\boldsymbol{\theta})}\backslash D'(\boldsymbol{\theta}) = \left[\mathrm{int}\ C'(\boldsymbol{\theta})\backslash D'(\boldsymbol{\theta})\right] \cup \left[\partial C'(\boldsymbol{\theta})\backslash D'(\boldsymbol{\theta})\right] \tag{40}$$

If $\boldsymbol{\xi} \in \mathrm{int}\ C'(\boldsymbol{\theta})\backslash D'(\boldsymbol{\theta}) \subseteq \mathrm{int}\ C'(\boldsymbol{\theta})$, there must be some such $\epsilon'$ by the openness of $\mathrm{int}\ C'(\boldsymbol{\theta})$ in $C'(\boldsymbol{\theta})$.

We can then decompose the boundary region $\partial C'(\boldsymbol{\theta})\backslash D'(\boldsymbol{\theta})$ as follows, where we've dropped the dependence on $\boldsymbol{\theta}$ for brevity.

$$\begin{aligned}
\partial C'\backslash D' &= \partial\left[C \cap \complement\mathrm{int}\ \tilde{D}\right] \cap \complement\left[\tilde{D} \cup D\right] \\
&= \left[(\partial C \cap \complement\mathrm{int}\ \tilde{D}) \cup (C \cap \partial\complement\mathrm{int}\ \tilde{D})\right] \cap \complement\tilde{D} \cap \complement D \\
&= \left[(\partial C \cap \complement\mathrm{int}\ \tilde{D}) \cup (C \cap \partial\tilde{D})\right] \cap \complement\tilde{D} \cap \complement D \\
&= \left[\partial C \cap \complement\mathrm{int}\ \tilde{D} \cap \complement\tilde{D} \cap \complement D\right] \cup \left[C \cap \partial\tilde{D} \cap \complement\tilde{D} \cap \complement D\right] \\
&= \left[\partial C\backslash\left(\tilde{D} \cup D\right)\right] \cup \left[C \cap \left(\partial\tilde{D}\backslash\tilde{D}\right) \cap \complement D\right] \\
&= \partial C\backslash\left(\tilde{D} \cup D\right) \subseteq \partial C\backslash D.
\end{aligned}$$

The first equality in the final line follows from the assumed closure of $\tilde{D}(\boldsymbol{\theta})$ implying that $\partial\tilde{D}(\boldsymbol{\theta})\backslash\tilde{D}(\boldsymbol{\theta}) = \varnothing$.

By analogous decomposition to eq. (40), we have that $\boldsymbol{\xi} \in \partial C(\boldsymbol{\theta})\backslash D(\boldsymbol{\theta})$ must meet (VC) with respect to $\mathcal{P}_{\uparrow}$. By assumption 6 and observation 1, we then know that the solution must "flow into" either $\mathrm{int}\ \tilde{D}$ or into $C(\boldsymbol{\theta})\backslash\mathrm{int}\ \tilde{D}(\boldsymbol{\theta})$. Because $\tilde{D}$ is closed, flow into its interior requires that $\boldsymbol{\xi} \in \tilde{D}(\boldsymbol{\theta}) \subseteq D'(\boldsymbol{\theta})$, which we need not consider. This leaves only flows into $C(\boldsymbol{\theta})\backslash\mathrm{int}\ \tilde{D}(\boldsymbol{\theta})$, which by construction (definition 20) is equivalent to $C'(\boldsymbol{\theta})$, and therefore satisfies (VC) with respect to $\mathcal{P}'_{\uparrow}$ with $\epsilon'$ as described in observation 1. Thus, $\mathtt{instint}_{\uparrow}$ (definition 20) preserves the conditions for existence as outlined in proposition 1.

**Unique Flowing Solution Everywhere (P3).** Assumption 3 is preserved trivially, since `instint` (definition 20) does not alter $F$, $\Theta$, or $\mathcal{S}$, which are the only system elements involved in assumption 3.

In the remaining results, we use the following observation, leaving its verification from definitions to the reader.

**Observation 4.** *Let $A, B : \Theta \rightrightarrows \mathcal{S}$ be two set-valued maps. Then the graph and set operations commute:*

$$\mathcal{G}(A \backslash B) = \mathcal{G}(A) \backslash \mathcal{G}(B), \quad \mathcal{G}(A \cap B) = \mathcal{G}(A) \cap \mathcal{G}(B), \quad \mathcal{G}(A \cup B) = \mathcal{G}(A) \cup \mathcal{G}(B).$$

**Outer Semi-Continuity of the Flow Set (P4).** We need to show the outer semi-continuity of $C'$ at every $\boldsymbol{\theta} \in \Theta$. Recall from definition 20 that $C'(\boldsymbol{\theta}) = C(\boldsymbol{\theta}) \backslash \operatorname{int} \tilde{D}(\boldsymbol{\theta})$. By observation 4, $\mathcal{G}(C') = \mathcal{G}(C(\boldsymbol{\theta})) \backslash \mathcal{G}(\operatorname{int} \tilde{D}(\boldsymbol{\theta}))$. By assumption 5, we have the outer semi-continuity of $C$, which directly implies the closure of its graph. By assumption 2, we have that $\mathcal{G}(\operatorname{int} \tilde{D}(\boldsymbol{\theta}))$ is open. Thus, $\mathcal{G}(C')$ is closed, and therefore by (Goebel et al., 2012, Lemma 5.10) $C'$ is outer semi-continuous.

**Borel Jump Set Graph (P5).** Recall from definition 20 that $D'(\boldsymbol{\theta}) = \tilde{D}(\boldsymbol{\theta}) \cup D(\boldsymbol{\theta})$. The Borel $\sigma$-algebra is closed under unions, and thus by observation 4 the graph $\mathcal{G}(D')$ must also be Borel.

**Borel Measurable, Single-Valued Jump Map (P6).** We want to show both that $\mathtt{last}(G')(\boldsymbol{x}, \boldsymbol{\theta}) = \{g'(\boldsymbol{x}, \boldsymbol{\theta})\}$ for some $g'$ and all $\boldsymbol{x}, \boldsymbol{\theta} \in \operatorname{dom} \mathtt{last}(G')$ — i.e. that $\mathtt{last}(G')$ is single valued — and that $g'$ is a Borel-measurable function of initial conditions and parameters on the domain of the intervened jump map. By definition 20, we have that

$$\mathtt{last}(G')(\boldsymbol{x}, \boldsymbol{\theta}) = \begin{cases} \boldsymbol{x} \in D(\boldsymbol{\theta}) \backslash \tilde{D}(\boldsymbol{\theta}) & \mathtt{last}(G)(\boldsymbol{x}, \boldsymbol{\theta}) \\ \boldsymbol{x} \in \tilde{D}(\boldsymbol{\theta}) & \tilde{G}(\boldsymbol{x}, \boldsymbol{\theta}) \end{cases} \tag{41}$$

Now, we have by (I2) (assumption 2) that $\tilde{G}$ is single-valued, and by (P6) (assumption 5) that $G$ is single-valued. Thus, there must be some $g'$ such that $\mathtt{last}(G')(\boldsymbol{x}, \boldsymbol{\theta}) = \{g'(\boldsymbol{x}, \boldsymbol{\theta})\}$ on the domain of $\mathtt{last}(G')$.

We now want to show that $g'$ is Borel-measurable for every $\boldsymbol{x}, \boldsymbol{\theta} \in \operatorname{dom} \mathtt{last}(G')$. Note that we can equivalently write the following, where we use the lower-case $\tilde{g}$ and $g$ in reference to the functions that yield the singletons arising from evaluations of $\tilde{G}$ $\mathtt{last}(G)$.

$$g'(\boldsymbol{x}, \boldsymbol{\theta}) = g(\boldsymbol{x}, \boldsymbol{\theta}) \mathbb{I}\Big[\boldsymbol{x} \in D(\boldsymbol{\theta}) \backslash \tilde{D}(\boldsymbol{\theta})\Big] + \tilde{g}(\boldsymbol{x}, \boldsymbol{\theta}) \mathbb{I}\Big[\boldsymbol{x} \in \tilde{D}(\boldsymbol{\theta})\Big]. \tag{42}$$

Note now that the indicator functions involving the jump sets can be written as piecewise functions over a partition defined by the graph of the jump sets. We have assumed that the graphs of $\tilde{D}$ and $D$ are Borel. Further, by observation 4, both indicator functions can be written as piecewise over Borel partitions, meaning they must be Borel measurable. Again, by (I2) (assumption 2) we have that $g$ and $\tilde{g}$ are Borel-measurable. Therefore, $g'$ must also be Borel-measurable.

Having shown the preservation of each sub-condition listed in assumption 5, this concludes the proof. Indeed, if $\mathcal{P}_\uparrow$ meets those sub-conditions, then $\mathcal{P}_\uparrow'$ will as well. In other proofs, this result can be trivially applied to conclude that $\mathtt{lower}(\mathcal{P}_\uparrow')$ fulfills assumption 5. $\qquad\square$

# F Proof that Lowering Induces Existence, Uniqueness, and Measurability

Lemma 1 follows immediately from the following, more precise statement.

**Lemma 5** (Existence, Uniqueness, and Measurability of $\mathcal{P}$). *Consider a parameterized hybrid system $\mathcal{P}$ that meets assumption 5. $\mathcal{P}$, then, fulfills the conditions for basic existence (proposition 1), basic uniqueness (proposition 2), and $t^+$ measurability (definition 12).*

*Proof.* This result follows directly from combining lemma 6 and corollary 2, both stated in the following sections. $\qquad\square$

### F.1 Lowering Preserves Existence and Induces Uniqueness

**Lemma 6** (Lowering Preserves Existence and Induces Uniqueness). *Consider parameterized hybrid system $\mathcal{P}_\uparrow$ that can be lowered (definition 14) to construct a system $\mathcal{P}$ meeting assumption 5. $\mathcal{P}$ then fulfills conditions for basic existence and uniqueness (propositions 1 and 2).*

We split this proof into two components, one for the preservation of existence, and another for the induction of uniqueness.

#### Lowering Preserves Existence

*Proof.* Recall from definition 14 (`lowering`) that $\mathcal{P}' = (C, F, D', G', \mathcal{S}, \Theta)$, with

$$D' = \boldsymbol{\theta} \mapsto D(\boldsymbol{\theta}) \backslash \{\boldsymbol{\xi} \in \mathcal{S} : \text{(VC) holds for } \boldsymbol{\xi}\}. \tag{43}$$

Now, pick some $\boldsymbol{\theta} \in \Theta$ and note that $\overline{C(\boldsymbol{\theta})} = C(\boldsymbol{\theta})$, which follows from the basic conditions on $\mathcal{P}$ asserting that $C(\boldsymbol{\theta})$ is closed. By $\mathcal{P}$ fulfilling proposition 1, we have that every $\boldsymbol{\xi} \in \left[\overline{C(\boldsymbol{\theta})} \cup D(\boldsymbol{\theta})\right] \backslash D(\boldsymbol{\theta}) = C(\boldsymbol{\theta}) \backslash D(\boldsymbol{\theta})$ must meet (VC). It will be sufficient, analogously, to show that every $\boldsymbol{\xi} \in C(\boldsymbol{\theta}) \backslash D'(\boldsymbol{\theta})$ also meets (VC), which is precisely the same condition because `lower` affects neither $C$ nor $F$. Note that

$$
\begin{aligned}
C(\boldsymbol{\theta}) \backslash D'(\boldsymbol{\theta}) =& C(\boldsymbol{\theta}) \backslash \left[D(\boldsymbol{\theta}) \backslash \{\boldsymbol{\xi} \in \mathcal{S} : \text{(VC) holds for } \boldsymbol{\xi}\}\right] \\
=& C(\boldsymbol{\theta}) \cap \complement \left[D(\boldsymbol{\theta}) \cap \complement \{\boldsymbol{\xi} \in \mathcal{S} : \text{(VC) holds for } \boldsymbol{\xi}\}\right] \\
=& C(\boldsymbol{\theta}) \cap \left[\complement D(\boldsymbol{\theta}) \cup \{\boldsymbol{\xi} \in \mathcal{S} : \text{(VC) holds for } \boldsymbol{\xi}\}\right] \\
=& \left[C(\boldsymbol{\theta}) \cap \complement D(\boldsymbol{\theta})\right] \cup \left[C(\boldsymbol{\theta}) \cap \{\boldsymbol{\xi} \in \mathcal{S} : \text{(VC) holds for } \boldsymbol{\xi}\}\right] \\
\subseteq& \left[C(\boldsymbol{\theta}) \backslash D(\boldsymbol{\theta})\right] \cup \{\boldsymbol{\xi} \in \mathcal{S} : \text{(VC) holds for } \boldsymbol{\xi}\}.
\end{aligned}
$$

This yields two cases under which we must check that (VC) holds. For the first case, we know already that (VC) holds for all $\boldsymbol{\xi} \in \left[C(\boldsymbol{\theta}) \backslash D(\boldsymbol{\theta})\right]$. For the second, we have by construction that (VC) holds. By the subset relation, these cases subsume the desired set.

We thus have our sufficient condition, that (VC) is met for any $\boldsymbol{\xi} \in C(\boldsymbol{\theta}) \backslash D'(\boldsymbol{\theta})$. As we have placed no constraints on $\boldsymbol{\theta}$, this holds for the entirety of $\Theta$. $\qquad\square$

#### Lowering Induces Uniqueness

*Proof.* Uniqueness for $\mathcal{P}'$ involves three conditions on the hybrid system as stated in proposition 2, which we will review in reverse order of complexity. Before proceeding, recall from definition 14 (`lowering`) that $\mathcal{P}' = (C, F, D', G', \mathcal{S}, \Theta)$, where for all $\boldsymbol{\theta} \in \Theta$ and $\boldsymbol{x} \in \mathcal{S}$

$$D'(\boldsymbol{\theta}) = D(\boldsymbol{\theta}) \backslash \{\boldsymbol{\xi} \in \mathcal{S} : \text{(VC) holds for } \boldsymbol{\xi}\} \tag{44}$$

$$G'(\boldsymbol{x}, \boldsymbol{\theta}) = \texttt{last}(G)(\boldsymbol{x}, \boldsymbol{\theta}) \tag{45}$$

Recall, also, the convention that $G'_{\boldsymbol{\theta}}(\boldsymbol{x}) = G'(\boldsymbol{x}, \boldsymbol{\theta})$ for all $\boldsymbol{x}, \boldsymbol{\theta} \in \mathcal{S} \times \Theta$. Now, pick some $\boldsymbol{\theta} \in \Theta$.

**Condition (a)** requires the uniqueness of solutions to the differential inclusion on the flow set. This condition is precisely what we have presupposed in assumption 3, except that we make the stronger claim that uniqueness holds for any flow in $\mathcal{S} \supseteq \overline{C(\boldsymbol{\theta})}$.

**Condition (c)** requires that $G'_{\boldsymbol{\theta}}$ is single-valued on the jump set. We have assumed in (P6) that $\texttt{last}(G)$ is no more than single-valued on dom $G$, which implies that it is single valued on dom $G_{\boldsymbol{\theta}}$ for every fixed $\boldsymbol{\theta} \in \Theta$. Additionally, by construction of $D'$ and the basic conditions on $\mathcal{P}$, we have that

$$D'(\boldsymbol{\theta}) \subseteq D(\boldsymbol{\theta}) \subset \text{dom } G_{\boldsymbol{\theta}} = \text{dom } G'_{\boldsymbol{\theta}} \tag{46}$$

Thus, $G'_{\boldsymbol{\theta}}$ is exactly single valued on $D'$.

**Condition (b)** requires that the solution cannot flow from the overlap of the jump and flow sets. Precisely, for every $\boldsymbol{\xi} \in \overline{C(\boldsymbol{\theta})} \cap D'(\boldsymbol{\theta})$, (VC) as used in definition 13 *does not* hold. By assumption 4 on $\mathcal{P}$, we have that $C(\boldsymbol{\theta}) = \overline{C(\boldsymbol{\theta})}$, and recalling definition 13, it is sufficient to show that (VC) does not hold for any $\boldsymbol{\xi}$ in the following set:

$$
\begin{aligned}
\overline{C(\boldsymbol{\theta})} \cap D'(\boldsymbol{\theta}) &= C(\boldsymbol{\theta}) \cap \left[ D(\boldsymbol{\theta}) \backslash \{ \boldsymbol{\xi} \in \mathcal{S} : \text{(VC) holds for } \boldsymbol{\xi} \} \right] \\
&= C(\boldsymbol{\theta}) \cap D(\boldsymbol{\theta}) \cap \complement \{ \boldsymbol{\xi} \in \mathcal{S} : \text{(VC) holds for } \boldsymbol{\xi} \} \\
&= C(\boldsymbol{\theta}) \cap D(\boldsymbol{\theta}) \cap \{ \boldsymbol{\xi} \in \mathcal{S} : \neg\text{(VC) holds for } \boldsymbol{\xi} \} \\
&\subseteq \{ \boldsymbol{\xi} \in \mathcal{S} : \neg\text{(VC) holds for } \boldsymbol{\xi} \} .
\end{aligned}
$$

This concludes the proof. $\qquad\square$

## F.2 Lowering Induces Measurability

We first state sufficient conditions for measurability, and then prove that sufficiency. Ultimately, this yields a corollary stating that lowering induces measurability. We make use of the intermediate results and definitions established in appendix F.3.

**Assumption 7** (Collected Conditions for Measurability). Consider parameterized hybrid system $\mathcal{P} = (C, F, D, G, \mathcal{S}, \Theta)$. Assume that $\mathcal{P}$

(M1)  is $t^+$ uniquely evaluable (definition 10);

(M2)  has a unique solution to its differential inclusion everywhere (assumption 3);

(M3)  has an outer semi-continuous and closed flow set $C(\boldsymbol{\theta})$ at every $\boldsymbol{\theta} \in \Theta$;

(M4)  $G$ is single-valued on dom $G$, with $G(\boldsymbol{x}, \boldsymbol{\theta}) = \{ g(\boldsymbol{x}, \boldsymbol{\theta}) \}$, and $g$ Borel-measurable for all $\boldsymbol{x}, \boldsymbol{\theta} \in$ dom $G$.

**Theorem 2** (Measurability of Solution). *Consider parameterized hybrid system $\mathcal{P} = (C, F, D, G, \mathcal{S}, \Theta)$ and its time-parameterized solution map $\varphi$ (definition 11). If $\mathcal{P}$ meets assumption 7, then $\varphi$ is $t^+$ measurable (definition 12).*

*Proof.* Under assumption 7, finite jump times and values are Borel measurable in $\boldsymbol{\xi}, \boldsymbol{\theta}$ (lemma 7). Additionally, under assumption 3, the solution is Borel-measurable in $\boldsymbol{\xi}, \boldsymbol{\theta}$ up to the first jump (F2-3). We are thus able to write the time-parameterized solution as follows, where $t_0(\boldsymbol{\xi}, \boldsymbol{\theta}) = 0$ always. For all $t \in [0, t^+), \boldsymbol{\xi} \in \mathcal{S}$, and $\boldsymbol{\theta} \in \Theta$:

$$
\varphi(t; \boldsymbol{\xi}, \boldsymbol{\theta}) = \sum_{j=1}^{\infty} \mathbb{I} \left[ t_{j-1}(\boldsymbol{\xi}, \boldsymbol{\theta}) \leqslant t < t_j(\boldsymbol{\xi}, \boldsymbol{\theta}) \right] \phi \left( t - t_{j-1} \big( \boldsymbol{\xi}_{j-1}(\boldsymbol{\xi}, \boldsymbol{\theta}), \boldsymbol{\theta} \big), 0; \boldsymbol{\xi}_j(\boldsymbol{\xi}, \boldsymbol{\theta}), \boldsymbol{\theta} \right) \quad (47)
$$

which comprises a countable sum over Borel-measurable functions of $\boldsymbol{\xi}, \boldsymbol{\theta}$, and is therefore itself Borel measurable. Note that, while we have only shown Borel-measurability for $\boldsymbol{\xi}_j(\boldsymbol{\xi}, \boldsymbol{\theta}) = \phi(t_{j-1}(\boldsymbol{\xi}, \boldsymbol{\theta}), j; \boldsymbol{\xi}, \boldsymbol{\theta})$ when $t_{j-1}(\boldsymbol{\xi}, \boldsymbol{\theta}) < t^+$, the joint requirement that $t_{j-1}(\boldsymbol{\xi}, \boldsymbol{\theta}) \leqslant t < t^+$ avoids those unmeasurable cases. $\qquad\square$

**Corollary 2** (Lowering Induces Measurability). *Consider parameterized hybrid system $\mathcal{P}_\uparrow$ that can be lowered (definition 14) to construct a system $\mathcal{P}$ meeting assumption 5. $\mathcal{P}$ then fulfills conditions for the $t^+$ measurability of its time-parameterized solution map $\varphi$ (definition 11).*

*Proof.* Assumption 5, when combined with the fact that "lowering" (definition 14) induces uniqueness and preserves existence (lemma 6), subsumes or implies conditions sufficient for the result (assumption 7 and theorem 2). In particular, (M2) maps to (P3), (M3) maps to (A1) and (P4), and (M4) maps to (P6). For (M1), note that $t^+$ measurability requires only $t^+ \geqslant 0$ in addition to existence and uniqueness, which come from lemma 6. $\qquad\square$

*Proof of lemma 5.* This result follows directly from combining lemma 6 in appendix F.1 and corollary 2 above. $\qquad\square$

### F.3  Measurability of Jump Times and Values

**Definition 21** (Flowable Region). Consider parameterized hybrid system $\mathcal{P} = (C, F, D, G, \mathcal{S}, \Theta)$. For all $\boldsymbol{\theta} \in \Theta$, let $C_F(\boldsymbol{\theta})$ denote the set of states from which there exist a flowing solution (respecting $F_{\boldsymbol{\theta}}$) that remains in $C(\boldsymbol{\theta})$ after its start. Precisely, this means that there exist $\epsilon > 0$ and an absolutely continuous function $z : [0, \epsilon] \rightarrow \mathcal{S}$ such that $z(0) \in C_F(\boldsymbol{\theta})$ and $\dot{z}(t) \in F(z(t))$ for almost all $t \in [0, \epsilon]$ and $z(t) \in C(\boldsymbol{\theta})$ for all $t \in (0, \epsilon]$.

**Observation 5.** *Consider parameterized hybrid system $\mathcal{P} = (C, F, D, G, \mathcal{S}, \Theta)$ that has a unique solution to its differential inclusion everywhere (assumption 3), and where $C(\boldsymbol{\theta})$ is closed for every $\boldsymbol{\theta} \in \Theta$. In this case, the closure of the flowable region is $\overline{C_F(\boldsymbol{\theta})} = C(\boldsymbol{\theta})$.*

*Proof.* From assumption 3, for every $\boldsymbol{\theta} \in \Theta$, we have that an absolutely continuous function $z$ exists from every $z(0) = \boldsymbol{\xi} \in \mathcal{S} \supseteq C_F(\boldsymbol{\theta})$ that satisfies $F_{\boldsymbol{\theta}}$. Every interior point $\boldsymbol{\xi} \in \text{int } C(\boldsymbol{\theta})$, then, must be in $C_F(\boldsymbol{\theta})$, as some flow must be possible while remaining in $C(\boldsymbol{\theta})$. With the closure of the flow set, we thus have $C(\boldsymbol{\theta}) = \overline{\text{int } C(\boldsymbol{\theta})} \subseteq \overline{C_F(\boldsymbol{\theta})}$. Now, for points $\boldsymbol{\xi} \in \mathcal{S} \backslash \text{int } C(\boldsymbol{\theta})$, note that flow into $C(\boldsymbol{\theta})$ is only possible from $\partial C(\boldsymbol{\theta}) \subseteq C(\boldsymbol{\theta})$. This ensures that $C_F(\boldsymbol{\theta})$ cannot contain points outside of $C(\boldsymbol{\theta})$, further implying that $\overline{C_F(\boldsymbol{\theta})} \subseteq \overline{C(\boldsymbol{\theta})} = C(\boldsymbol{\theta})$. Thus, by a two-sided inclusion, we have $\overline{C_F(\boldsymbol{\theta})} = C(\boldsymbol{\theta})$ for every $\boldsymbol{\theta} \in \Theta$. $\qquad \square$

**Lemma 7** (Measurability of Jump Times and Values). *Consider parameterized hybrid system $\mathcal{P} = (C, F, D, G, \mathcal{S}, \Theta)$ and its solution map $\phi$. If $\mathcal{P}$ meets assumption 7, then the time of the $j > 0$'th jump,*

$$t_j(\boldsymbol{\xi}, \boldsymbol{\theta}) = \sup\{t \mid (t, j-1) \in dom\, \phi(\cdot, \cdot; \boldsymbol{\xi}, \boldsymbol{\theta})\} \in \mathbb{R}_{\geqslant 0} \cup \{\infty\} \tag{48}$$

*is a Borel measurable function of $\boldsymbol{\xi}, \boldsymbol{\theta}$.*

*Additionally, the solution values at these jump times*

$$\boldsymbol{\xi}_{j+1}(\boldsymbol{\xi}, \boldsymbol{\theta}) = \phi(t_j(\boldsymbol{\xi}, \boldsymbol{\theta}), j; \boldsymbol{\xi}, \boldsymbol{\theta}) \tag{49}$$

*are also Borel measurable functions of $\boldsymbol{\xi}, \boldsymbol{\theta}$ if $t_j(\boldsymbol{\xi}, \boldsymbol{\theta}) < t^+$.*

*Proof.* Let $t_1(\boldsymbol{\xi}, \boldsymbol{\theta}) = \sup\{t \mid (t, 0) \in \text{dom } \phi(\cdot; \boldsymbol{\xi}, \boldsymbol{\theta})\}$ be the first jump time. Note that if the set $\{(\boldsymbol{\xi}, \boldsymbol{\theta}) : t_1(\boldsymbol{\xi}, \boldsymbol{\theta}) \geqslant \alpha\}$ is Borel for all $\alpha \in \mathbb{R}$, then $t_1$ must be Borel measurable. Indeed, we can write that set as a countable intersection of Borel sets, which implies Borelness. Below, we use the closure of the "flowable region" $\overline{C_F(\boldsymbol{\theta})}$ (definition 21) to rewrite the pre-image on $\mathcal{S} \times \Theta$ of the first jump occurring at or after time $\alpha$. In particular, we use its closure in order to include the time at which the jump occurs (by including states that flow can reach but not flow from). Note also that, by observation 5, we have that $\overline{C_F(\boldsymbol{\theta})} = C(\boldsymbol{\theta})$ under conditions already provided in assumption 7. We have

$$\{(\boldsymbol{\xi}, \boldsymbol{\theta}) : t_1(\boldsymbol{\xi}, \boldsymbol{\theta}) \geqslant \alpha\} = \left\{(\boldsymbol{\xi}, \boldsymbol{\theta}) : \phi(\tau, 0; \boldsymbol{\xi}, \boldsymbol{\theta}) \in \overline{C_F(\boldsymbol{\theta})} = C(\boldsymbol{\theta}) \; \forall \tau \in [0, \alpha]\right\}$$

$$= \bigcap_{\tau \in \mathbb{Q} \cap [0, \alpha]} \{(\boldsymbol{\xi}, \boldsymbol{\theta}) : (\boldsymbol{\theta}, \phi(\tau, 0; \boldsymbol{\xi}, \boldsymbol{\theta})) \in \mathcal{G}(C)\}$$

Assumption 7 requires that $C(\boldsymbol{\theta})$ is outer semi-continuous at all $\boldsymbol{\theta} \in \Theta$. This holds if and only if its graph $\mathcal{G}(C)$ is closed (Sanfelice, 2021, pg. 49). Closed sets are Borel, so $\mathcal{G}(C)$ must be Borel.

The Borelness of $\{(\boldsymbol{\xi}, \boldsymbol{\theta}) : (\boldsymbol{\theta}, \phi(\tau, 0; \boldsymbol{\xi}, \boldsymbol{\theta})) \in \mathcal{G}(C)\}$ follows from the Borelness of $\mathcal{G}(C)$ and $\phi$ being continuous in $\boldsymbol{\xi}, \boldsymbol{\theta}$ on $[0, t]$, and therefore Borel measurable.

The ability to write the set as a countable intersection over rationals is justified by the standard argument. For any fixed $\tau \in [0, \alpha]$, choose a sequence $(\tau_n) \subset \mathbb{Q} \cap [0, \alpha]$ such that $\tau_n \rightarrow \tau$. This is always possible due to the density of $\mathbb{Q} \cap [0, \alpha]$ in $[0, \alpha]$. If for each $n \in \mathbb{N}$ we have

$$(\boldsymbol{\theta}, \phi(\tau_n, 0; \boldsymbol{\xi}, \boldsymbol{\theta})) \in \mathcal{G}(C), \tag{50}$$

then, because $\phi$ is continuous in time and $\mathcal{G}(C)$ is closed, the limit is also included

$$(\boldsymbol{\theta}, \lim_{n \rightarrow \infty} \phi(\tau_n, 0; \boldsymbol{\xi}, \boldsymbol{\theta})) = (\boldsymbol{\theta}, \phi(\tau, 0; \boldsymbol{\xi}, \boldsymbol{\theta})) \in \mathcal{G}(C). \tag{51}$$

Countable intersections of Borel sets are Borel, and thus $t_1$ must be Borel measurable.

We must now expand from the measurability of the first jump to the measurability of all jumps. We can write the second jump time as follows, with $g$ being a function that, when evaluated, returns the single value of $G$ (M4).

$$\boldsymbol{\xi}_2(\boldsymbol{\xi}, \boldsymbol{\theta}) = g\left(\phi\left(t_1\left(\boldsymbol{\xi}, \boldsymbol{\theta}\right), 0; \boldsymbol{\xi}, \boldsymbol{\theta}\right), \boldsymbol{\theta}\right) \tag{52}$$

$$t_2\left(\boldsymbol{\xi}, \boldsymbol{\theta}\right) = t_1\left(\boldsymbol{\xi}, \boldsymbol{\theta}\right) + t_1\left(\boldsymbol{\xi}_2(\boldsymbol{\xi}, \boldsymbol{\theta}), \boldsymbol{\theta}\right) \tag{53}$$

We have that $g$ is Borel-measurable on the domain of $G$ (M4) and that, by definition of a parameterized hybrid system (definition 1), $D(\boldsymbol{\theta}) \subset \operatorname{dom} G_{\boldsymbol{\theta}}$, and therefore know that $g$ will only be evaluated where it is assumed to be measurable (M4). Additionally, we have that $\phi$ is Borel-measurable for $j = 0$ up to and including $t_1(\boldsymbol{\xi}, \boldsymbol{\theta})$ (F2-3). Thus, $\boldsymbol{\xi}_2$ is the composition of Borel-measurable functions and is therefore itself Borel-measurable. $t_2$, subsequently, is the sum of a Borel-measurable function and the composition of Borel-measurable functions. The measurability of $t_{j+1}$ for $j > 0$ then follows from its recursive form. We use $h^{(n)}(x)$ to represent the $n$-fold composition of $h$ with itself. By standard inductive arguments we have

$$h_{\boldsymbol{\theta}}(x) = g\left(\phi\left(t_1\left(x, \boldsymbol{\theta}\right), 0; x, \boldsymbol{\theta}\right), \boldsymbol{\theta}\right) \tag{54}$$

$$\boldsymbol{\xi}_{j+1}(\boldsymbol{\xi}, \boldsymbol{\theta}) = h_{\boldsymbol{\theta}}^{(j)}(\boldsymbol{\xi}); \quad \boldsymbol{\xi}_1(\boldsymbol{\xi}, \boldsymbol{\theta}) = \boldsymbol{\xi} \tag{55}$$

$$t_{j+1}(\boldsymbol{\xi}, \boldsymbol{\theta}) = \sum_{i=1}^{j+1} t_1(\boldsymbol{\xi}_i(\boldsymbol{\xi}, \boldsymbol{\theta}), \boldsymbol{\theta}) \tag{56}$$

As $t_{j+1}$ comprises only sums of compositions of Borel-measurable functions, it must also be Borel-measurable. Additionally, note that $\boldsymbol{\xi}_{j+1}(\boldsymbol{\xi}, \boldsymbol{\theta}) = \phi(t_j\left(\boldsymbol{\xi}, \boldsymbol{\theta}\right), j; \boldsymbol{\xi}, \boldsymbol{\theta})$ can also be written as the composition of Borel-measurable functions, thereby proving the measurability of jump values. $\quad\square$

## G   Probabilities of Causation and Fishery Management

### G.1   Historical Context for the Fishery Management Problem

Notions of causal necessity and sufficiency are often productively employed in policy discourse, especially where competing interests require human-understandable justifications as to whether a particular policy is sufficient and/or necessary to achieve desired outcomes. Recall the control theoretic settings involving state-dependent, instantaneous interventions that we have enumerated in the introduction: health-related lockdown measures, interest rate adjustments, and many engineering problems involve cost benefit tradeoffs, where policies are designed to be *sufficient* for the benefits, but only as costly as *necessary*. In modern resource management, for example, tragedies of the commons frequently demand a challenging balance between ecological objectives and short and long-term economic outcomes. Additionally, such cases often involve models that our interventional semantics is designed to operate on.

Fishery management offers a particularly rich set of problems where the probabilities of causation can help streamline policy discourse. Over the last few decades, numerous fishery management crises have followed a similar arc: first, growing markets and new technologies result in overfishing to unsustainable biomass levels; then, regulators impose strict catch quotas, gear restrictions, data collection requirements, area closures, and other measures designed to allow stocks to rebuild; after rebuilding stocks, fishing resumes, ideally at more sustainable levels. In 2000, for example, the NMFS and NOAA[21] announced emergency regulatory measures in response to the failure of the Pacific coast groundfish fishery (Anon, 2000). This was followed by an economically tumultuous rebuilding period of around 10 years (Warlick et al., 2018), after which fishing restrictions changed and loosened (Anon, 2010a). Similarly, the 1990s saw significant declines in the Atlantic swordfish fishery (Neilson et al., 2013). In 2000, the ICCAT[22] established an ultimately successful 10-year plan to rebuild the stock (Neilson et al., 2013; Anon, 2010b).

Naturally, these measures were not without significant economic consequences and backlash, both short and long term (Anon, 2000; Cramer et al., 2018; Anon, 2007b). Indeed, in the United States,

---

[21]That is, National Marine Fisheries Service and National Oceanic and Atmospheric Administration.

[22]That is, International Commission for the Conservation of Atlantic Tunas.

the Magnuson-Stevens Act (MSA) mandates the multi-objective of avoiding unnecessary economic sacrifice while pursuing long-term economic and ecological outcomes (Anon, 1976, 1996, 2007a, 2018). Myriad ecological and bio-economic dynamical systems approaches were developed during and after these crises to better balance competing objectives (Lee et al., 2000; Ortiz et al., 2010; Restrepol et al., 2011; Taylor et al., 2022). On some occasions, post-mortems were employed to, for example, determine the degree to which rebuilding success was due to management actions or to natural factors such as species biology Neilson et al. (2013). In essence, the goal of such efforts, as stated in the MSA, is to identify and implement sufficient rebuilding measures that would induce no more economic hardship than necessary.

## G.2 Formal Probabilities of Causation

The formal definitions of the probabilities of causation were originally provided by Pearl (1999). These queries are traditionally defined for binary treatment $X$ and outcome $Y$ variables — we enumerate those binary definitions here, and then develop some intuition. In our fishery management example (section 6), we expand to the non-binary setting in keeping with definitions provided by Kawakami et al. (2024) for scalar treatment and outcome variables.

**Definition 22** (Probabilities of Causation). Let $X, Y$ be binary variables within a structural causal model $M$, and let $x$, $x'$, $y$, $y'$, denote the propositions $X = 1$, $X = 0$, $Y = 1$, $Y = 0$ respectively. Denote by $Y_x$ and $Y_{x'}$ the counterfactual outcomes obtained by performing the do-interventions $\texttt{do}(X = 1)$ and $\texttt{do}(X = 0)$.[23] The probabilities of causation Pearl (1999), then, are defined as follows:

$$PN(x, y) := P(Y_{x'} = 0 \mid x, y), \tag{57}$$

$$PS(x, y) := P(Y_x = 1 \mid x', y'), \tag{58}$$

$$PNS(x, y) := P(Y_x = 1, Y_{x'} = 0). \tag{59}$$

$PN(x, y)$ quantifies the probability that $x$ was necessary to produce outcome $y$; $PS(x, y)$ quantifies the probability that $x$ alone would suffice to produce $y$; and $PNS(x, y)$ jointly quantifies the event that $x$ is both necessary and sufficient for outcome $y$.

To compute the probability of necessity, we consider only (condition on) worlds where the events $x$ and $y$ occurred, and then evaluate the probability of $Y$ being false if we intervene to make $X$ false. Similarly, the capacity to produce an outcome — the probability of sufficiency — is computed by conditioning on $X$ and $Y$ being false, and evaluating the probability of $Y$ being true if we now intervene to make $X$ true. A notion balancing the dimensions of necessity and sufficiency is the probability of necessity and sufficiency, which is *not* a function of the separate probabilities. To evaluate $PNS$, we do *not* condition either way,[24] but rather evaluate the probability that both intervening to make $X$ true results in $Y_x = 1$ and intervening to make $X$ false results in $Y_{x'} = 1$.

## G.3 Multi-Year Horizon

In our analysis of the fishery management problem, we analyze only the year-long time scale, but we *can* define a multi-season model with an arbitrarily long time horizon. Note that, here, we will need to prepend time to the state vector, which becomes $[t, z, h_{1:3}, b_{1:3}] = \boldsymbol{x} \in \mathcal{S} = \mathbb{R}^8_{\geqslant}$. See table 2 for a full labeling of model parameters and states.

First, model the season's starting condition via a jump set that triggers at the beginning of each year. Jumps at the season's start obey a map that (1) resets the integrated catch $z$ to zero and (2) sets fishing harvest rates to their noisy, non-null values. Let $\theta_{h_2} \sim \mathcal{N}(.7, .07)$ and $\theta_{h_3} \sim \mathcal{N}(.07, .007)$ be elements of $\boldsymbol{\theta}$, season-start times be $\mathbb{Z}_{\geqslant 0}$ (i.e. the beginning of each year), and $\texttt{rstatint}$ (definition 19) be a generalization of $\texttt{statint}$ (definition 17) that applies the jump map at countably many times.[25] Let $\mathcal{P}_0 = ((C, F, \varnothing, \cdot), \mathcal{S}, \Theta)$ describe the fishery in its unfished, natural state, where

---

[23]The canonical $\texttt{do}$ intervention $\texttt{do}(X_i = x_i)$ fixes the structural equation for $X_i$ to a constant, i.e. $s_i = x_i$.

[24]Any conditioning here would bias the outcome. Suppose $X$ and $Y$ are causally disconnected. If you condition on $x, y$, you make $Y_x$ true by default, and if you condition on $x', y'$, you make $Y_{x'}$ false, effectively making one of the components satisfied for free, which is undesirable.

[25]Definition 19 also requires a binary auxiliary variable to be part of the state-space — without loss of generality, however, we omit that here for smoother exposition. Additionally, this construction assumes the same

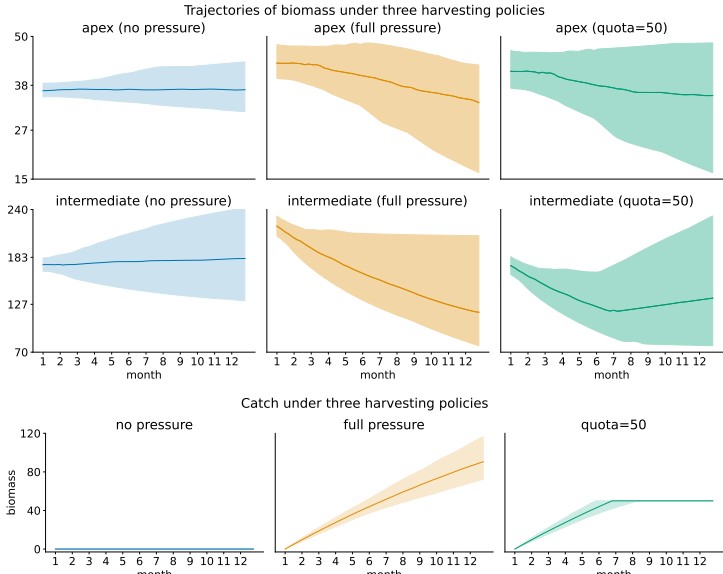

Figure 3: Examples of the biomass trajectories of apex and intermediate predators, as simulated from the model proposed by Zhou & Smith (2017). The panels comprise simulations with, (left) no fishing pressure, (center) fishing pressure kept up throughout the year, (right) fishing regulators ending the season when reported catch meets the total allowable catch (TAC) quota of 50 biomass units.

| name | notation | in season | after season |
|---|---|---|---|
| time | $t$ | | |
| total catch | z | | |
| fishing pressure forage | $h_1$ | 0 | 0 |
| fishing pressure intermediate | $h_2$ | $h_2 \sim \mathsf{Normal}(.7, .07)$ | 0 |
| fishing pressure apex | $h_3$ | $h_3 \sim \mathsf{Normal}(.07, .007)$ | 0 |
| biomass forage | $b_1$ | | |
| biomass intermediate | $b_2$ | | |
| biomass apex | $b_3$ | | |
| desired outcome lower threshold | $\gamma$ | 130 | 130 |
| TAC quota | $q_i$ | | |

Table 2: Parameters and notation for the fishery example.

$\mathcal{S} = C = \mathbb{R}^8_{\geqslant 0}$, and $\cdot$ simply indicates the irrelevance of the jump map in the natural state of the fishery.

$$\tilde{G}_s \left( \boldsymbol{x}, \boldsymbol{\theta} \right) = \{[t, 0, 0, \theta_{h_2}, \theta_{h_3}, b_{1:3}]\} \tag{60}$$

$$\mathcal{P}_s = \mathtt{rstatint} \left( \mathcal{P}_0, \mathbb{Z}_{\geqslant 0}, \tilde{G}_s \right) \tag{61}$$

The season's end can be described by setting the harvest rates to zero when the catch exceeds a threshold $q_i$ (with $i \in \{1, 2\}$). From these, we can construct parallel worlds with the same random initial conditions and parameters.

---

fishing pressure year over year. This can be generalized to independently sampled pressures at each year, though some additional theoretical machinery would be required for an infinite time horizon. See Teel & Hespanha (2015) for one possibility.

$$\tilde{D}_{q_i}(\boldsymbol{\theta}) = \mathbb{R}_{\geqslant 0} \times \{z \in \mathbb{R}_{\geqslant 0} \mid z \geqslant q_i\} \times \mathbb{R}_{\geqslant 0}^6 \tag{62}$$

$$\tilde{G}_{q_i}(\boldsymbol{x}, \boldsymbol{\theta}) = \{[t, z, 0, 0, 0, b_{1:3}]\} \tag{63}$$

$$\mathcal{P}_{q_i} = \texttt{instint}\left(\mathcal{P}_s, \tilde{D}_{q_i}, \tilde{G}_{q_i}\right) \tag{64}$$

$$\mathcal{R}_s = (\mathcal{P}_s, \boldsymbol{\xi}, \boldsymbol{\theta}); \mathcal{R}_{q_1} = (\mathcal{P}_{q_1}, \boldsymbol{\xi}, \boldsymbol{\theta}); \mathcal{R}_{q_2} = (\mathcal{P}_{q_2}, \boldsymbol{\xi}, \boldsymbol{\theta}) \tag{65}$$

### G.4  Narrative Fishery Management Example

In the main body of the paper, we emphasized the construction of the probabilities of causation, rather than their application. Still, some readers may appreciate a more narrative structure around these concepts. We provide that here.

**Example 5** (Probabilities of Causation for Total Allowable Catch (TAC) Quotas)**.** Now, suppose a new commercial fishery is being opened up and that, in the first year, fishery managers allow commercial fishing year-round. $\mathcal{R}_s$ models this world (or equivalently, $\mathcal{R}_{q_1}$ when the TAC quota $q_1$ is sufficiently large so as to have zero probability of being reached). This results in a *failure* to preserve the intermediate level biomass above the desired level $\gamma$. Suppose $\gamma = 130$ units. Facing ecological scrutiny, fishery managers ask: given that we allowed year-round fishing and failed to achieve our outcome, what TAC quota would have a high probability of being sufficient for success? This is a probability of sufficiency query. They introduce a strict TAC quota of 30 units with a relatively high probability of sufficiency (fig. 4). In the next season, they succeed in meeting biomass targets. Subsequently, however, economic interests and local representatives insist that such a low, strict TAC was *not* necessary to achieve this outcome. They point out that, in comparison to a more lenient TAC of 50 units, there is a low probability that the strict TAC of 30 was necessary (fig. 5). Fishery managers, in turn, worry that the probability of success with a TAC of 50 might be too low. Before the start of the next season, stakeholders resolve the disagreement by identifying a TAC that yields a high probability of sufficiency *and* necessity when contrasted with year-round fishing ($\mathcal{R}_s$), all while avoiding stricter catch limitations that are not justified by gains in the probability of necessity and sufficiency (fig. 6).

The simulated results presented here ran on a consumer grade laptop in the order of one hour.

### G.5  Event-Time Attribution

In the main body of the paper (and in example 5), we analyzed the probabilities of causation as they relate to contrastive policies. In other words, we asked causal attribution questions at the policy level. Queries about the probabilities of causation, however, such as "was $x$ necessary to achieve $y$", are ambiguous when a real world event $x$ is multi-faceted and potential alternative actions are plentiful. In example 5, we mapped the events $x$ and $x'$ onto particular TAC quotas. In this next example, however, we will define our event of interest as involving the *time* at which an intervention occurs. We can make this precise by constructing twin worlds using the tools provided in this paper — particularly by additionally employing the static intervention $\texttt{statint}$ (definition 17).

**Example 6** (Probability of Necessity of State-Dependent Intervention Timing)**.** Consider worlds where the season ends before some time $\lambda$, and biomass goal $\gamma$ is achieved at a later time $\tau$ (for instance, at the end of the year). Fishery managers wonder whether they might fail to meet their biomass goals if, contrary to fact in those cases, the season had ended at or after time $\lambda$. The relevant question is: was ending the season before time $\lambda$ necessary for achieving the biomass goal? We can answer this question by asking a probability of necessity query. Unlike in example 5, however, the causal attribution question relates to the *time* at which the intervention occurs. Consider the following binary predicates, where $T(\varphi_{q_1})$ extracts the time at which the season ends due to the TAC quota $q_1$:[26]

$$X = \mathbb{I}\left[T(\varphi_{q_1}) < \lambda\right]; \qquad Y = \mathbb{I}\left[b_{q_1}^\tau \geqslant \gamma\right] \tag{66}$$

---

[26]See appendix I for information and general notation for extracting event times from solutions to hybrid systems. And note that, in lemma 7, we prove that jump times are Borel-measurable in $\boldsymbol{\xi}$ and $\boldsymbol{\theta}$.

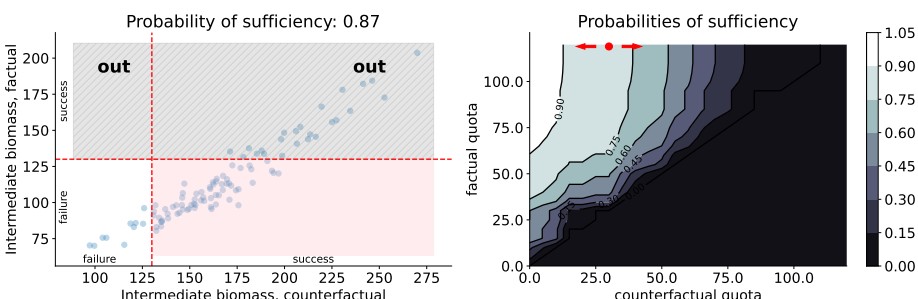

Figure 4: Step 1 in the example narrative (example 5). Within the first year of commercial fishing, the fishery has no quota (here it is enough to set it to $q = 120$, which is never met), and falls below sustainable biomass. Conditioning on this failure, the regulators seek an intervention with a high probability of changing this outcome next time along the counterfactual dimension. They implement a strict TAC of 30, evaluating the probability of sufficiency (the probability of achieving sustainable biomass above the desired threshold of 130) to be 0.87.

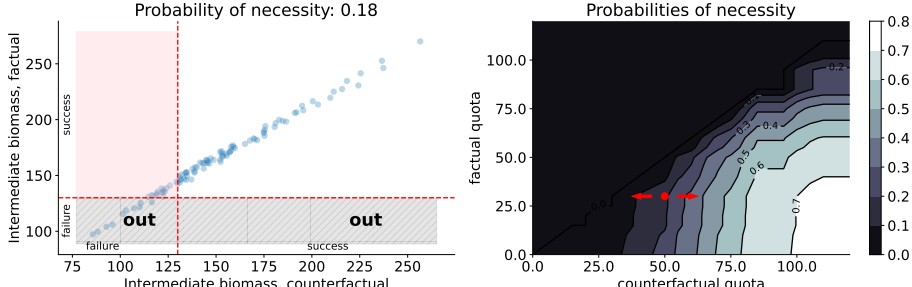

Figure 5: Step 2 in the example narrative (example 5). The season ran with a TAC of 30 units, and the intermediate biomass target reference limit ($\gamma = 130$) was met. Conditioning on this, parties interested in increasing the fishing quota ask whether such a low TAC was *necessary*. They seek an alternative quota along the counterfactual dimension that, when contrasted with the factual TAC of 30, reveals the factual TAC as probably unnecessary. As a counterexample, they choose a TAC of 50, which yields a relatively low probability of necessity (.18).

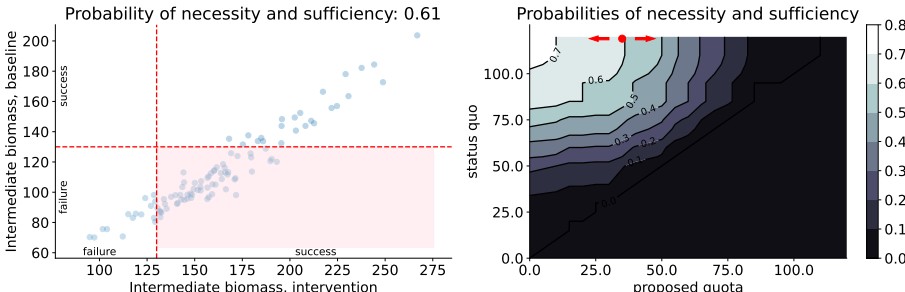

Figure 6: Step 3 in the example narrative (example 5). This time, before the season starts and prior to seeing what the outcome will be, both sides aim to find a quota with a large probability of both necessity and sufficiency. They contrast proposed TAC quotas with a baseline, status quo TAC of 120 units (never met). They notice that the probability surface flattens out above .60, meaning further improvement in the probability of necessity and sufficiency would require excessive limitations in quota. Ultimately, they agree on a quota that results in a value above $0.6$, i.e., a TAC quota of $0.35$.

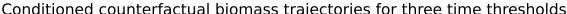

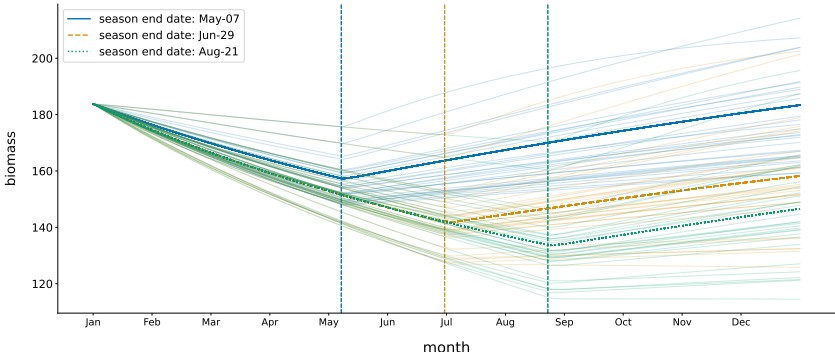

Figure 7: Samples from the Bayesian dynamics based on the fishery model presented by Zhou & Smith (2017), but with the season ending at different times. We show three end dates and their effect on the biomass at the intermediate trophic level.

Recall that the probability of necessity is $P(Y_{\mathtt{do}(X=0)} = 0 \mid X = 1, Y = 1)$ (with shorthand $P(y'_{x'} \mid x, y)$, see table 1). To coherently characterize this in our example, we must define what it means to perform the intervention $\mathtt{do}(X = 0)$. Given our definition of the predicate $X$ above, $\mathtt{do}(X = 0)$ suggests an intervention that results in a world where the season ends at or after $\lambda$, with all else (such as the noise or the resulting fishing pressure of 0) remaining equal. Importantly, there are many such worlds, which means the probability of necessity must adopt an existential flavor: "under exogenous noise where $X = 1$ and $Y = 1$, what is the probability that all worlds consistent with intervention $\mathtt{do}(X = 0)$ fail to meet the outcome?"[27] To precisely define this set of interventional worlds, we build off notation from example 5, and consider a twin world under a *static* intervention occurring at some time $\lambda' \geqslant \lambda$, but with the same interventional jump map utilized in the world $\mathcal{P}_{q_1}$.

$$\{\mathcal{P}_{\lambda'} : \lambda' \geqslant \lambda\} \text{ where } \mathcal{P}_{\lambda'} = \mathtt{statint}\left(\mathcal{P}_s, \lambda', \tilde{G}_{q_1}\right) \tag{67}$$

As described following definition 18, the trigger dynamics of the state-dependent intervention can also be considered a sort of "treatment mechanism" determining the time at which a static intervention occurs. By constructing a world where direct control over the intervention timing is possible, we are able to disentangle these mechanisms. Importantly, note that while $\mathcal{P}_{\lambda'}$ is constructed via a transformation on $\mathcal{P}_s$, it is equivalent to a single-season world constructed from an intervention on $\mathcal{P}_{q_1}$ that directly controls the season-ending time independently of causally upstream events in the system's simulation.

$$Y_{\mathtt{do}(X=0)} = 0 \iff \mathbb{I}\left[\mathtt{b}^{\tau}_{\lambda'} < \gamma\right] \forall \lambda' \geqslant \lambda \tag{68}$$

Note that if $\mathtt{b}^{\tau}_{\lambda'}$ monotonically decreases as $\lambda' \to \infty$, then we can equivalently write the event $y'_{x'}$ as $\mathbb{I}\left[\mathtt{b}^{\tau}_{\lambda'} < \gamma\right]$. Indeed, under our model and distributions on $\xi$ and $\theta$, this is the case, and so we can finally precisely express the probability that ending the season before time $\lambda$ is causally necessary to achieve the biomass outcome:

$$P(y'_{x'} \mid x, y) = P(\mathbb{I}\left[\mathtt{b}^{\tau}_{\lambda'} < \gamma\right] \mid \mathbb{I}\left[T(\varphi_{q_1}) \leqslant \lambda\right], \mathbb{I}\left[\mathtt{b}^{\tau}_{q_1} \geqslant \gamma\right]) \tag{69}$$

Unlike in example 5, conditioning on the factual interventional event *is* required, because knowing that the season ended before $\lambda$ carries information about the model parameters: the earlier the TAC quota is met (at times prior to $\lambda$), the faster the catch rate. Faster catch rates stem from some combination of higher fishing pressure ($h_{1:3}$), higher initial biomass ($b_{1:3}$ at $t = 0$), higher growth

---

[27]This can equivalently be stated: "...probability that there does not exist a world under $\mathtt{do}(X = 1)$ where the outcome is met." See recent work by Li & Pearl (2024) and Kawakami et al. (2024) for more information on how non-binary variables lead to these kinds of existential statements. This also relates to abstract interventions, which has been studied by Beckers & Halpern (2019); Beckers et al. (2019).

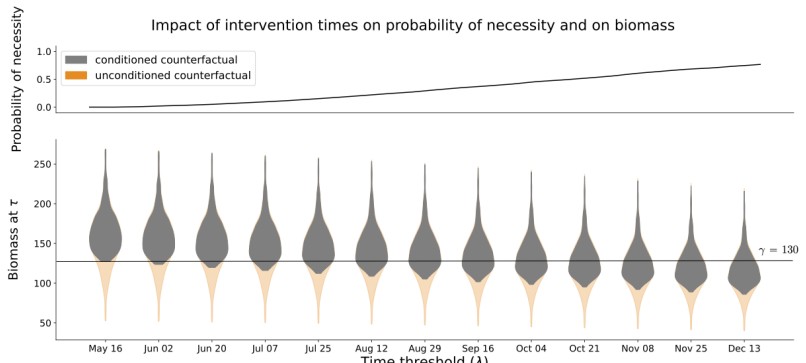

Figure 8: For each $\lambda_i$ from a grid of intervention times we (1) condition on the season ending before $\lambda_i$ and on the successful outcome, and (2) we intervene so that the end of the season occurs at $\lambda_i$. The top panel shows the probability that ending the season before $\lambda_i$ was necessary to achieve biomass targets, while the bottom panel shows the counterfactual biomass distribution under interventions ending the season at various $\lambda_i$. In the violin plot, we differentiate between counterfactual uncertainty for all worlds (orange and gray), and counterfactual uncertainty after selecting only worlds where the TAC quota was reached before $\lambda_i$ *and* biomass goals were met at $\tau$. In other words, the gray violins show biomass probabilities if regulators had ended the season at $\lambda_i$ *in cases where they ended before $\lambda_i$ and met biomass goals.* The probability of necessity, then, is the proportion of the gray distribution falling below the target level of $\gamma$.

rates, etc., all of which influence whether the biomass target will be achieved under alternative season-ending times, even after conditioning on success in the factual world.[28]

Returning to the example, consider a range over fixed threshold $\lambda$, approximated by a finite sequence $(\lambda_i)$. For each $\lambda_i$, we (1) condition on the season ending before $\lambda_i$ and on the intermediate biomass at $\tau$ being above $\gamma$, and (2) intervene so that the end of the season occurs at $\lambda_i$ (and not earlier). The relevant probability of necessity query is whether intervening before $\lambda_i$ was necessary for the success. That is, for each $\lambda_i$ we inspect the posterior predictive distribution of the intermediate biomass at $\tau$ under the intervention, and inspect the probability that this outcome is below $\gamma$. The results of an estimation are available in fig. 8.

The simulated results presented here ran on a consumer grade laptop in the order of one hour.

## H   Holling-Tanner Fishery Model

The fishery management model presented by Zhou & Smith (2017) describes the population dynamics for a given trophic level according to the Holling-Tanner model:

$$\frac{dB}{dt} = rB\left(1 - \frac{B}{K}\right) - MB - FB, \tag{70}$$

where $B$ is the biomass of the species, $r$ is the intrinsic growth rate, $K$ is the carrying capacity, $M$ is the mortality rate due to predation, and $F$ is the fishing mortality rate. Elsewhere in the paper, we have avoided using Zhou & Smith's notation, so-as to avoid overloads with the hybrid system literature. In our paper, we use $h$ instead of $F$, and the lowercase $b$ for biomass, with subscript $i$ indicating trophic level. In this appendix section, however, we will use Zhou & Smith's notation.

---

[28]In other words, we do not necessarily have exogeneity here: $Y_x \not\perp\!\!\!\perp X$; $Y_{x'} \not\perp\!\!\!\perp X$. This further aligns with understanding the state-dependent trigger dynamics as a sort of "treatment mechanism." Exogeneity, here, is violated because ξ and θ can be considered "parents" of both $X$ and $Y$. In violating exogeneity, non-parametric identification results for this example may be out of reach (Li & Pearl, 2024; Kawakami et al., 2024). As this paper does not address non-parametrics, however, a parametric identification of initial conditions and parameters would be sufficient for the identifiability of PNS/PN/PS. Parametric identification in hybrid systems has been studied by Johnson (2023).

The mortality rate due to predation is modeled as:

$$M = \frac{pB_{\text{pred}}}{D + B}, \tag{71}$$

where $p$ is the maximum predation rate, $B_{\text{pred}}$ is the biomass of the predator, and $D$ is the biomass at which predation reaches half its maximum.

The carrying capacity for a predator species is given by:

$$K = eB_{\text{prey}}, \tag{72}$$

where $e$ is the efficiency of converting prey biomass into predator biomass.

The bottom trophic level dynamics follows:

$$\frac{dB_{\text{forage}}}{dt} = r_1 B_{\text{forage}} \left( 1 - \frac{B_{\text{forage}}}{K_1} \right) - M_{12} B_{\text{forage}} - F_{\text{forage}} B_{\text{forage}}, \tag{73}$$

where $M_{12}$ is the mortality rate due to predation from intermediate predators.

Species in the intermediate level act as both predator and prey:

$$\frac{dB_{\text{intermediate}}}{dt} = r_2 B_{\text{intermediate}} \left( 1 - \frac{B_{\text{intermediate}}}{e_{12} B_{\text{forage}}} \right) - M_{23} B_{\text{intermediate}} - F_{\text{intermediate}} B_{\text{intermediate}}. \tag{74}$$

The top trophic level follows:

$$\frac{dB_{\text{apex}}}{dt} = r_3 B_{\text{apex}} \left( 1 - \frac{B_{\text{apex}}}{e_{23} B_{\text{intermediate}}} \right) - M_3 B_{\text{apex}} - F_{\text{apex}} B_{\text{apex}}. \tag{75}$$

The catch rate for the intermediate trophic level is given by the following — note that, in the main body of our paper, we use $z$ for the integrated catch, meaning Catch below corresponds to $\dot{z}$.

$$\text{Catch}_{\text{intermediate}} = F_{\text{intermediate}} B_{\text{intermediate}}. \tag{76}$$

Fishing efforts for each trophic level are assumed to remain constant over time unless intervened on.

$$\frac{dF_i}{dt} = 0, \quad i \in \text{forage, intermediate, apex}. \tag{77}$$

## I Practical Utilities for Tracking Intervention Times and Values

In many counterfactual estimands, we must translate an event's characteristics from one world to another. To do so, we require the ability to extract certain event properties from a hybrid system's solution. By recording event specifications in auxiliary state variables, these can be straightforwardly read off of solution evaluations at any particular time. First, consider an original system $\mathcal{P}$ with time recorded faithfully in the first dimension, and compatible interventional jump set $\tilde{D}$ and jump map $\tilde{G}$. Assume the intervention preserves the faithful recording of time and that $\tilde{G}$ is single valued everywhere.

To start, we augment the state space with an intervention jump counter $j$, an intervention time $t_k$, and an intervention value $v_k$. Our goal is to record the time and jump value corresponding to the $k$'th occurrence of the intervention. Let $\tilde{\mathcal{S}} = \mathcal{S} \times \mathbb{R}_{\geq 0}^2 \times \mathbb{R}$ and $\tilde{\Theta} = \varnothing$, and augment the state space accordingly.

$$\mathcal{P}' = \texttt{spaug}\left( \mathcal{P}, \tilde{\mathcal{S}}, \tilde{\Theta} \right) \tag{78}$$

Now, augment the original interventional specification to appropriately track event details in these auxiliary state variables. For all admissible inputs, and fixed integer $k$, let

$$\tilde{D}'(\boldsymbol{\theta}) = \tilde{D}(\boldsymbol{\theta}) \times \tilde{\mathcal{S}} \tag{79}$$

$$\tilde{G}'\left(\boldsymbol{x}, t, j, t_k, v_k, \boldsymbol{\theta}\right) = \tilde{G}\left(\boldsymbol{x}, t, \boldsymbol{\theta}\right) \times \begin{cases} [j+1, t, v] & k = j+1 \\ [j+1, t_k, v_k] & k \neq j+1 \end{cases} \tag{80}$$

The intervention time and value can then be read directly off of a solution satisfying the constraints of $\texttt{instint}\left(\mathcal{P}', \tilde{D}', \tilde{G}'\right)$. To track whether an event occurred $k$ times, we can initially set, for example, $t_k = -1$. A positive $t_k$ would indicate that the event had occurred. This can be switched to $\infty$ instead for more natural time inequalities if boundedness of the states is not a concern.

## I.1  Notation for Extracting Times and Values

We now describe some general notation for extracting values of $t_k$ and $v_k$ from a solution. Consider a hybrid arc $\phi_i$ satisfying a system arising from the $i$'th $\texttt{instint}$ transformation of that system, where the interventional jump set and map had been augmented to record its $k$'th jump. Given the solution's time parameterization $t \mapsto \varphi(t; \boldsymbol{\xi}, \boldsymbol{\theta})$, we use $T_i^{(k)}\left(\varphi_{m \geqslant i}(\cdot; \boldsymbol{\xi}, \boldsymbol{\theta})\right)$ for the time at which the $k$'th jump occurred, and $V_i^{(k)}\left(\varphi_{m \geqslant i}(\cdot; \boldsymbol{\xi}, \boldsymbol{\theta})\right)$ to extract the state's value immediately following the jump. When clear from context, the function $V$ extracts only one element of the state. The caveat that $m \geqslant i$ simply specifies that properties of the $k$'th jump due to transformation $i$ can be read off of a solution to any further transformed system. As shorthand, we sometimes write $\varphi_{m \geqslant i} = \varphi_{m \geqslant i}(\cdot; \boldsymbol{\xi}, \boldsymbol{\theta})$, taking the random inputs as implicit. Additionally, in settings involving interventions that occur only once in the relevant time window, or where the order of interventional transformation is clear and denoted using a symbolic subscript like $s$, we use, for example, $T_s(\varphi_s)$ to extract the event's time.

## J  State-Dependent Intervention in the Forward Euler Representation

Consider a forward-Euler approximation of a system of ODEs, where, for simplicity, we will assume that $\Delta t > 0$. If $f$ is the right-hand side of the continuous-time differential equation $x' = f(x)$, we can write structural equations $x_t = x_{t-\Delta t} + f(x_{t-\Delta t}, u)\Delta t$, where $t \geqslant 0$, $x_t \in \mathbb{R}$ is the value of the state variable $x$ at time $t$, $u \in \mathbb{R}$ is a fixed realization of exogenous noise (representing unknown parameters, for example), and $x_0$ is fixed to some constant initial condition. Suppose now that we wish to intervene such that the system jumps according to a function $g = x \mapsto x + 1$ when $x$ falls to some threshold $\tau$. To implement this, we must modify the structural equation for $x_t$ for all $t$ under question. That is, we replace the original structural equation with the following piecewise construction.

Let $\bar{x}_t = x_{t-\Delta t} + f(x_{t-\Delta t}, u)\,\Delta t$ denote the value that would be obtained under the original (non-intervened) Euler update, and let $D = \{x \in \mathbb{R} \mid x \leqslant \tau\}$ denote the domain in which the jump is triggered. Then the intervened structural equation can be approximated as follows:

$$x_t = \begin{cases} \bar{x}_t \in D & g(\bar{x}_t) \\ \bar{x}_t \notin D & \bar{x}_t. \end{cases} \tag{81}$$

