# OpenReview forum: "A Counterfactual Semantics for Hybrid Dynamical Systems"
_NeurIPS.cc/2025/Conference — NeurIPS 2025 poster_

### Official Review · Reviewer_o1zp · 2025-06-24

**Clarity:** 4
**Significance:** 2
**Originality:** 4
**Rating:** 4
**Confidence:** 4

**Summary:**

This paper proposes to endow the theory of hybrid systems with the notion of causality, by defining the notion of interventions in this theory. The main result is that this intervention preserves the unique existence of a measurable solution if the original system also had this property.

**Questions:**

In Definition 4, Equation (3), should $D(\theta)$ be $D'(\theta)$?

In Example 3, what is the reason you condition on a normal distribution with mean at the observations and arbitrary variance $\sigma^2$, rather than just at the observations?

**Ethical Concerns:**

["NO or VERY MINOR ethics concerns only"]

**Final Justification:**

I see no reason to change my positive evaluation of the paper after the authors' rebuttals.

**Limitations:**

Yes

**Paper Formatting Concerns:**

No formatting concerns.

**Quality:**

3

**Strengths And Weaknesses:**

**Strengths**

I am a researcher in causality but I am not familiar at all with the theory of hybrid systems, but I found it a refreshing attempt to model causality for dynamical systems in this theory, without the need to rely on SCMs. I am aware of a few works that attempt to extend the theory of SCMs to dynamical systems (or stochastic processes in general), but I have always been somewhat of a sceptic of their success (and indeed, I'm a sceptic of whether these attempts *can* even be successful).

I'm not convinced from this paper that hybrid systems will be *the* theory in which causality should be mathematised for dynamical systems, and in particular, I think a theory of causality that can naturally incorporate a time element should subsume this theory. This theory, from my understanding, is explicitly designed for dynamical systems. However, I still find it an interesting attempt, and give a high evaluation for its originality.

I didn't go through the long proof of Theorem 1 in the appendix, but the result is believable and I trust the maths will be sound. The statement and the purpose of Theorem 1 is clear in the context of the theory the authors are trying to build.

The paper is in general a well-written paper; it was a pleasure to read. I am grateful to the authors for putting effort into the writing.

**Weaknesses**

Despite the fact that this is in general a well-written paper, there are some places where extensive references to the appendix makes the structure hard to follow. The reference to Appendix A.1 on line 112 is fine, as this is just an easy preliminary definition, but I think writing out Definition 6 in the main body rather than just referring to it (e.g. on line 129) would make it easier to follow the mathematics. The same goes for references to Appendices A.3 and A.4 on lines 137 and 138. What is mathematically problematic is using the notation $\mathcal{G}$ on line 178, since this was only introduced in Appendix A.1 in Equation (7) and not defined in the main body. Of course, I understand the space limitations, and the authors are in the best position to decide how to use the space allocated in the main body, but their decision does make it slightly more difficult to follow the maths for those who want to.

Also, for me, Assumption 1 and the subsequent result in Lemma 1 was slightly disappointing. I think I was expecting a stronger result that guarantees unique existence of a solution, rather than requiring the system to be "lowered" from another system that is assumed to have a unique solution.

**Minor Comments**

In Theorem 1, instint needs to be non-italic, to be consistent with the rest of the paper.

In Definition 5, the displayed equation needs a full stop.

In the displayed equation of Example 3, the last semi-colon needs to be replaced by a full-stop. Also, in this displayed equation, in the subscript of the expectation, you use $p$, a notation that hasn't been introduced. Of course it is obvious that this is the distribution of $\xi$ and $\theta$, but still you should introduce notations. Personally, I do not like using random variables as arguments of distributions. Or are you assuming a density here?

The displayed equation in Example 4 also needs a full stop.

---

> ### Author Rebuttal · Authors · 2025-07-31
>
> We thank the reviewer for their interest and engagement, and appreciate that they found the paper a pleasant read! Indeed, regardless of whether hybrid systems should be *the* substrate for dynamical causal theory (or whether there can be a single theory), we certainly hope to have demonstrated some value in bringing the rich literature on hybrid systems to bear for causal aims. This paper represents an initial test of a meta-hypothesis that we may be able to make rapid progress by equipping established modeling paradigms with causal semantics based on an understanding of intervention as model transformation.
>
> ## Appendix References
>
> *The reviewer commented: "...there are some places where extensive references to the appendix makes the structure hard to follow..."*
>
> We sincerely thank the reviewer for identifying some of these flow issues — these can be difficult for authors familiar with the work to pinpoint. Especially with an additional camera-ready page, we expect to be able to resolve a number of the problems you highlighted.
>
> ## Assumption 1/Lemma 1
>
> *The reviewer commented: "...Assumption 1 and the subsequent result in Lemma 1 was slightly disappointing. I think I was expecting a stronger result that guarantees unique existence of a solution, rather than requiring the system to be "lowered" from another system that is assumed to have a unique solution."*
>
> Assumption 1 supposes only that a unique solution exists for the continuous part of the original system. For hybrid systems especially, this is not sufficient for the uniqueness or even existence of non-trivial solutions. The primary logical step taken by Assumption 1/Lemma 1 is to move from the "hybrid basic conditions" (as proposed by Goebel et al. (2012)) to a unique, measurable solution. Importantly, those basic conditions imply neither solution uniqueness nor measurability. We should also point out that the only additions to Assumption 1 over and above the "basic conditions" from Goebel et al. (2012) (see our Assumption 4 for a restatement of their basic conditions) are the unique solution of the differential inclusion, the outer-semi continuity of $C$, and that $\mathcal{G}(D)$ is Borel. Ultimately, these additional assumptions on the original system are quite lenient, and are added primarily for their role in the measurability result.
>
> ## On Minor Comments
>
> *The reviewer commented: "In the displayed equation of Example 3, the last semi-colon needs to be replaced by a full-stop. Also, in this displayed equation, in the subscript of the expectation, you use $p$, a notation that hasn't been introduced. Of course it is obvious that this is the distribution of and $\xi$ and $\theta$, but still you should introduce notations. Personally, I do not like using random variables as arguments of distributions. Or are you assuming a density here?"*
>
> On further review, since we do not use this notation elsewhere in the paper, we will just remove that subscript $p$ and condition on $\boldsymbol{v}_0$ in the expectation.
>
> We will also implement the other suggested corrections described here regarding equation stops, etc.
>
> ## In Definition 4, Equation (3), should $D$ be $D'$?
>
> Using $D$ here simplifies the theory a bit. The required relationship (from Goebel et al.'s basic conditions) is simply that $D'(\theta) \subset \textrm{dom } G'(\cdot, \theta)$, which is satisfied here without requiring that the more complex expression for $D'$ be used in the definition of $G'$. Essentially, the domain of $G'$ might cover regions outside of the jump set, but $G'$ will not be "queried" there.
>
> ## On the Noise Model of Example 3
>
> *The reviewer commented: "In Example 3, what is the reason you condition on a normal distribution with mean at the observations and arbitrary variance $\sigma^2$, rather than just at the observations?"*
>
> Because the state $\boldsymbol{w}_0$ is a deterministic function of initial conditions and parameters, conditioning on it directly leads to collapsed posteriors on those unknowns. In theory, this is fine, but numerically, this is challenging to work with. Many practical models, therefore, include some kind of independent observation noise (ideally, of course, one that maps to a real measurement phenomenon). Our inclusion of the noise model is simply in an effort to better align with practical usage.
>
> This can be resolved other ways, of course — the most robust being to use a hybrid system with stochastic dynamics (i.e. Hybrid SDEs), which we are excited to explore in future work.
>
> ## References
>
> Goebel, R., Sanfelice, R. G., and Teel, A. R. (2012). Hybrid dynamical systems: Modeling, stability, and robustness. Princeton university press.

---

> > ### Comment · Reviewer_o1zp · 2025-08-04
> > **Thank you!**
> >
> > Thank you for the clarifications. I will keep my positive evaluation of the paper, best of luck!

---

### Official Review · Reviewer_2WL1 · 2025-06-26

**Clarity:** 3
**Significance:** 3
**Originality:** 3
**Rating:** 5
**Confidence:** 4

**Summary:**

This paper introduces a new formal semantics for counterfactual inference in hybrid dynamical systems, framing interventions as transformations of the system’s constraints. Under minimal requirements over the interventions, it is proven that these interventions preserve existence, uniqueness, and measurability of solutions. This framework is then applied to a fishery management case study, demonstrating its ability to answer counterfactual questions in hybrid dynamical systems.

**Questions:**

- By “lowering” an “upstream” system into a lower one, does this only mean that the system is adjusted to prefer flowing solutions to solutions with many jumps, or are there other differences as well? Could you clarify how the “preferflow” condition selects flowing solutions over solutions with many jumps, for example, in the case where you might have two solutions like “flow-jump-flow" vs. “flow-flow-jump"?

- To make counterfactuals identifiable, does this framework assume that the outcome and/or functions are monotonic? Since counterfactuals are generally not identifiable without assumptions, but the counterfactual probabilities in the Appendix were point-identified rather than bounded, it seemed to imply that some assumptions were being made?

**Ethical Concerns:**

["NO or VERY MINOR ethics concerns only"]

**Final Justification:**

The authors have provided a detailed response that addressed my questions, and I will maintain my positive score.

**Limitations:**

addressed in weaknesses

**Quality:**

3

**Strengths And Weaknesses:**

## Strengths

- The paper presents a strong theoretic contribution by introducing a novel framework for counterfactual inference in hybrid dynamical systems, which is particularly valuable for comparing and verifying policies within these systems.

- The paper is very well-written and easy to follow. I found the case study especially helpful for understanding how the framework can be applied to real-world models of dynamical systems to estimate probabilities of causation.

- The proofs appear to be comprehensive and detailed, although I have not been able to go through them thoroughly.

## Weaknesses

- The paper currently does not contain a link to the anonymised codebase. However, the authors have stated in the NeurIPS checklist that they will provide this in the camera-ready version.

- It may be beneficial to include some background on causal inference (e.g., definitions of SCMs, interventions, and counterfactuals) for readers less familiar with this work.

- Some of the assumptions made which enable counterfactual identifiability (e.g., monotonicity over outcomes/functions) could be stated explicitly to improve clarity.

---

> ### Author Rebuttal · Authors · 2025-07-31
>
> We thank the reviewer for their kind and critical comments, and are especially pleased that they found the paper easy to follow.
>
> ## On Background
>
> *The reviewer commented: "It may be beneficial to include some background on causal inference (e.g., definitions of SCMs, interventions, and counterfactuals) for readers less familiar with this work."*
>
> Motivated by the reviewer's feedback (and that of 27qb), we will be adding some additional context and background connecting our work to common intuitions relating to SCMs.
>
> ## On the Lowering Operation
>
> Two reviewers made very similar comments, so we have combined our response.
>
> *Reviewer 27qb commented: "...could you elaborate on: 1) the implications of choosing a flow-preferring convention on the causal interpretation of interventions, and 2) whether your theoretical results (especially measurability and uniqueness of solutions under intervention) are robust to alternative resolution strategies?"*
>
> *Reviewer 2WL1 commented: "By “lowering” an “upstream” system into a lower one, does this only mean that the system is adjusted to prefer flowing solutions to solutions with many jumps, or are there other differences as well? Could you clarify how the “preferflow” condition selects flowing solutions over solutions with many jumps...?"*
>
> Our choice of a flow-preferring lowering operation comes largely from its consistency with solver implementations that rely on "crossing events" along the jump set boundary. In such implementations (and in theory), the solution is allowed to flow *along* the boundary as long as it doesn't cross into it. This implies that "grazing" the boundary will not cause a jump. This boundary behavior sums up the implications on interventions. It is also important to note that, in many practical settings, numerical inaccuracies of solutions along set boundaries can dominate the theoretical nuances of whether those act as open or closed.
>
> A jump-preferring approach can be more challenging to compute with (detecting jump-inducing boundary grazes, for example, is a challenge), but does still induce uniqueness (Sanfelice et al., 2023). We conjecture that measurability also holds under a jump-preferring lowering operation, but a formal proof remains future work. Unfortunately, since we rely on a closed flow set in our measurability proof (line 1033), confirming measurability under a jump preferring lowering operation doesn't immediately follow.
>
> ## Counterfactuals and Identifiability Assumptions
>
> *The reviewer commented: "Some of the assumptions made which enable counterfactual identifiability (e.g., monotonicity over outcomes/functions) could be stated explicitly to improve clarity."*
>
> *The reviewer commented: "To make counterfactuals identifiable, does this framework assume that the outcome and/or functions are monotonic? Since counterfactuals are generally not identifiable without assumptions, but the counterfactual probabilities in the Appendix were point-identified rather than bounded, it seemed to imply that some assumptions were being made?"*
>
> Regarding the counterfactual probabilities in the appendix (and indeed any of the estimands we present here, including the PNS quantities), note that any measurable, deterministic function can act as a push-forward of random inputs. The resulting random outputs can then be compared and (conditional) expectations can be taken thereof. Interestingly, as long as $\xi$ and $\theta$ start with a well-defined distribution (e.g., a minimally informed Bayesian prior), causal estimands remain well defined even in non-identified regimes — they may not be particularly informative, however, if they require averaging over irreducible uncertainty. With infinite data in the Bayesian setting, for example, posteriors will reduce to degenerate distributions that amount to "slices" of the prior distribution. Witty et al. (2022) provide further discussion and intuition.
>
> In our case study on the probabilities of causation, we effectively push a prior over $\xi$ and $\theta$ through the measurable solution map, and then through conditional expectations. Importantly, we do *not* first try to recover non-parametric causal relationships — after only observing, for example, a single, factual sequence of events — and then evaluate PNS quantities given our inferences about those relationships. That task does, indeed, require a monotonic relationship between treatment and outcome if point estimates of PNS are desired (Kawakami et al., 2024). Alternatively, as mentioned, one can also average over irreducible uncertainty. Put differently, if you are working directly with the full joint distribution over potential outcomes (like we are here), then you can estimate the PNS quantities directly — non-identification in this setting simply means that some of your uncertainty in that joint distribution cannot be resolved with more data.
>
> We should also clarify a potential point of confusion: while monotonicity is not required to coherently define and evaluate PNS quantities in our setting, it is computationally advantageous when working with the existential predicates present when computing PNS quantities on continuous-valued variables. On line 1239 (in the appendix), for example, we employ monotonicity in the relationship between biomass and season length to redefine an otherwise existential predicate as a single inequality. Without monotonicity, we would instead need to approximate the existential predicate using a sampling procedure and some continuity assumptions.
>
> While identification concerns are outside the scope of this paper, future work is well poised to address the *statistical* question of whether certain estimands (e.g., differences in outcomes between an original and intervened world) can be asymptotically identified given data only from the original world. Naturally, without a coherent counterfactual semantics and proof that intervened outcomes exist and are measurable, this statistical task would not be possible. A promising place to start is with Johnson's (2023) work on the "persistence of excitation", which studies conditions under which hybrid system parameters are identifiable from data.
>
> ## References
>
> Witty, S., Jensen, D., and Mansinghka, V. (2022). SBI: A Simulation-Based Test of Identifiability for Bayesian Causal Inference (arXiv:2102.11761). arXiv. http://arxiv.org/abs/2102.11761
>
> Kawakami, Y., Kuroki, M., and Tian, J. (2024). Probabilities of causation for continuous and vector variables. In N. Kiyavash and J. M. Mooij (Eds.), Proceedings of the fortieth conference on uncertainty in artificial intelligence (Vol. 244, pp. 1901–1921). PMLR. https://proceedings.mlr.press/v244/kawakami24a.html
>
> Johnson, R. (2023). Parameter Estimation for Hybrid Dynamical Systems. University of California Santa Cruz.

---

### Official Review · Reviewer_27qb · 2025-07-02

**Clarity:** 2
**Significance:** 3
**Originality:** 4
**Rating:** 5
**Confidence:** 2

**Summary:**

This work develops a counterfactual semantics for hybrid dynamical systems, which are models that combine continuous dynamics (e.g., physical processes) with instantaneous, state-triggered events (e.g., control policies, thresholds). While these systems are widely used in engineering and applied sciences, existing causal inference tools do not adequately support reasoning about the discontinuous, dynamically triggered interventions these models involve.
To address this gap, the authors formalize interventions as transformations to system constraints, rather than structural equations, and establish conditions under which these interventions preserve key system properties (solution existence, uniqueness, and measurability). This approach enables the use of standard causal estimands, like treatment effects or probabilities of causation, in a hybrid systems context.
A case study on fishery management illustrates the practical use of the framework.

**Questions:**

* It is not clear to me why hybrid system constraints sacrifices immediate graphical identifiability criteria. Is it possible to express the same problems via (parametric) SCMs with different states?

* While I understand that this work introduces a fundamentally different approach, it would be helpful to explicitly clarify why classical frameworkd, particularly those based on SCMs, are inadequate in this context. How does the proposed framework go beyond SCMs and their dynamic extensions? For instance, what is the added value compared to existing approaches such as those by Mooij and collaborators (e.g., Blom & Mooij, 2023; Boeken & Mooij, 2024; Rubenstein et al., 2018) (I understand that these focused on equilibrium, but technically why they cannot be easily extended to the context of this oaoer)? What about threshold-based dynamic SCMs (Zan et al, 2024), or discrete-time dynamic SCMs (Wang et al., 2016; Assaad, 2025)?
In other words, why does modeling continuous-time systems with state-triggered interventions necessitate a new framework? Although this is briefly discussed in the related work section, I believe the paper would benefit from more concrete examples or illustrations that clearly highlight where existing methods fall short and how the proposed approach overcomes those limitations. This would greatly help readers appreciate the significance and practical impact of the contribution.


* In your framework, you adopt a flow-preferring resolution to handle solution non-uniqueness at the boundary of the flow and jump sets (Definition 3). Given that other strategies like jump-preferring are also used (Teel & Hespanha, 2015), could you elaborate on: 1) the implications of choosing a flow-preferring convention on the causal interpretation of interventions, and 2) whether your theoretical results (especially measurability and uniqueness of solutions under intervention) are robust to alternative resolution strategies?


* I think (but not sure) that the citation in line 59 should be replaced by Shpitser and Pearl, 2008. In their 2006 paper they focused on interventional distributions whereas in their 2008 papers they extended the results to counterfactuals.

* The authors also point out the limitation that they focuss on a finite time regimes. I do not see this as a real limitation, as I think in most systems the number of regimes are finite. Do the authors agree or am I missing something? DO think most systems have finite or infinite regimes?
Nevertheless, extending this work to infinite time regimes is an interesting future work ....


References:

* T. Blom and J. Mooi. Causality and independence in perfectly adapted dynamical systems. 2023

* P. Boeken, J. Mooij. Dynamic Structural Causal Models, 2024

* P. Rubenstein, S. Bongers, B. Schoelkopf, J. Mooij, From Deterministic ODEs to Dynamic Structural Causal Models, 2018

* L. Zan, C. Assaad, E. Devijver, E. Gaussier, A. Aït-Bachir. On the Fly Detection of Root Causes from Observed Data with Application to IT Systems, 2024

* Z. Wang, Y. Liang, D. Zhu and T. Li. The Relationship of Discrete DCM and Directed
Information in fMRI based Causality Analysis, 2016

* C. Assaad. Towards identifiability of micro total effects in summary causal graphs with latent confounding: extension of the front-door criterion, 2025

**Ethical Concerns:**

["NO or VERY MINOR ethics concerns only"]

**Final Justification:**

The authors addressed all my concerns. I was convinced and I’m now even more convinced of the relevance and value of this work.

**Limitations:**

Yes

**Paper Formatting Concerns:**

No concerns

**Quality:**

3

**Strengths And Weaknesses:**

Strengths:

The work bridges causal inference and hybrid systems theory, enabling more reliable and expressive causal reasoning in settings involving closed-loop control and real-time decision-making.

Weaknesses:

In my opinion this paper does not motivate enough the need of this work, it would be interesting to explain more why other approches fail in this context. In addition, as it is acknowledged by the authors, graphical identifiability criteria are not provided and unlike classical graphical criteria, the criteria given int his paper are parametric.
Moreover, I think, for most of the audience of the conference, it would be very complicated to understand the paper without going through the appendix.

---

> ### Author Rebuttal · Authors · 2025-07-31
>
> We thank the reviewer for the positive feedback and for pushing us to improve the motivation — their critiques have helped us to further clarify the ways in which this contribution relates to existing work. We'll address the motivation questions first.
>
> ## Motivation
>
> *The reviewer commented: "While I understand that this work introduces a fundamentally different approach, it would be helpful to explicitly clarify why classical frameworks, particularly those based on SCMs, are inadequate in this context...what is the added value compared to existing approaches such as those by Mooij and collaborators...? What about threshold-based dynamic SCMs (Zan et al, 2024), or discrete-time dynamic SCMs (Wang et al., 2016; Assaad, 2025)? In other words, why does modeling continuous-time systems with state-triggered interventions necessitate a new framework? ..."*
>
> This work builds a bridge between two very mature fields, and making such a connection necessitates a natural choice: should we extend existing causal frameworks or hybrid systems frameworks? Primarily motivated by the specific challenges we hoped to address, we chose the latter for reasons we discuss below. That said, there certainly are useful connections to be made to existing causal frameworks.
>
> Overall, we thank the reviewer for pressing for this context, and agree that it will be quite helpful to our readers. We will expand on these comparisons in our related works section.
>
> **Extending (Discrete Time) SCMs?**
>
> *The reviewer commented: "...[why do] hybrid system constraints [sacrifice] immediate graphical identifiability criteria. Is it possible to express the same problems via (parametric) SCMs with different states?"*
>
> Notwithstanding the challenges associated with discretizing continuous time systems (e.g., from stiffness), discrete time structural causal models (such as those employed by Zan et al. (2024), Wang et al. (2016), and Assaad (2025)) do afford an instructive mapping to the continuous time domains. As Hansen and Sokol (2014) point out, the discrete time setting can be viewed as an Euler approximation to a continuous state trajectory. Under this interpretation (assume we take it to the limit with infinitesimally small $\Delta t$), we can write an infinite sequence of structural equations, where $f$ is the right-hand side of the continuous time differential equation $x' = f(x)$:
>
> $$x_t = x_{t-\Delta t} + f\left(x_{t-\Delta t},\theta\right) \Delta t.$$
>
> A static-time intervention with a state-dependent jump value (though not a state-dependent jump *time*), then, can be encoded as a soft intervention (Correa and Barenboim, 2020) that swaps the above structural equation with one governed by a difference (as opposed to differential) equation:
>
> $$x_t = g\left(x_{t-\Delta t},\theta\right).$$
>
> One can even encode a *state-dependent* intervention, but it requires a soft-intervention at every point in time. Consider:
>
> $$
> x_t =
> \begin{cases}
> g(x_{t - \Delta t}, \theta) & \text{if } x_{t - \Delta t} \in D(\theta) \\\\
> x_{t - \Delta t} + f(x_{t - \Delta t}, \theta)\ \Delta t & \text{if } x_{t - \Delta t} \in C(\theta)
> \end{cases}
> $$
>
> We'll highlight two downsides here. First, representing a state-dependent intervention as a soft intervention affecting all endogenous nodes is challenging to work with under graphical theory. Secondly, the Euler representation is not the typical choice of hybrid systems theorists. This makes post-intervention solution existence, uniqueness, stability, etc. challenging to prove, without much to gain from the existing causal inference literature.
>
> **Extending Constraint-Based Causal Models?**
>
> Recent developments in constraint-based causal modeling offer another potential starting point.
>
> For example, Beckers et al. (2023) extend SCMs in order to handle logical constraints (such as unit conversions), while Blom et al. (2019) interpret equilibrium equations of dynamical systems, along with their corresponding algebraic invariants, as a collection of constraints. In both cases, a model is characterized by a collection of constraints, and interventions are defined as transformations of those constraints (e.g., by changing, disabling, or enabling them). At a high level, our work operates very similarly.
>
> Hybrid systems, however, are characterized by a unique class of constraints requiring special considerations around Zeno behavior, set-valued theory, non-uniqueness even in "well-posed" cases, set-valued stable points, etc. In short, analyzing the complexities of hybrid systems is made easier via direct use of existing hybrid systems frameworks, rather than existing causal frameworks.
>
> That said, we ultimately consider our work to be complementary to that of Mooij and colleagues (and other constraint based causal machinery), and we look forward to future integrations of that developing framework with work on theory, simulation, and inference involving hybrid systems.
>
> ## Flow Preference and Lowering
>
> Two reviewers made very similar comments, so we have combined our response.
>
> *Reviewer 27qb commented: "...could you elaborate on: 1) the implications of choosing a flow-preferring convention on the causal interpretation of interventions, and 2) whether your theoretical results (especially measurability and uniqueness of solutions under intervention) are robust to alternative resolution strategies?"*
>
> *Reviewer 2WL1 commented: "By “lowering” an “upstream” system into a lower one, does this only mean that the system is adjusted to prefer flowing solutions to solutions with many jumps, or are there other differences as well? Could you clarify how the “preferflow” condition selects flowing solutions over solutions with many jumps...?"*
>
> Our choice of a flow-preferring lowering operation comes largely from its consistency with solver implementations that rely on "crossing events" along the jump set boundary. In such implementations (and in theory), the solution is allowed to flow *along* the boundary as long as it doesn't cross into it. This implies that "grazing" the boundary will not cause a jump. This boundary behavior sums up the implications on interventions. It is also important to note that, in many practical settings, numerical inaccuracies of solutions along set boundaries can dominate the theoretical nuances of whether those act as open or closed.
>
> A jump-preferring approach can be more challenging to compute with (detecting jump-inducing boundary grazes, for example, is a challenge), but does still induce uniqueness (Sanfelice et al., 2023). We conjecture that measurability also holds under a jump-preferring lowering operation, but a formal proof remains future work. Unfortunately, since we rely on a closed flow set in our measurability proof (line 1033), confirming measurability under a jump preferring lowering operation doesn't immediately follow.
>
> ## Finite Time Regimes
>
> *The reviewer commented: "...I do not see [the focus on finite time regimes] as a real limitation, as I think in most systems the number of regimes are finite...extending this work to infinite time regimes is an interesting future work."*
>
> In the context of dynamical systems, "finite time" essentially refers to "transient" as opposed to "equilibrium" regimes. We agree with the reviewer that this is more a matter of paper scope than a limitation. Indeed, asymptotic stability results already exist that rely primarily on the "hybrid basic conditions" we borrow from Goebel et al. (2012) (see our Assumption 4 in the appendix, and Ch. 3 by Goebel et al.). One challenge here is that hybrid systems often equilibrate to periodic behavior/sets rather than to points. In fact, "non-constant asymptotic dynamics" have been considered already for non-hybrid systems by Rubenstein et al. (2018). In sum, proper handling of infinite time/equilibrium regimes for hybrid system counterfactuals is well within striking distance of future work.
>
> ## Citation Issue
> *The reviewer commented: "...the citation in line 59 should be replaced by Shpitser and Pearl, 2008..."*
>
> Yes, this is correct. The 2008 paper also further develops the twin network setting, which aligns with understanding intervention as transformation.
>
> ## References
>
> L. Zan, C. Assaad, E. Devijver, E. Gaussier, A. Aït-Bachir. On the Fly Detection of Root Causes from Observed Data with Application to IT Systems, 2024
>
> Z. Wang, Y. Liang, D. Zhu and T. Li. The Relationship of Discrete DCM and Directed Information in fMRI based Causality Analysis, 2016
>
> C. Assaad. Towards identifiability of micro total effects in summary causal graphs with latent confounding: extension of the front-door criterion, 2025
>
> Hansen, N. and Sokol, A. (2014). Causal interpretation of stochastic differential equations. Electronic Journal of Probability, 19(none). https://doi.org/10.1214/EJP.v19-2891
>
> Correa, J. and Bareinboim, E. (2020). General transportability of soft interventions: Completeness results. Advances in neural information processing systems (R-68; Vol. 33, pp. 10902–10912).
>
> Beckers, S., Halpern, J. Y., and Hitchcock, C. (2023). Causal Models with Constraints. https://proceedings.mlr.press/v213/beckers23a/beckers23a.pdf
>
> Blom, T., Bongers, S., and Mooij, J. M. (2019). Beyond Structural Causal Models: Causal Constraints Models (arXiv:1805.06539).
>
> Sanfelice, R., Wintz, P., Copp, D., and Nanez, P. (2023). Behavior in the Intersection of C and D. https://hyeq.github.io/simulink/intersection-of-C-and-D
>
> Goebel, R., Sanfelice, R. G., and Teel, A. R. (2012). Hybrid dynamical systems: Modeling, stability, and robustness. Princeton university press.
>
> Rubenstein, P. K., Bongers, S., Schoelkopf, B., and Mooij, J. M. (2018). From Deterministic ODEs to Dynamic Structural Causal Models (arXiv:1608.08028).

---

> > ### Comment · Reviewer_27qb · 2025-08-02
> >
> > Thank you for your thorough and detailed response. I believe you have addressed all of my questions clearly and convincingly. I’m now even more convinced of the relevance and value of this work.

---

### Official Review · Reviewer_cV92 · 2025-07-03

**Clarity:** 3
**Significance:** 4
**Originality:** 4
**Rating:** 5
**Confidence:** 1

**Summary:**

This paper presents a new intervention semantics for hybrid dynamical systems where the dynamics is continuous but the changes to the dynamics can be discrete. The main contribution is developing a formal intervention semantics in a hybrid dynamical system, as the changes in the constraints due to the intervention. Many of the mathematical contents addressing the formalism are provided in the Appendix. The causal effects and the counterfactuals are also defined in the examples in Section 5. Application to counterfactual inference is presented in a case study and the Appendix.

**Questions:**

In the introduction, the HVAC system is mentioned as an example of a hybrid dynamical system.
Could you provide a brief hypothetical scenario and example to illustrate how the proposed ideas and theory align?
Alternatively, it could involve continuing the dosage example from Example 1.
I am unable to follow all the new notations in the paper and the meanings of the mathematical definitions.

Theorem 1 looks interesting. Can you explain it in practical terms?

For the counterfactual outcome in Example 4, what are the assumptions around unobserved confounders in the hybrid dynamical systems?

**Ethical Concerns:**

["NO or VERY MINOR ethics concerns only"]

**Final Justification:**

I will keep the current score.

**Limitations:**

YES

**Quality:**

3

**Strengths And Weaknesses:**

**Strength**
The main strength of this paper is in its theoretical contribution to developing a new intervention semantics for hybrid dynamical systems.

**Weakness**
Due to the emphasis on the theory, I think this paper is too abstract and not suitable for a general AI/ML audience.

---

> ### Author Rebuttal · Authors · 2025-07-31
>
> Thanks to the reviewer for their questions and feedback. Below, we address the reviewer's questions and concerns in the order they appear in the review.
>
> ## On Our Theoretical Emphasis
>
> *The reviewer commented: "Due to the emphasis on the theory, I think this paper is too abstract and not suitable for a general AI/ML audience."*
>
> We believe an important practical contribution of our work is to create a conceptual bridge between previously disconnected research communities, which benefits both. For the causal inference community, our work extends the reach of causal inference to widely used hybrid dynamical systems. For the hybrid dynamical systems community, our work brings a taxonomy of causal questions, and a mechanized procedure for answering those questions from models and data.
>
> Beyond that, our theoretical results serve immediate practical ends. For example, without ensuring the existence and measurability of intervened outcomes, one can neither write coherent causal estimands nor reliably implement them in software (see below for discussion of theorem 1, which may help clarify this). We also note that our results apply to any parametric hybrid system fitting our definition, which includes those that characterize the flow map with neural ODEs (Chen et al., 2018).
>
> ## On HVAC and Dosage
>
> *The reviewer commented: "In the introduction, the HVAC system is mentioned as an example of a hybrid dynamical system. Could you provide a brief hypothetical scenario and example to illustrate how the proposed ideas and theory align? Alternatively, it could involve continuing the dosage example from Example 1. I am unable to follow all the new notations in the paper and the meanings of the mathematical definitions."*
>
> Both the HVAC and dosage cases involve a thresholding behavior. When the state reaches a particular threshold (temperature/drug concentration, respectively), the state is instantaneously modified (the cooler turns on/a dosage is administered). Such thresholding behavior can be encoded as a transformation of the constraints that characterize a hybrid system. If the transformation is used to construct parallel worlds — some with the thresholding behavior, and others without — then outcomes in those worlds can be compared using standard causal inference machinery. In other words, we can coherently reason about the effects of an HVAC controller even though *when* the controller activates may depend on exogenous randomness. We will ensure these intuitions are communicated in a way that more clearly connects the dosage example to those enumerated in the introduction (including the HVAC example).
>
> ## On Theorem 1
>
> *The reviewer commented: "Theorem 1 looks interesting. Can you explain it in practical terms?"*
>
> Theorem 1 states that a finite composition of interventions (defined here as a particular class of transformation to the constraints that define a hybrid system) will maintain conditions sufficient for the existence, uniqueness, and measurability of the intervened system's solution. This is a key requirement when comparing outcomes across original and intervened worlds. If the intervened world's outcome does not exist, for example, that comparison is not possible. Similarly, if the outcomes are not measurable, we cannot coherently define distributions on those outcomes.
>
> ## On Example 4 and Unobserved Confounding
>
> *The reviewer commented: "For the counterfactual outcome in Example 4, what are the assumptions around unobserved confounders in the hybrid dynamical systems?"*
>
> Because a hybrid system's solution is a deterministic function of initial conditions $\xi$ and system parameters $\theta$ (note that a structural equation is also deterministic with respect to its inputs), the conditional $\xi, \theta \mid y_0^\tau = \bar{y}_0^\tau$ will often result in a degenerate distribution — for example, a delta in the identified case, or perhaps a Gaussian with a singular covariance matrix in a non-identified case. Importantly, even in the non-identified case, the expectation remains well defined due to Theorem 1. Counterfactual conditionals behave similarly in standard acyclic SCMs. Because we do not address identification in this paper, we omit discussion of unmeasured confounding (or other sources of non-identification) for these examples, instead emphasizing the coherence of the estimands as a consequence of Theorem 1. In practice, of course, you would often need to condition on additional data or have sufficient domain knowledge in order to properly constrain the problem.
>
> Regarding unmeasured confounding specifically — while that is a common source of non-identification in acyclic structural causal models, a more frequent concern in hybrid systems relates to the "persistence of excitation" (Johnson, 2023). In short, you often need to observe *transient* dynamics (as opposed to equilibrated dynamics) in order to identify the effects of interventions. Of course, latent state elements also cause problems for identification here, though the deterministic coupling of the state elements in a hybrid system requires careful handling.
>
> For further discussion regarding identification, see the heading "Counterfactuals and Identifiability Assumptions" in our response to reviewer 2WL1.
>
> ## References
>
> Chen, R. T. Q., Rubanova, Y., Bettencourt, J., and Duvenaud, D. K. (2018). Neural ordinary differential equations. Advances in Neural Information Processing Systems, 31. https://proceedings.neurips.cc/paper_files/paper/2018/file/69386f6bb1dfed68692a24c8686939b9-Paper.pdf
>
> Johnson, R. (2023). Parameter Estimation for Hybrid Dynamical Systems. University of California Santa Cruz.

---

### Decision · Program_Chairs · 2025-09-17

**Decision:**

Accept (poster)

**Comment:**

As emphasized by reviewers, this paper make an interesting attempt at connecting hybrid dynamical systems to causality by introducing a well grounded counterfactual semantics. Despite the formalism required by this kind of work, reviewers have appreciated the clarity of the presentation and the potential it has to bridge the field of causality and dynamical systems.